# Ambient aerosol properties in the remote atmosphere from global-scale in-situ measurements

Charles A. Brock[1], Karl D. Froyd[1,2], Maximilian Dollner[3], Christina J. Williamson[1,2], Gregory Schill[1,2], Daniel M. Murphy[1], Nicholas J. Wagner[1,2], Agnieszka Kupc[3], Jose L. Jimenez[2,4], Pedro Campuzano-Jost[2,4], Benjamin A. Nault[2,4], Jason C. Schroder[2,4], Douglas A. Day[2,4], Derek J. Price[2,4], Bernadett Weinzierl[3], Joshua P. Schwarz[1], Joseph M. Katich[1,2], Siyuan Wang[1,2], Linghan Zeng[5], Rodney Weber[5], Jack Dibb[6], Eric Scheuer[6], Glenn S. Diskin[7], Joshua P. DiGangi[7], ThaoPaul Bui[8], Jonathan M. Dean-Day[9], Chelsea R. Thompson[1,2], Jeff Peischl[1,2], Thomas B. Ryerson[1], Ilann Bourgeois[1,2], Bruce C. Daube[10], Róisín Commane[11,12], Steven C. Wofsy[10]

[1]Chemical Sciences Laboratory, National Oceanic and Atmospheric Administration, Boulder, Colorado 80305, United States
[2]Cooperative Institute for Research in Environmental Sciences, University of Colorado, Boulder, Colorado 80309, United States
[3]University of Vienna, Faculty of Physics, Aerosol Physics and Environmental Physics, Vienna, 1090, Austria
[4]Department of Chemistry, University of Colorado, Boulder, Colorado 80309, United States
[5]School of Earth and Atmospheric Sciences, Georgia Institute of Technology, Atlanta, Georgia 30332, United States
[6]Earth Systems Research Center, Institute for the Study of Earth, Oceans, and Space, University of New Hampshire, Durham, New Hampshire 03824, United States
[7]Langley Research Center, National Aeronautics and Space Administration, Hampton, Virginia 23681, United States
[8]Ames Research Center, National Aeronautics and Space Administration, Moffett Field, California 94035, United States
[9]Bay Area Environment Research Institute, Moffett Field, California 94035, United States
[10]Department of Earth and Planetary Sciences, Harvard University, Cambridge, Massachusetts 02138, United States
[11]School of Engineering and Applied Sciences, Harvard University, Cambridge, Massachusetts 02138, United States
[12]Now at: Earth and Environmental Sciences, Lamont-Doherty Earth Observatory, Columbia University, Palisades, New York 10964, United States

*Correspondence to*: Charles A. Brock (charles.a.brock@noaa.gov)

**Abstract.** In situ measurements of aerosol microphysical, chemical, and optical properties were made during global-scale flights from 2016-2018 as part of the Atmospheric Tomography Mission (ATom). The NASA DC-8 aircraft flew from ~84 °N to ~86 °S latitude over the Pacific, Atlantic, Arctic, and Southern oceans while profiling nearly continuously between altitudes of ~160 m and ~12 km. These global circuits were made once each season. Particle size distributions measured in the aircraft cabin at dry conditions and with an underwing probe at ambient conditions were combined with bulk and single-particle composition observations and measurements of water vapor, pressure and temperature to estimate aerosol hygroscopicity and hygroscopic growth factors and calculate size distributions at ambient relative humidity. These

reconstructed, composition-resolved ambient size distributions were used to estimate intensive and extensive aerosol properties, including single scatter albedo, asymmetry parameter, extinction, absorption, Ångström exponents, and aerosol optical depth (AOD) at several wavelengths, as well as CCN concentrations at fixed supersaturations and lognormal fits to four modes. Dry extinction and absorption were compared with direct, in situ measurements, and AOD derived from the extinction profiles was compared with remotely sensed AOD measurements from the ground-based Aerosol Robotic Network (AERONET); this comparison showed no substantial bias.

The purpose of this work is to describe the methodology by which ambient aerosol properties are estimated from the in situ measurements, provide statistical descriptions of the aerosol characteristics of different remote air mass types, examine the contributions to AOD from different aerosol types in different air masses, and provide an entry point to the ATom aerosol database. The contributions of different aerosol types (dust, sea salt, biomass burning, etc.) to AOD generally align with expectations based on location of the profiles relative to continental sources of aerosols, with sea salt and aerosol water dominating the column extinction in most remote environments and dust and biomass burning (BB) particles contributing substantially to AOD, especially downwind of the African continent. Contributions of dust and BB aerosols to AOD were also significant in the free troposphere over the North Pacific.

Comparisons of lognormally fitted size distribution parameters to values in the Optical Properties of Aerosols and Clouds (OPAC) database commonly used in global models show significant differences in the mean diameters and standard deviations for accumulation-mode particles and coarse-mode dust. In contrast, comparisons of lognormal parameters derived from the ATom data with previously published ship-borne measurements in the remote marine boundary layer show general agreement.

The dataset resulting from this work can be used to improve global-scale representation of climate-relevant aerosol properties in remote air masses through comparison with output from global models and assumptions used in retrievals of aerosol properties from both ground-based and satellite remote sensing.

# 1 Introduction

Atmospheric aerosols are important components of the atmospheric system, interacting chemically and physically with gas-phase components and affecting climate processes through aerosol-radiation and aerosol-cloud interactions (IPCC, 2013). We use the term "aerosol" to indicate a population of non-cloud (non-activated) particles that are suspended in and interacting with air and its reactive gas-phase constituents. In this terminology, any given air parcel may contain multiple, externally mixed aerosol types (for example a sea-salt aerosol and a dust aerosol may coexist within the same air parcel). Global chemistry-climate models usually represent atmospheric aerosols using bulk, modal, or binned microphysical schemes that apportion various components into size classes. These representations of aerosol properties are often dynamic, allowing for chemical reactions, growth, coagulation, dilution, cloud nucleation, in-cloud production, and dry and wet deposition. To effectively simulate the role of atmospheric aerosol in climate processes, models must adequately represent the mass, composition, and phase of different aerosol types and their distribution amongst particle sizes, the spatial and temporal distribution of the components, their mixing state (existing as external mixtures with different compositions or internal mixtures with blended compositions), their optical properties (often a function of particle size), and their hygroscopic properties and suitability to serve as cloud condensation nuclei (CCN). Underlying these properties are the physical and chemical processes actually being represented in the simulations, including emissions of particles and gas-phase precursors, atmospheric transport, gas-phase, heterogeneous, and aqueous chemistry, cloud processing, evaporation, wet and dry deposition, and transformations such as condensation and coagulation. Simulating these disparate processes and properties is a challenging task for global-scale models, which must balance detailed, size-dependent representations of these mechanisms against computational efficiency. There is an imperative for improving aerosol representation in global models: the largest source of uncertainty in understanding climate sensitivity remains aerosol-radiation and aerosol-cloud interactions (the direct and indirect effects, respectively).

Global chemistry–climate models often evaluate their performance based on comparison to remote sensing observations from satellites and from ground-based sensors such as the Aerosol Robotic Network (AERONET; Holben et al., 1998). The satellite products most often used are aerosol optical depth (AOD) from sensors such as the Moderate Resolution Imaging

Spectroradiometer (MODIS) on NASA's Aqua and Terra satellites, the Multi-angle Imaging Spectroradiometer (MISR) on Terra, and the Visible Infrared Imaging Radiation Suite (VIIRS) instrument on the Suomi National Polar-orbiting Partnership satellite. Additional information on aerosol characteristics such as the angular dependence of scattered light (the phase function) and single scatter albedo $\omega_0$ (the ratio of light scattering to the sum of scattering and absorption) can be derived from multi-angle techniques such as from AERONET and MISR, while multi-angle polarimetric data can yield information on the particle size distribution and absorption coefficient (Dubovik et al., 2019). In general, algorithms to generate such additional information on aerosol properties from remote sensing measures require *a priori* assumptions about aerosol characteristics because the retrievals are under-constrained (e.g., Dubovik et al., 2000). In the case of AERONET, aerosol properties such as column-averaged aerosol phase function and $\omega_0$ can be derived with confidence only in cases where AOD exceeds 0.4 (Dubovik et al., 2000; Holben et al., 2006), much more turbid than is typical of the atmosphere away from large continental sources of pollution, dust, and biomass burning. Extrapolating intensive aerosol properties such as $\omega_0$ from measurements at high AOD values to cleaner regions may lead to substantial biases (Andrews et al., 2017). In a recent overview paper, Kahn et al. (2017) stated that "at present, it seems unlikely that particle microphysical and chemical properties can be retrieved from remote sensing measurements alone at the level of accuracy required to substantially reduce uncertainties in total direct aerosol radiative forcing (DARF), its anthropogenic component, aerosol–cloud interactions, horizontal material transports, surface–atmosphere aerosol fluxes, and air quality–related applications."

In this work we make use of in situ measurements made on a research aircraft during the Atmospheric Tomography Mission (ATom), a series of global scale, representative (Katich et al., 2018; Strode et al., 2018) tropospheric observations over the remote Pacific and Atlantic Oceans and portions of the Arctic and Antarctic Oceans, to provide detailed descriptions of the aerosols encountered. We sampled an airstream through an inlet, dried it, and used in-cabin instruments to determine the microphysical and chemical characteristics of the dried aerosol. We then calculated the ambient aerosol properties by accounting for hygroscopic growth to ambient humidity, and developed statistics for a number of dry and ambient aerosol properties for the different air mass types encountered. These data, which cover single transects over the two ocean basins in each of four seasons, do not represent a climatology of aerosol characteristics, but provide a representatively sampled

"snapshot" of particle properties that can be compared with simulations of these properties to help identify issues in model output and reveal processes that may be inadequately represented. The overarching goal of this paper is to describe how the in situ measurements are combined into a single, consistent description of the aerosol microphysical, chemical, hygroscopic, and optical properties listed in Table 1, to present a summary of aerosol properties in different air masses encountered during ATom, and to provide an entry point to the ATom dataset for use in modeling and remote sensing investigations of atmospheric composition and climate.

## 2 Methods

### 2.1 The Atmospheric Tomography Mission

The ATom mission was an airborne measurement program that investigated the composition of the remote marine troposphere over four seasons. Science flights took place from 29 July–23 August 2016, 26 January–21 February 2017, 29 September–27 October 2017, and 24 April–21 May 2018, named ATom-1 through ATom-4, respectively (Thompson et al., submitted manuscript, 2021). The NASA DC-8 aircraft, a large, four-engine, intercontinental-range commercial aircraft adapted for scientific measurements (NASA, 2015), flew from southern California southward to near the equator and back, then north to the Arctic Ocean, southward over the Pacific Ocean to New Zealand, across the Southern Ocean to Chile, northward to the Azores, across the North American Arctic to Alaska, and back to California (Fig. 1). On ATom-3 and -4, the aircraft flew southward from Chile over the Antarctic Peninsula and Weddell Sea. On ATom-1, the aircraft flew from Greenland to California without crossing the North American Arctic to Alaska. The routes northward across the South Atlantic and across eastern Canada and Greenland varied due to airport availability and weather conditions.

During these flights, the DC-8 made repeated en-route ascents and descents from the maximum flight altitude permitted by aircraft performance and air traffic control (ATC) to within ~160 m of the surface (visibility and ATC permitting), and back, similar to the HIAPER Pole-to-Pole Observations (HIPPO) study using the smaller National Science Foundation Gulfstream G-V aircraft (Wofsy, 2011). We consider a total of 625 atmospheric profiles, including both descents and ascents, in this study. The DC-8 maintained level flight for several minutes at the lowest and highest altitudes, and when

required by ATC or to save fuel; at all other times it was constantly ascending or descending at ~450 m min$^{-1}$. The flight routes were pre-planned and not adjusted except to avoid hazardous flight conditions such as deep convection. Pre-planned, multi-level flight patterns were made 12 times in the marine boundary layer (MBL) to investigate vertical fluxes over the

remote oceans.

## 2.2 Instruments

The DC-8 aircraft carried a substantial payload of in situ meteorological, gas-phase and aerosol instruments and limited radiation instruments. Measurements included reactive nitrogen compounds, volatile organic compounds (VOCs), photo-products and oxygenated species, tracers, actinic flux, meteorological parameters, and aerosol composition and size

distribution (Thompson et al., manuscript in preparation, 2021). This work focuses exclusively on the aerosol observations and also uses measurements of $O_3$, CO, pressure, temperature, water vapor, and GPS-derived aircraft location.

The aerosol size distribution instruments and their performance during ATom have been described in detail in several previous publications, which provide comprehensive documentation of the quality of the ATom aerosol dataset. Williamson et al. (2019) detail the function and performance of a multi-channel battery of condensation particle counters (NMASS:

nucleation-mode aerosol size spectrometer) used to count and size particles with diameters ($D_p$) from ~3 to ~55 nm. Kupc et al. (2018) describe the calibration and performance of an ultra-high sensitivity aerosol spectrometer (UHSAS, Droplet Measurement Technologies, Longmont, CO, USA), an optical particle counter that measures the particle size distribution from ~60 nm (0.06 μm) to 1.0 μm diameter. Brock et al. (2019) detail how these instruments are combined with a laser aerosol spectrometer (LAS, TSI Inc., St. Paul, MN, USA) to generate continuous, 1 s particle size distribution measurements

from 3 nm to 4.8 μm diameter, referred to as the aerosol microphysical properties (AMP) size distribution. Brock et al. (2019) also describe the sampling system, uncertainties, and data products associated with these dry particle size distribution measurements and show that data from these instruments are internally consistent and also agree with independently measured aerosol composition and extinction measurements within expected uncertainties.

Section 2.3 below describes in detail how dry size distributions and aerosol composition data from the in-cabin instruments are combined with data from an underwing cloud and aerosol spectrometer (CAS, Droplet Measurement Techniques, Longmont, CO, USA; Baumgardner et al., 2001; Spanu et al., 2020). The CAS is a nearly open-path laser optical particle counter that measures the size distribution of aerosol and cloud particles with diameters from 0.5–50 µm at near-ambient conditions.

Aerosol composition was determined using two mass spectrometers and black and brown carbon measurements. Froyd et al. (2019) provide a detailed description of how data from a single-particle laser ionization mass spectrometer (PALMS; particle analysis by laser mass spectroscopy) are combined with particle size distributions to determine the size-resolved composition and mixing state of particles with $D_p$ from 0.14–4.8 µm. In addition, a high-resolution time-of-flight aerosol mass spectrometer (HR-ToF-AMS, hereafter AMS for brevity, Aerodyne Inc., Billerica, USA; DeCarlo et al., 2006; Canagaratna et al., 2007; Schroder et al., 2018; Hodzic et al., 2020), which provides bulk composition of particles with geometric $D_p$ from ~0.02 to ~0.7 µm, with detection efficiencies >50% between ~0.05 and ~0.5 µm (Guo et al., 2021), collected data over ~46 s every minute, and reported with 1 s and 1 min. time resolutions (Jimenez et al., 2021). The AMS can also provide size-dependent non-refractory composition information using particle time-of-flight measurement mode, but in the free troposphere this often requires extensive time averaging which is impractical to apply during the vertical profiles.

Measurements of refractory black carbon (rBC, Petzold et al., 2013) were provided by a single particle soot photometer (SP2; Gao et al., 2007; Schwarz et al., 2010; Katich et al., 2018). This instrument uses laser-induced incandescence to measure the rBC mass within individual particles from 90 to 550 nm in diameter, in the accumulation mode size range, on a 1 s time basis (with frequent null detections at this rate at the concentrations found in ATom). The rBC mass concentration data were corrected to reflect accumulation mode rBC particles outside of the detection range of the instrument by using a lognormal distribution fitted for the average rBC size distribution for each flight, eliminating time periods near takeoff and landing, to calculate a scaling factor. That single correction factor per flight, which increased rBC mass concentrations less than a factor of 1.1 (Katich et al., 2018), was applied to the 1s data for that particular flight. The rBC data were then averaged, with zeros, to the 60 s AMS sampling times, with an uncertainty of ~30%. Information on the size distribution of

the rBC and on the thickness of non-refractory coatings on the rBC particles, which are used to calculate optical properties of the rBC, were obtained by accumulating data over longer time periods (Supplemental Materials Table S6).

Brown carbon (BrC) absorption at wavelengths from 300–700 nm was determined by off-line analysis of aerosol filter samples collected over times ranging from <5 minutes at low altitude to ~15 minutes at high altitude during ATom2–4 (Zeng et al., 2020). A total of 1,074 filters from the ATom mission, including 2–3 blanks per flight, were analyzed. Water extracts from the filter were further filtered to remove insoluble absorbing particles, then introduced into a liquid waveguide where the spectral absorption was measured with a spectrophotometer. The absorption of BrC by chromophores in the aqueous

sample was then converted to aerosol absorption as described in Sect. 2.7.2.

We also use 1 s data from a precision open-path water vapor concentration sensor (Podolske et al., 2003) with an uncertainty of ±5% and from the meteorological measurement system (Scott et al., 1990) of temperature measured within uncertainty of ±0.3 K and of pressure with an uncertainty of ±0.3 hPa, yielding an uncertainty in relative humidity with respect to water (RH) that ranges from ± ~7% (of the value) in the warm, tropical marine boundary to ± ~6% (of the value) in the cold, dry

lower stratosphere. To identify stratospheric air, we use measurements of CO and $O_3$, which were measured using a multipass optical absorption cell (McManus et al., 2005) and chemiluminescence (Ryerson et al., 1998), respectively. Aerosol measurements can be contaminated by particles resuspended from the inlet walls due to hydrometeor collisions (Murphy et al., 2004). Throughout this analysis, we use data that were obtained only in cloud-free air, based on altitude-varying thresholds for RH, T, number concentration and a measure of the particle volume size distribution in the CAS size

range (Dollner et al., manuscript in preparation, 2021). We include MBL data that are within the CAS "aerosol-cloud transition regime" category as archived in the broader ATom dataset (Wofsy et al., 2018), because excluding data from this category would remove substantial quantities of the data within the moist MBL, which often dominate column-integrated optical properties. All concentration units are reported at standard temperature and pressure (STP; 1013 hPa and 273.15 K); however, extensive optical properties such as extinction and absorption coefficients are reported at ambient temperature and

pressure conditions and, where indicated, at ambient RH.

## 2.3 Determining the composition-dependent aerosol size distribution

### 2.3.1 Overview of methodology

Calculating ambient aerosol properties relies upon combining data from multiple sizing and compositional instruments to develop a comprehensive description of the size-dependent composition and mixing state of the aerosol. From this information the hygroscopic growth and refractive index, which are essential to estimating optical properties of the hydrated aerosol, can be estimated. Figures 2 and 3 show how data from the four size distribution instruments are combined with data from the four composition instruments and compositional and optical models to determine the ambient optical properties. Because the primary purpose of determining the composition-dependent aerosol size distribution is to calculate optical properties, we begin this section by providing an overview of how these size distributions are applied for this purpose, using Figs. 2 and 3 as a guide.

The overarching approach is to assign compositions and mixing states to particles within each size bin of the measured particle number size distribution. Once this has been accomplished, refractive index and hygroscopicity for each particle type in each size bin can be estimated, and dry and ambient optical properties calculated. There is considerable detail hidden in the first steps shown in the left portion of Fig. 2–how data from different sizing and composition instruments are combined to produce the composition-resolved size distributions. Figure 3 provides clearer depiction of this process. For all particle types except rBC, the dry aerosol size distribution is determined from the in-cabin AMP instruments (NMASS+UHSAS+LAS) and the underwing CAS probe. Aerosol volume, surface area, number, and effective diameter can be readily calculated directly from the size distribution. To calculate optical and hygroscopic properties, size-dependent compositional information derived from the AMS and PALMS measurements is mapped to the number size distribution. To be clear, mass concentrations measured by the AMS and PALMS instruments are not directly used; rather, the relative composition as a function of size is applied to the measured number size distributions, which are then used to determine the mass concentration and optical and hygroscopic properties of each component. This represents a marked departure from other datasets, and is motivated by the ability of the PALMS instrument to identify the number fractional abundance of externally mixed refractory aerosol types (e.g., sea salt, dust), as well as by its inability to independently provide quantitative

information on mass concentrations. The PALMS data, which for ATom provides number fractional abundances of eight

particle types (plus an unclassified fraction) in each of four size ranges, must be mapped to independently measured size

distributions to quantify the mass concentrations of those particle types (Froyd et al., 2019).

Refractory particles identified by the PALMS instrument are assumed to be present as externally mixed aerosol components,

each of which is described by an independent size distribution. In contrast, non-refractory organic-inorganic particles

measured by the AMS and PALMS instruments are assumed to be internally mixed using the volume-weighted Zdanovskii–

Stokes–Robinson (ZSR) mixing rule (Stokes and Robinson, 1966), assuming no interaction between components, to infer

particle hygroscopicity. Light scattering at ambient RH conditions is calculated by estimating the hygroscopic growth factor

based on this measured composition, calculating the amount of aerosol water at ambient RH, using the same ZSR mixing

rule to estimate ambient refractive index, and applying Mie theory for a homogeneous sphere. As shown in Fig. 3, for

particle sizes <0.05 μm in diameter, the composition of the aerosol is largely unmeasured, but is assumed internally mixed

and represented by the bulk composition reported by the AMS instrument. From 0.05–0.14 μm diameter, the aerosol is

assumed to be internally mixed and the composition is exclusively based on the AMS measurement. This means that any

dust, sea-salt, or other refractory particles that contribute to this portion of the size distribution are substituted with the AMS

composition. (Note that the AMS can measure submicrometer sea salt (Ovadnevaite et al., 2012; Hodzic et al., 2020), but

during ATom there was little sea salt detected by the AMS in this size range and only the PALMS-detected sea salt,

primarily in the coarse mode, is considered.) From 0.14–0.25 μm diameter, the size distribution is split into the number

fractional contribution of each of eight particle types based on PALMS classification, with AMS composition applied to non-

refractory particle types. For particles with diameters from 0.25–~4 μm, the PALMS particle types alone are used, with

regional averaging as needed to improve statistics. For particles with diameters from ~4–50 μm, there are no compositional

measurements due to inlet performance and the PALMS particle types from the 1.13–~4 μm diameter range are applied.

Three light-absorbing components are assumed to be present: mineral dust, BrC, and rBC (Sect. 2.7.2). Light absorption due

to dust is directly calculated from the dust size distribution using Mie theory and an assumed refractive index with a

wavelength-dependent imaginary component. Light absorption due to BrC and rBC is treated entirely separately from these

calculations. Absorption from BrC is estimated from measurements of water-soluble absorption in aqueous filter extracts,

from which a parameterization relating BrC absorption to the abundance of rBC and biomass burning particles is derived.

Absorption from BrC is then calculated from the measured abundance of these surrogates using this parameterization.

Absorption due to rBC is calculated using core-shell Mie theory applied to airmass-averaged rBC size distributions and

coating thicknesses, from which mass absorption cross sections (MACs) are determined. These MACs, which are assumed to

be independent of RH, are then used to estimate absorption from fast-response measurements of rBC mass. Detailed

descriptions of the methods used to determine the composition-resolved size distribution and calculate the reported aerosol

parameters are given in Section 2.3.2 below.

### 2.3.2 Detailed description of methodology

The PALMS instrument measures mass spectra of ion fragments from the laser-induced thermal desorption of individual

aerosol particles (Thomson et al., 2000). Each positive mass spectrum is classified into one of several categories, or types,

using spectral signatures based on laboratory calibrations: sea salt, biomass burning, mixed sulfate/organic mixtures (which

may also contain nitrate, ammonium, and other inorganic ions), soil dust, heavy fuel oil combustion, meteoric material,

alkali salts, elemental carbon (EC), and an unclassified fraction (Froyd et al., 2019). During ATom, particles in the

"unclassified" fraction represented $8.8 \pm 8.6\%$ of all the detected particles and are treated as sulfate/organic particles in this

analysis, resulting in eight total particle types based on the mass spectral signatures. Largely because of variability in the

sampling efficiency of particles into the laser beams, by itself the PALMS instrument does not quantify absolute chemical

concentrations of the particles (Froyd et al., 2019). Instead, PALMS places particles into compositional categories such as

dust, sea salt, and mixed sulfate-organic particles, to which physical characteristics such as refractive index and

hygroscopicity are assigned. Based on laboratory calibrations, the sulfate and organic mass fractions of non-refractory

particle types (sulfate/organic mixtures, biomass burning particles composed mostly of organic material, and stratospheric

meteoric particles composed primarily of sulfuric acid with a small core of condensed meteoric material) can be estimated

from the PALMS mass spectra (Froyd et al., 2019). Because each individual particle measured by PALMS is

aerodynamically sized prior to laser ablation, each can be classified by both compositional type and size, and the number

fraction of each compositional type can be determined for a given particle size range (Froyd et al., 2019). The size-resolved PALMS composition data are converted from aerodynamic to geometric $D_p$ by applying a particle density and shape for each class. However, the PALMS cannot directly measure a composition-based size distribution because it is limited by data rate, typically ~4 s$^{-1}$, and because it has size-dependent sampling biases. Instead, a statistical description of aerosol composition in specific size classes determined from PALMS can be combined with independently measured particle size distributions to provide a size distribution for each of the particle types (Froyd et al., 2019). For this analysis, the PALMS particle types were aggregated over four size ranges (0.14–0.25 μm, 0.25–0.63 μm, 0.63–1.13 μm, and >1.13 μm); four bins provide a satisfactory tradeoff between number of bins, counting statistics per bin, and spatial resolution for the ATom mission (Froyd et al., 2019). Within each of these size ranges, the different size particles contribute unevenly to the compositional statistics depending on their abundance and the efficiency of detection (Froyd et al., 2019). Depending on ambient concentrations, time averaging may be needed to achieve statistical significance. Once adequate compositional statistics are developed as described below, the accumulated data in the four size ranges are mapped onto the independently measured particle size distributions from the AMP instruments (Fig. 3; Froyd et al., 2019; Murphy et al., 2021).

In the remote troposphere during ATom, the aerosol with $D_p \geq 0.14$ μm was composed of distinct particle types (with one of the most common types being internally mixed sulfate/organic). Thus, to calculate optical and hygroscopic properties, we do not assume a weighted internal mixture of the chemical components, but rather treat the total aerosol as an externally mixed collection of independent size distributions, each composed of one PALMS compositional type mapped onto the particle size distributions. For particles with $D_p < 0.14$ μm, for which the PALMS instrument provides limited statistics over the averaging times used here, we assume the particles are composed of a non-refractory internal mixture with composition given by the AMS instrument which provides submicron bulk composition measured over $D_p$ ~0.05–0.5 um (Guo et al., 2021; Fig. 3). Further, the AMS composition is applied to the sulfate/organic, biomass burning, EC, and meteoric particle types for the 0.14–0.25 μm PALMS size range, over which diameters the AMS samples with unity efficiency (Guo et al., 2021).

Throughout this work, we average all data to a 60 s time base determined by the AMS reporting interval. The 60 s data frequency we use translates into a vertical resolution of ~450 m given the typical ascent and descent rates in the middle and

lower troposphere, with somewhat better vertical resolution at altitudes >9 km during ascents as climb rates dropped. As noted by Hodzic et al. (2020), in background conditions during ATom a substantial fraction of the AMS organic aerosol (OA) concentrations were below detection limit, and included negative values. We substitute negative AMS values with

zeros only when calculating hygroscopic or optical properties (Sects. 2.5 and 2.7, respectively).

The PALMS data presented here were accumulated over 3 minute time periods and then interpolated to the same o 1 minute time interval as the AMS data. However, if fewer than five particles were classified by the PALMS instrument in each PALMS size range over the three-minute period, average compositional information based on much more extensive spatial averaging was applied to that size range. If the time interval in question were in the MBL, typical PALMS compositional

statistics from the MBL were applied (Fig. 3). Similarly, if the aircraft were in the lower stratosphere (as identified by CO <100 ppbv and $O_3$ >100 ppbv or >300 ppbv in the southern or northern latitudes, respectively), in a BB plume (tropospheric BB particle number fractions >0.5 and AMS OA mass >1 µg m$^{-3}$), or a dust plume (dust mass fraction >0.3 and volume concentration for $D_p$>1 µm more than 2 µm$^3$cm$^{-3}$), representative compositional statistics from these air masses were applied to the PALMS size range in question.

For PALMS data with poor statistics (fewer than five particles in a PALMS size range) in the free troposphere (FT), regionally averaged particle composition statistics were applied (Fig. 3). This situation most often applied to particles with $D_p$ >1.13 µm, which have very low number concentrations. For the four PALMS size ranges, from smallest to largest, the regionally averaged compositions were applied to 11%, 3%, 61%, and 89% of the 19,921 60 s samples, respectively. These regionally averaged compositions were separately calculated and applied depending on whether the DC-8 was over the

Pacific or Atlantic Ocean, and whether it was in Antarctic/Southern Ocean, southern midlatitude, tropical, northern midlatitude, or Arctic air masses. The latitudinal boundaries of these air mass types is provided in the Supplemental Materials (Table S1). These same air mass classifications serve as a way to organize the final data products that are the objective of this effort (Sect. 3.3).

Our treatment of the aerosol as an external mixture of discrete aerosol types as quantified by the PALMS, AMS, and SP2

instruments simplifies the actual complex mixing state of the aerosol. Particles identified as dust are assumed to have a

sulfate-organic coating, which is accounted for in the density, shape factor, refractive index, and hygroscopicity of the particles (Froyd et al., 2019). But more complex particles composed mixtures of rBC, dust, and sulfate-organic components may result from coagulation or cloud processes, and are not accounted for in this approach. Such complex mixtures of black carbon (BC, not measured with an SP2 instrument), organics, dust, and sulfate have been observed in the continental-scale

outflow from Asia (Clarke et al., 2004) and Africa (China et al., 2014). However, based on PALMS mass spectra of individual particles, the simplified treatment of the mixing state of the aerosol in ATom is justified for much of the remote ATom dataset, in which many primary particles have been removed and well-aged, secondary particles dominate (Froyd et al. 2019, Hodzic et al., 2020). We explicitly treat rBC particles with coatings using airmass-based averages of rBC core size and coating thickness as measured by the SP2 instrument (Sect. 2.7.2). Over all the ATom flights, rBC cores were present in

1.4% of the aerosol by number over the SP2 size range (90–550 nm), while in identifiable BB plumes 4.3% of these particles had rBC cores. Sulfate-organic coatings on dust are typically ~5–10% of the dust particle mass (Froyd et al., 2019). The sulfate and organic masses calculated by integrating the composition-resolved size distribution (see Supplemental Materials) were consistent within ~20% with sulfate and organic masses directly measured by the AMS instrument, with $r^2>0.84$ (Supplemental Fig. S4). This agreement indicates that substantial non-refractory sulfate and organic components were not

"hidden" on other particle types (e.g., fine-mode sea salt) and were adequately accounted for in the PALMS classification scheme used here, and supports our treatment of PALMS particle types as independent, external mixtures.

The aerosol sampling inlet used for the AMP measurements on the DC-8 aircraft, a shrouded solid diffusor inlet designed by A. Clarke (University of Hawaii) and evaluated by McNaughton et al. (2007), excludes most particles with ambient $D_p$ >5 µm at low altitude, with the 50% passing efficiency falling to ~3.2 µm at ~12 km (McNaughton et al., 2007; Brock et al.,

2019). In addition, the LAS optical particle counter, which measures the size distribution of the coarse mode using a red laser, suffers from sizing ambiguities in the size range from ~1 to ~2 µm due to Mie oscillations in the scattering cross section. The LAS also has poor coarse-mode counting statistics due to a sample flow rate of ~1 $cm^3s^{-1}$. For these reasons, we use data from the under-wing CAS probe, which has an optically defined sample flow rate of ~50 $cm^3s^{-1}$ (Spanu et al., 2020), for particles with $D_p$ >1.01 µm. The CAS suffers from similar sizing ambiguities as does the LAS. However, a data

processing scheme similar to the technique described by Walser et al. (2017), combined with a Monte Carlo method (Dollner et al., manuscript in preparation, 2022), is used to retrieve a size distribution, with uncertainties, that minimizes these biases. In this process, a range of possible ambient size distributions that are consistent with the scattering signal and the PALMS-based determination of particle types in the largest size range (1.14 to ~4.8 µm; Fig. 3) are calculated. For these calculations the refractive indices in Table 2 are used, and water uptake and the non-sphericity of dust are taken into account. Size

distributions at dry conditions are then calculated using the hygroscopic growth factors in Table 2. The median size distribution is chosen from the resulting set of possible solutions and these "dried" CAS data are combined with the AMP measurements to provide the continuous dry size distributions over $D_p$ from 3–50 µm.

Refractory BC particles are treated separately from the rest of the aerosol measured during ATom. The SP2 instrument reports the mass of rBC cores with spherical volume-equivalent diameter from 90–500 nm as a function of time. Statistics

regarding the size distribution of the rBC cores, as well as estimates of the average coating thickness on them, can be obtained with extensive averaging at the rBC concentrations found in ATom (outside of pollution layers and biomass burning plumes). The size distribution and coating thickness on rBC particles were averaged over the same air mass regions as were the PALMS data when counting statistics were insufficient (Sect. 3.3). As described in Sect. 2.7.2, the averaged, coated size distributions from the SP2 measurements are used to estimate the absorption and other optical properties.

However, the rBC size distribution is not combined with the other size distribution measurements, which are assumed to represent the purely scattering aerosol and dust. In other words, we assume two independent types of size distributions: 1) the composition-dependent size distributions, derived from the AMS, PALMS, and size distribution measurements that together describe all non-absorbing aerosol components and dust, and 2) the size distributions of coated rBC particles from the SP2 instrument that are averaged over air mass types and used to calculate MAC values as described in section 2.7.2.

(Note that coated rBC particles would also be measured by the size distribution instruments, but would be treated as other particle types–a minor error given low rBC abundance. )

Note that the PALMS instrument reports an "EC" (or "soot") compositional class, which is closely related to the rBC particles measured by the SP2 instrument. However, because the PALMS distinguishes only a very small (and uncertain)

fraction of all particles containing EC (Murphy et al., 2006), we simply assign all EC particles detected by PALMS to the

non-absorbing "sulfate/organic" class for the purpose of calculating aerosol optical and hygroscopic properties (although the

EC class is tracked separately in data files in case it might be useful in future analyses). Particles in the EC class are included

in the "sulfate/organic" component in all figures. Light-absorbing rBC particles are assumed to be adequately represented by

the more quantitative SP2 measurements alone.

**2.4 Modal fits to dry size distributions**

In global models, aerosol optical properties depend upon an accurate description of the size-resolved composition of dry

particles, which is often described by lognormal parameters that represent different aerosol modes. To compare with these

representations, lognormal fits were made to each mode (nucleation, Aitken, accumulation, and coarse) of the dry size

distributions measured during ATom. The lognormal equation used is

$$\frac{dX}{dlog_{10}D_p} = \frac{Xln(10)}{\sqrt{2\pi}ln(\sigma_g)} exp\left\{-0.5\left(\frac{ln(D_p/D_{g,x})}{ln(\sigma_g)}\right)^2\right\},$$   (1)

where the three fitted parameters are $X$, which represents number or volume, the geometric standard deviation $\sigma_g$, and the

geometric mean diameter, $D_{g,x}$. These fits were made to the volume-weighted size distribution for the coarse ($D_p$>1 μm) and

accumulation (0.08> $D_p \leq$ 1 μm) modes, and to the number distribution for the Aitken (0.012> $D_p \leq$0.08 μm) and nucleation

(0.03≥ $D_p \leq$0.012 μm) modes. The fits began with the coarse mode and proceeded toward the nucleation mode. Once fitted,

each larger mode was subtracted from the size distribution and the fit of the next smallest mode was made from the residual

size distribution. This fitting method is described in more detail in the Supplemental Materials, and comparisons of

integrated number, surface, and volume for the fitted size distributions and the raw size distributions are given in

Supplemental Tables S2-S4. All descriptions of aerosol properties are based on the measured, rather than fitted, size

distributions, unless otherwise noted.

## 2.5 Calculating ambient size distributions

To determine the growth of the dry particles to ambient diameter at the measured ambient water vapor saturation ratio (=$RH$/100), the hygroscopicity must be estimated for each of the aerosol types. The hygroscopicity of the particles is described by $\kappa$ using $\kappa$-Köhler theory (Petters and Kreidenweis, 2007). In this parameterization, the wet particle diameter $D_{drop}$ can be determined at a given water vapor saturation ratio $S(D_{drop})$ as

$$S(D_{drop}) = \frac{D_{drop}^3 - D_p^3}{D_{drop}^3 - D_p^3(1-k)} exp\left(\frac{4\sigma_{drop}M_w}{RT\rho_w D_{drop}}\right),$$

(2) where $D_p$ is

the diameter of the dry particle, $\sigma_{drop}$ is the surface tension of the droplet (0.072 J m$^{-2}$), $R$ is the universal gas constant (8.314 J mol$^{-1}$), $T$ is the ambient air temperature (K), and $\rho_w$ and $M_w$ are the density and molecular weight of water (1000 kg m$^{-3}$ and 0.018 kg mol$^{-1}$, respectively). For particles whose non-refractory composition is described by the AMS (all particles with $D_p$<0.14 µm and the sulfate/organic, biomass burning, meteoric, and EC fractions between 0.14 and 0.25 µm), an algebraic inorganic electrolyte composition model (Zaveri, 2005) was used to calculate the concentrations of ammonium sulfate, ammonium bisulfate, letovicite, sulfuric acid, ammonium nitrate, ammonium chloride, nitric acid, and hydrochloric acid from the AMS measurements of sulfate, nitrate, ammonium, and chloride. For this calculation, negative AMS values (which can occur due to background signal subtraction; Jimenez et al., 2021) were set to zero. The $\kappa$ from these electrolytic species (Table 2) was applied using the volume-weighted ZSR mixing rule to estimate the inorganic $\kappa$ for each data point. The $\kappa$ of the OA was estimated using the ratio of O/C reported by the AMS as

$$\kappa_{OA} = 0.19 \times (O/C) - 0.0048,$$

(3)

following Rickards et al. (2013). The $\kappa_{OA}$ values were smoothed with a running 10-point binomial smoothing algorithm to reduce noise. The project-wide average organic $\kappa_{OA}$ from this method was 0.18±0.03. An analysis of the relationship between $\kappa_{OA}$ and the O/C ratio (Nakao, 2017) found that volatility and solubility are also key parameters in determining $\kappa_{OA}$, but we lack the additional information on such properties needed to provide a revised estimate. The value of $\kappa_{OA}$ = 0.18 is higher than those commonly measured or assumed at continental locations. However, in the very remote airmasses that comprised the bulk of the ATom sampling, the OA was highly oxidized and chemically processed (Hodzic et al., 2020). The Zaveri/$\kappa$-Köhler approach was used successfully to simulate observed aerosol hygroscopic growth over a wide range of aerosol

compositions in the southeastern United States (Brock et al., 2016a). For the ATom data, the value of $\kappa$ was estimated as a volume-weighted sum of the $\kappa$ values of the non-refractory organic and inorganic components from the AMS measurements and the inorganic composition model, using the values listed in Table 2. The ATom project-mean value of $\kappa$ from the AMS measurements was 0.55±0.18, due to the highly oxidized OA and the abundance of acidic sulfate species present.

For particles with $D_p >0.25$ μm in the PALMS sulfate/organic, BB, meteoric, and EC compositional classes, $\kappa$ was estimated using the PALMS-measured organic mass fraction, $F_{org}$,

$$\kappa =(1-F_{org})\times0.73 + F_{org}\times0.17, \tag{4}$$

assuming particles were composed acidic sulfate components, using the project-mean inorganic $\kappa$ from the AMS, and organic material (Froyd et al., 2019). Equation (4) is a mass-weighted implementation of the ZSR mixing rule, again assuming no chemical interactions between the organic and inorganic components. Nitrate mass fraction is not quantified by PALMS for the non-refractory particle classes, but this is likely produces only a minor bias in $\kappa$ because nitrate concentrations were small (Nault et al., 2021). For example, for submicron sizes, the median AMS nitrate mass fraction was 2.4%, with 25th and 75th percentiles of 0.9% and 4.6%, respectively, when total AMS concentrations were positive.

For a pure organic aerosol ($F_{org}$=1), Eq. 4 yields $\kappa_{org}$ =0.17, close to the AMS project-wide value of $\kappa_{org}$=0.18 from Eq. 3. Using Eq. 4, the project-wide mean value of $\kappa$ for non-refractory PALMS particle types with $D_p$>0.25 μm was 0.52±0.09, similar to the AMS value of 0.54 for smaller particles. The $\kappa$ values for each aerosol type in the largest PALMS size class (1.13<$D_p \leq$ 4.8 μm) were applied to particles with $D_p >$ 4.8 μm.

Applying the values of $\kappa$ listed in Table 2, the $RH$ determined from measured static air temperature and water vapor mixing ratios, and Eq. 2, the dry size distributions for sea salt, BB, sulfate/organic, soil dust, heavy fuel oil combustion, meteoric material, and alkali salts were used to calculate ambient size distributions for each composition class. The contribution of water was calculated from the difference between the wet and dry size distributions for each composition class.

## 2.6 Calculating cloud condensation nuclei

The concentrations of cloud condensation nuclei (CCN) at several fixed supersaturations were calculated based on the observed dry size distributions and the composition determined from the AMS and the inorganic composition model (Zaveri et al., 2005). To calculate the critical wet diameter, $D_{crit}$, Eq. 2 was iteratively solved with different $D_{drop}$ using a fixed $D_p$ and a fixed $\kappa$ determined from the AMS measurements as described in Sect. 2.5 until the maximum supersaturation $S_{max}$ was found. This process was repeated for different $D_p$ until $S_{max}$ matched the supersaturation for which the CCN concentration was being calculated, giving $D_{crit,dry}$, the dry $D_p$ that yielded $D_{crit}$ for a given $\kappa$ and $S_{max}$. The number size distribution was then integrated across all $D_p \geq D_{crit,dry}$, yielding the calculated CCN concentration for that minute of flight. The AMS-derived $\kappa$ values were chosen to infer $D_{crit,dry}$ as these generally fall into the size range where composition is best constrained by the AMS. For ATom, CCN concentrations were calculated for fixed supersaturations of 0.05%, 0.1%, 0.2%, 0.5%, and 1.0%.

## 2.7 Calculating dry and ambient optical properties

### 2.7.1 Scattering

Scattering was calculated for each of the composition-based size distributions independently as

$$\sigma_{s,i}(\lambda) = \int_{3\,nm}^{50\,\mu m} \frac{\pi}{4} D_p^2 \alpha_{s,i}(D_p, n_i, \lambda) N_i(D_p) dlog_{10}(D_p),$$ (5) where $\sigma_{s,i}$ is

the scattering coefficient (m$^{-1}$) caused by composition type $i$ (Sect. 2.3), $\alpha_{s,i}$ is the scattering efficiency at wavelength $\lambda$ calculated from Mie theory using refractive index $n_i$ (Table 2), and $N_i$ is the number concentration (m$^{-3}$) of particles of composition $i$ within the logarithmic size interval $dlog_{10}(D_p)$. All particle types were treated as spherical in shape for optical calculations. Scattering was calculated for the wavelengths of 340, 380, 405, 440, 532, 550, 670, 870, 940, and 1020 nm, which match common wavelengths for the AERONET sunphotometers and satellite measurements of AOD. The refractive indices in Table 2 are not adjusted for wavelength; this is a small potential bias in the context of other assumptions and approximations in the calculation. All particle types are treated as purely scattering, spherical in shape and internally homogeneous for optical calculations, with the exception of the absorbing components rBC, BrC, and mineral dust, which are described in Sect. 2.7.2. Non-refractory particles with $D_p$<0.25 μm, and all particles with $D_p$<0.14 μm, are treated as fully

mixed, multi-component mixtures based on the AMS-derived composition and the ZSR mixing state representation introduced in Sect. 2.3.1. The dry particle refractive index is calculated as the volume-weighted mean refractive index of

455 contributing components. This calculation is further simplified for non-refractory particles with $D_p > 0.25$ µm using just the PALMS organic and sulfate mass fractions (Froyd et al., 2019), and applying organic and ammonium sulfate real refractive indices of ~1.48 to both of these components. Total scattering is the sum of the scattering from the individual composition-based size distributions ($\sigma_{s,tot}(\lambda) = \sum_i \sigma_{s,i}(\lambda)$).

To calculate scattering coefficient of the aerosol at ambient RH, the effects of hygroscopic growth were considered. The

460 diameter of every particle was adjusted based on growth factors for that aerosol type calculated as described in Sect. 2.5, and the refractive index was adjusted to the volume weighted mean of dry particle and water refractive indices. Scattering coefficients were also calculated for the particle size distributions at fixed RH values of 70, 80, and 85% at 532 nm wavelength. These values were used to fit a parametric curve describing *f(RH)*, the RH dependence of scattering, as described in Sect. 2.7.4.

**2.7.2 Absorption**

The aerosol absorption coefficient ($\sigma_a$, in m$^{-1}$) is determined for three aerosol components: refractory black carbon as measured by laser-induced incandescence by the SP2 instrument (rBC), brown carbon (BrC) extrapolated from measurements of liquid absorption in aqueous filter extracts, and absorption due to mineral dust particles identified by the PALMS instrument. The absorption for each of these components is calculated differently. Absorption due to rBC is

470 determined using core/shell Mie theory to calculate regionally representative values of absorption per unit mass (mass absorption cross-sections, or MACs) in different airmass types based on the observed size distribution of absorbing cores and the thickness of non-absorbing coatings. These MAC values are then multiplied by the observed 60s-average rBC concentrations to get $\sigma_{a,rBC}$ values. Absorption due to BrC is only roughly approximated, using the liquid absorption measured in aqueous extracts from infrequent filter samples, correcting these values for assumed non-soluble BrC and for

aerosolization, and developing a proxy relationship between $\sigma_{a,BrC}$ and measured rBC and BB particle concentrations.

Neither rBC nor BrC absorbing components are considered in the calculation of optical properties for any of the other particle types, for which we use Mie theory assuming homogeneous, uncoated spheres. For mineral dust, a refractive index with a wavelength-dependent imaginary component is applied to the measured, 60s dust size distributions, and $\sigma_{a,dust}$ is explicitly calculated using Mie theory assuming homogeneous spherical particles. Details of the calculations of $\sigma_a$ for these three absorbing components follows.

Absorption due to rBC was calculated using measurements of rBC core size and coating thickness from the SP2 instrument, averaged over the air mass type. Coating thickness could be determined only from the subset of cores with rBC mass between ~2.5 and 6 fg (~140–330 nm volume equivalent diameter), but this average coating thickness was applied to all rBC cores measured (Gao et al., 2007). The coated size distributions were used to calculate mass absorption cross-sections at the same wavelengths of 340, 380, 405, 440, 532, 550, 670, 870, 940, and 1020 nm for each air mass type via core-shell Mie theory (Bohren and Huffman, 1998), assuming that the refractive index of the rBC (Moteki et al., 2010; Table 2) remains constant across these wavelengths (Bond et al., 2013). The calculated regional-average MACs were then multiplied by the 60 s-averaged rBC mass measured within each respective region to estimate absorption due to the rBC ($\sigma_{a,BC}$) on a 60 s time base. We assume that hygroscopic growth on coated rBC particles does not appreciably change the absorption coefficient through additional lensing effects, since substantial coatings on the aged rBC particles already existed. This assumption is supported by studies that have modeled of the effects of coating thicknesses on BC cores that show a saturation effect as coating thickness increases (e.g., Zanatta et al., 2018). It is important to note that this study is not designed to evaluate the characteristics of BC refractive index and morphology (e.g., core/shell), but that these parameters are assumed. These assumptions are discussed in more detail in Sect. 4.1.3.

Absorption due to dust particles ($\sigma_{a,dust}$)was calculated simultaneously with the dust scattering calculation using the complex refractive indices at three visible wavelengths for Saharan dust provided by Weinzierl et al. (2011). Based on these measurements we use a refractive index of 1.55+0.002i at a wavelength of 530 nm, with an Ångström coefficient of 3. We assume that water uptake by dust particles does not change the imaginary component of the refractive index; i.e., the absorbing minerals are insoluble, and we assume no lensing effects due to coatings or water uptake. However, the real

component of the refractive index was allowed to vary with water uptake based on the hygroscopicity of the dust (Table 2). Since this change in real refractive index affects $\sigma_{a,dust}$, this value is slightly different for dry and ambient RH conditions.

In addition to broad-spectrum absorption by rBC and dust, certain organic species absorb light in blue and near-UV wavelengths; these compounds are referred to as brown carbon (BrC). Most of the BrC in the remote atmosphere is believed to originate from biomass burning (e.g., Washenfelder et al., 2015). Absorption due to BrC may change with time from

emission due to photo-bleaching of chromophores or to secondary production of absorbing organic species (e.g., Forrister et al., 2015; Liu et al., 2020). Secondary production is believed to take place near combustion sources, while initial bleaching takes over time scales of a day ( Forrister et al., 2015; Wang et al., 2016; Wong et al., 2019; Wu et al., 2021). However, there is evidence that high-molecular-weight chromophores may persist in aged biomass burning plumes (Di Lorenzo and Young, 2016; Wong et al., 2017). Absorption from 300–700 nm wavelength due to water-soluble (WS) BrC was measured during

deployments 2–4 of the ATom mission (Zeng et al., 2020). These measurements were made using aqueous extracts from Teflon filters collected over 5–15 minute periods. Because of these long sampling periods, it is difficult to directly combine the BrC measurements with the 60s data used in this analysis. However, we can take advantage of the observed correlations between WS BrC absorption and rBC mass and between WS BrC absorption and the PALMS biomass burning mass (Supplemental Materials in Zeng et al., 2020) to roughly estimate the WS BrC at 365 nm at 60 s frequency. This proxy WS

BrC is calculated from a multivariate linear regression between these parameters and is then multiplied by a factor of 2 to approximately account for unmeasured BrC that is not extractable in water, and another factor of 2 to convert from bulk liquid absorption to aerosol absorption (Zeng et al., 2020). The final proxy relationship is

$$\sigma_{a,BrC}(365nm) = 4(a_1 M_{BB} + a_2 M_{BC}), \qquad (6)$$

where $a_1$ and $a_2$ are parameters from the multivariate linear regressions from ATom 3–4, and $M_{BB}$ and $M_{BC}$ are the mass concentrations of

the PALMS biomass burning particles and the SP2 rBC, respectively. Only values from ATom 3 and 4 were used for Eq. 6 because most BrC measurements during ATom-2 derived from two regions of burning in Africa and South America, while during ATom-3 and -4, more dilute smoke from a range of geographic regions was sampled. The values of $a_1$ and $a_2$ were

0.07 ± 0.06 and 5.4 ± 1.1 $m^2g^{-1}$, respectively. A two-sided linear regression between this proxy BrC and the measured values yielded a slope of 0.68 ± 0.06 and $r^2$=0.40.

Given the modest ability of the proxy BrC absorption to predict the measured values, as well as the uncertainty in accounting for water-insoluble BrC and in the conversion from liquid to aerosol absorption, this $\sigma_{a,BrC}$ is probably accurate to within only a factor of ~3. The absorption coefficients due to BrC at the wavelengths used to calculate scattering and extinction were estimated using an absorption Ångström exponent value of 5 based on the measured liquid absorbance from 300-700 nm (Zeng et al., 2020).

**2.7.3 Extinction**

Absorption due to BrC, rBC and dust was summed with total scattering calculated as described in Sect. 2.7.1 to provide total aerosol extinction:

$$\sigma_e(\lambda) = \sigma_{s,tot}(\lambda) + \sigma_{a,dust}(\lambda) + \sigma_{a,BC}(\lambda) + \sigma_{a,BrC}(\lambda). \tag{7}$$

During ATom-4, the SOAP (spectrometers for optical aerosol properties) instrument measured dry aerosol extinction at a

wavelength of 532 nm using cavity ringdown spectrometry (Langridge et al., 2011). For comparison with this direct extinction measurement, dry extinction at 532 nm was calculated for a truncated size distribution to match the SOAP instrument, which operated behind a 2 μm-aerodynamic-diameter impactor. This calculated extinction agreed within experimental uncertainties with the SOAP extinction (Fig. 4a), with a slope of 0.98 and a Pearson's regression coefficient ($r^2$) of 0.86. Similarly, the absorption calculated from the SP2 measurements at 532 nm as described in Sect. 2.7.2 agreed well

with the SOAP photoacoustic absorption spectrometer (Lack et al., 2012) during ATom-4 when the absorption signal was greater than the SOAP noise level of ~2×$10^{-6}$ $m^{-1}$ (Fig. 4b), with a slope of 0.88 and $r^2$=0.71. These comparisons between the calculations of extinction based on aerosol composition, size distribution, refractive index, and rBC mass and coating thickness, and independent, direct measurements of extinction and absorption provide confidence that the calculated optical properties represent the bulk submicron aerosol properties in the atmosphere with good fidelity.

## 2.7.4 Intensive optical properties

Intensive aerosol properties are those that do not vary with abundance. All intensive optical properties were calculated at wavelengths of 340, 380, 405, 440, 532, 550, 670, 870, 940, and 1020 nm. Single scatter albedo $\omega_0$ is the ratio of scattering to total extinction ($\sigma_{s,tot}/\sigma_e$). The value of $\omega_0$ was calculated for both the total dry size distributions as well as those at ambient RH. As described in Sect. 2.7.2, the absorbing component is calculated from regionally averaged MAC values multiplied by the 60 s rBC mass concentrations and from the proxy $\sigma_{a,BrC}(\lambda)$. We do not attempt to model absorption by adjusting the imaginary refractive index of the different components of the composition-resolved size distributions because this would be a severely underconstrained problem.

Mass extinction efficiency is the ratio of extinction to aerosol mass concentration. This parameter is calculated from the dry size distributions using the total dry extinction coefficient $\sigma_e$ and the total aerosol mass, which is the sum of the aerosol density for each composition component (Table 2) multiplied by the particle volume from the integrated size distribution for that component.

Phase function $P(\theta,\lambda)$ is the normalized angular distribution of light intensity scattered by an aerosol in angle $\theta$ relative to the incident radiation. For spherical (Mie) scatterers, it is defined as

$$P_i(\theta,\lambda) = \frac{I_i(\theta,\lambda)}{\int_0^\pi I_i(\theta',\lambda)\sin\theta'\,d\theta'} ,$$ (8),

where I is the intensity of the scattered light from an aerosol of composition class $i$. The asymmetry parameter $g$ is a simplified description of the phase function that is often used in radiative transfer approximations such as the Henyey–Greenstein phase function or the delta-Eddington approach, which are then applied within global-scale models. The asymmetry parameter for an aerosol of composition $i$ is defined as

$$g_i(\lambda) = \frac{1}{2}\int_0^\pi \cos\theta P_i(\theta,\lambda)\sin\theta\,d\theta.$$ (9)

As described by Moosmüller and Ogren (2017), practical values of $g_i$ in the atmosphere range from 0 (symmetrically scattered light) to +1 (purely forward-scattered light). Typical values for accumulation-mode dominated size distributions for mid-visible wavelengths are ~0.4–0.6, with larger values possible for size distributions with a substantial coarse fraction

(e.g., Andrews et al., 2006; Fiebig and Ogren, 2006). We calculate the total aerosol $g$ for both dry and ambient RH conditions from the scattering-weighted sum of the $g_i$ from each composition-based size distribution. The small contribution

of rBC and BrC to $P(\theta)$, $g$, and scattering coefficient is ignored.

The fine mode fraction $(\eta)$ is the fraction of the total extinction that is attributable to the fine mode (e.g., Anderson et al., 2005). This is a parameter that can be retrieved from remote sensing measurements and that apportions the light extinction between the fine (accumulation) mode, whose particles are mostly produced from combustion and secondary processes, and the coarse mode, whose particles are mostly generated by mechanical processes. Because some of the coarse-mode particles

extend into the submicron size range (and vice versa), we use the modal fits to the composition-based size distributions to calculate $\eta$. The refractive index and hygroscopicity of the coarse and fine modes used to calculate $\eta$ is calculated from the volume-weighted mean contribution of each composition class within one geometric standard deviation of the volume modal diameter of that mode.

The ratio of scattering at wavelength $\lambda$ at a given RH to that at dry conditions, or $f(RH)_\lambda$, can be parameterized

simply using a physically based function,

$$f(RH)_\lambda \equiv \frac{\sigma_{s,tot}(\lambda,RH)}{\sigma_{s,tot}(\lambda,RH=dry)} \simeq 1 + \kappa_{ext} \frac{RH}{100-RH}, \tag{10}$$

where $\kappa_{ext}$ is a fitted parameter that is related to, but not identical to, the $\kappa$ in $\kappa$-Köhler theory (Brock et al., 2016a). Because the dry size distributions are assumed to be measured at RH=0%, no correction to Eq. 10 to account for residual water (Titos et al., 2016; Kuang et al., 2017; Burgos et al., 2020) is applied. The value of $\kappa_{ext}$ was calculated for each 60 s data interval by

least-squares fitting of Eq. 10 to the scattering values calculated at the dry condition and at RH values of 70, 80, and 85%, for each of the 10 wavelengths considered. Separately, the value of $f(RH)$ was calculated for RH=85% for comparison with literature values (e.g., Burgos et al., 2020).

The Ångström exponent describes the power law relationship between extinction, scattering, or absorption and the wavelength of incident light:

$$\frac{\sigma_{x,\lambda}}{\sigma_{x,\lambda_0}} = \left(\frac{\lambda}{\lambda_0}\right)^{-\gamma_x},$$ (11)


where $x$ represents extinction ($e$), scattering ($s$), or absorption ($a$) and $\lambda$ is the wavelength of incident light, $\gamma$ is the Ångström exponent, and the naught subscript indicates a reference wavelength. Values of $\gamma_e$ and $\gamma_s$ are determined by making a least-squares fit to the calculated values of $\sigma_e$ and $\sigma_s$, respectively, over two wavelength ranges. The first of these, termed the UV-vis Ångström exponent, is determined by fitting to the values at 340, 380, 405, 440, 532, and 550 nm wavelength, while the

vis-IR Ångström exponent is calculated at the wavelengths of 670, 870, 940, and 1020 nm. The value of $\gamma_a$ for $\sigma_{a,BrC}$ is assumed to be 5 at all wavelengths (Zeng et al., 2020). For $\sigma_{a,rBC}$ , $\gamma_a$ is calculated from regionally averaged rBC size distributions using core/shell Mie theory (Sect. 2.7.2) for the UV-vis and vis-IR wavelength ranges. Because the raw scattering, extinction, and absorption coefficients at all 10 wavelengths are provided in the archived dataset, additional Ångström exponents using specific wavelength pairs can be calculated readily.

**2.8 Calculating aerosol optical depth**

During ATom the DC-8 executed repeated en-route ascents and descents between ~0.16 and ~12 km approximately every 30–60 minutes. By integrating ambient extinction or absorption vertically during each ascent or descent, extinction AOD and absorption AOD (AAOD) can be calculated. Because ambient extinction is calculated for each composition class, it is possible to determine the portion of AOD attributable to each of these classes, along with the associated water. This provides

a valuable dataset with which to apportion AOD amongst different aerosol types, and can be used to compare with model representations of AOD and with assumptions regarding aerosol types used in remote sensing retrieval algorithms.

To adequately represent atmospheric AOD and AAOD, each integrated profile should contain representative measurements in the MBL, where sea-salt aerosol often dominates total AOD. The profiles should also contain any optically significant layers, such as biomass burning and dust plumes, that may be present. To ensure that the profiles represent atmospheric

AOD, the following rules were used: 1) data were integrated over 1 km thick layers, 2) the profile must have extended from the bottom 1 km layer to at least 8 km in altitude, 3) the bottom two layers (0–2 km) both must have contained valid

extinction data, 4) no more than two layers above the required bottom two layers could have been discarded due to cloud screening, and 5) data were interpolated across up to two such discarded layers. There are typically one or two 60 s average data points within each layer for each profile. Of the total 625 oceanic profiles made during ATom, 463 met the criteria listed above. The number of profiles in different latitude regions over the Atlantic and Pacific Oceans are provided in Supplemental Table S5.

Atmospheric AOD was calculated as

$$AOD_\lambda = \sum_{j=0}^{N} \overline{\sigma_{e,\lambda,j}} \Delta z, \tag{12}$$ where $j$

represents each 1 km thick layer $\Delta z$ beginning at altitude $z=0$ km, and $\overline{\sigma_{e,\lambda,j}}$ is the ambient extinction coefficient for wavelength $\lambda$ averaged from the 60 s data within the layer. Absorption AOD (AAOD) is obtained by substituting $\sigma_a$ for $\sigma_e$.

## 3. Results

### 3.1 Aerosol extinction

Aerosol extinction was calculated for both the dry and ambient RH conditions, at STP as well as at ambient pressure and temperature. The difference between the ambient RH and dry extinction values provides the extinction due to H$_2$O. The spatial pattern of ambient total extinction and that due to the aerosol types that dominate AOD–biomass burning, sulfate/organic mixtures, sea salt, dust, and H$_2$O–are shown in Fig. 5. This figure shows the comprehensive coverage in altitude and latitude of the ATom flights and provides an overview of the spatial patterns of the contribution of different aerosol species to AOD. Total ambient extinction in the remote marine atmosphere (Fig. 5a, b) is dominated by sea salt (Fig. 5g, h) and associated water (Fig. 5k, l) in the MBL, with several notable exceptions. Biomass burning aerosol over the northern subtropical Atlantic, and to a lesser extent over the southern subtropical Atlantic and the tropical and northern midlatitude Pacific, at altitudes <4 km, is an important contributor to dry extinction (Fig. 5c, d; Schill et al., 2020). In general, the Northern Hemisphere has more biomass burning extinction than the southern. Contributions to extinction from sulfate/organic particles of mostly secondary origin (Fig. 5 e, f; Hodzic et al., 2020) are substantially higher in the Northern

than the Southern Hemisphere, especially over the Pacific, due to higher biogenic and anthropogenic emissions in the more continental Northern Hemisphere.

Extinction due to dust is important in the tropics and subtropics of the Atlantic Ocean due mostly to emissions from the Sahara Desert (Fig. 5i, j). There are also significant contributions to extinction from dust in the mid-latitudes of the Atlantic and in the free troposphere (FT) of the northern Pacific due to dust emitted from Asia and the Sahara (Froyd et al., submitted manuscript, 2021). There is very little extinction from dust in the Southern Hemisphere at altitudes >2 km, in sharp contrast with the Northern Hemisphere.

### 3.2 AOD and comparison with AERONET

The ambient extinction measured during each profile (Fig. 5) was vertically integrated as described in Sect. 2.8 to calculate AOD. Several of these profiles were relatively near AERONET sites. AERONET is an affiliation of ground-based remote sensing sites that use consistent methodologies, calibrations, and instrumentation to make sunphotometer measurements of AOD and, in cases of high atmospheric turbidity, aerosol optical and microphysical properties (Holben et al., 1998; 2006). These measurements provide an opportunity to compare AOD calculated through the complex process outlined in Figs. 2 and 3 with directly measured values. Individual profiles were selected for comparison with AERONET if 1) the location of the aircraft at the midpoint in time between the start and end of the profile was within 300 km of the AERONET site, and 2) if the midpoint time of the profile was within ±4 h of an AERONET data record. An exception was made for the Macquarie Island site, as it was the only AERONET site with data in the Southern Ocean. Macquarie Island was 421–601 km from the midpoint of the nearest three DC-8 profiles. There were no matches meeting criteria (1) and (2) between the ATom profiles and the ship-borne Maritime AERONET Network (Smirnov et al., 2009). For comparisons of AOD with the AERONET site at the Mauna Loa Observatory, which lies at 3.4 km altitude, the DC-8 profile was integrated upward beginning with the 3–4 km altitude bin. Version 3 Level 2.0 AERONET data were used for all comparisons, and the AOD at 532 nm was interpolated from observations at 500 and 675 nm using the Ångström equation (Eq. 11).

The stratospheric aerosol layer contributes ~0.005 to ~0.01 to mid-visible AOD measured by AERONET (e.g. Yang et al., 2017) , but not to that derived from the DC-8 profiles. The contribution of stratospheric AOD was determined using the Global Space-based Stratospheric Aerosol Climatology (GLOSSAC) v. 2.0 (NASA/LARC/SD/ASDC, 2018; Kovilakam et al., 2020). The mean values of stratospheric AOD at 532 and 1020 nm sampled along the aircraft flight track from the
starting to ending location of each profile were spatially interpolated from this dataset and estimated for other wavelengths using Eq. 11. These values were added to the AODs calculated from each ATom profile, and are significant contributors to AOD for the profiles with the lowest aerosol burdens.

A two-sided linear regression between the calculated and measured AOD, accounting for estimated uncertainties, produces a slope of 0.86 with $r^2$=0.76 (Fig. 6a). A logarithmic plot of the same data shows that values of AOD calculated from the
ATom aircraft data are generally lower than those from the AERONET sites, especially for AOD values <0.05 (Fig. 6b). The normalized mean bias for all of the data points is -0.07, suggesting a slight underestimate by the aircraft compared with the sunphotometers. Overall, 22 of the 32 comparison points are within a factor of two (Fig. 6b). We note that the average distance between the AERONET sites, excluding Macquarie Island, and the midpoint of the DC-8 profiles was 161 km. Further, the DC-8 performed slantwise profiles spanning ~25 minutes and ~300 km horizontally, while the AERONET sites
made direct-solar measurements. Past analysis has shown that comparisons between aircraft-derived AOD and those from sunphotometer sites must be made with great care, accounting for horizontal variability in aerosol characteristics and loading, even over the remote Pacific Ocean (e.g., Shinozuka et al., 2004). The comparisons between AODs derived from the ATom slantwise profiles and the nearest available AERONET sites should be considered as simple "sanity checks", rather than as robust, quantitative evaluations. More detailed analyses comparing ATom-derived AOD and values from high-
resolution satellite data and those calculated using global models are underway.

Figure 7 shows the calculated AOD for each profile with valid extinction data meeting the criteria in Sect. 2.8, amounting to 463 of the total 625 profiles made over the oceans. While there is great variability in AOD from these individual profiles, general patterns are evident. First, the Northern Hemisphere midlatitudes and polar regions have substantially higher AOD than the same latitudes in the Southern Hemisphere, often by a factor of two or more. This difference reflects the much

higher continental emissions of aerosols and precursors in the Northern Hemisphere. Second, the tropical and subtropical

Atlantic has the highest AOD values found during the ATom flights due to Saharan dust and strong emissions from African

biomass burning. Finally, low values of AOD, of order 0.02, are frequently found over the Southern Ocean and near the

Antarctic Peninsula. In the absence of high winds to produce abundant sea-salt aerosol (Shinozuka et al., 2004), these

regions of the troposphere generally have the least influence from anthropogenic and continental sources, and thus the least

aerosol extinction (although elevated concentrations of BB burning aerosol were detected in the UT of the Southern Ocean

during ATom-2; Fig. 5a, c). The contributions of different aerosol types to extinction profiles in different regions of the

atmosphere are examined in more detail in Sect. 3.3.1.

## 3.3 Aerosol characteristics in different air masses

To summarize and present the data, aerosol characteristics were averaged over the same spatial regions over which PALMS

free tropospheric compositions were averaged. These regions are schematically represented in Fig. 8, and include the tropics,

the midlatitude and polar regions for the northern and southern hemisphere and for the Pacific and Atlantic ocean basins, and

the northern and southern high latitude stratosphere. The precise latitudinal definitions of these regions were

based on analysis of the air mass characteristics encountered, and varied with each ATom deployment as indicated in Table

S1 in the Supplemental Materials. The top of the MBL in each profile was identified by manually inspecting the data for a

sharp gradient in temperature, dew-point temperature, wind speed and direction, and gas-phase tracers such as $O_3$, $NO_2$, CO

and $H_2O$, and in particle number, with relatively homogeneous mixing ratios below this altitude. The top of the MBL was

often quite ambiguous, particularly over colder waters where thorough atmospheric mixing may not take place. Different

definitions of the MBL height are unlikely to substantively change most conclusions given the relatively coarse temporal

resolution of the averaged data (~60 s) and the associated vertical resolution (~450 m). However, if aerosols with MBL

characteristics (e.g., high concentrations of sea-salt particles) are present above the identified top of MBL, they may skew

average compositions for the FT.

The stratosphere was defined as $O_3$ >100 ppbv and CO <100 ppbv in the Southern Hemisphere and $O_3$ >300 ppbv and CO <100 ppbv in the Northern Hemisphere. These definitions were chosen based on the occurrence of a mode of nearly pure sulfuric acid particles and particles with a meteoric core and sulfuric acid coating, indicating that the aircraft was sampling

predominantly stratospheric particles (Murphy et al., 2021). The maximum GPS-derived altitude reached by the DC-8 was 13.2 km, and much of the stratospheric air was sampled when the tropopause heights were low in the winter hemisphere or in tropopause folds. The maximum $O_3$ observed, 957 ppbv, was measured at an altitude of 11.3 km at 68º N latitude when CO was 22.2 ppbv.

Regardless of altitude or region, samples were classified as being in a biomass burning plume when the number fraction of

particles classified by PALMS as "biomass burning" by their potassium- and carbon-rich ion signatures (Hudson et al., 2004; Schill et al., 2020) was >0.5 and AMS-measured OA mass concentrations were >1 µg m$^{-3}$. Similarly, dust cases were identified when the number fraction of PALMS "mineral dust" particles was >0.3 and coarse mode volume was >2 µm$^3$cm$^{-3}$.

### 3.3.1 Extinction profiles

The contribution of different aerosol components to extinction varies significantly with altitude and air mass type. In Fig. 9

we present vertical profiles, averaged in 1 km bins, of the average contribution to extinction for the different aerosol types, for all of the ATom deployments. The fractional contributions of each aerosol type to extinction are shown in Supplemental Fig. S5.These profiles include all non-cloudy data within the geographic region, including data taken in the MBL, in dust and BB plumes, and in the stratosphere. Sea salt in the MBL and associated water dominates the extinction in most of the regions. However, there are notable exceptions. Over the Arctic (Fig. 9a), there are significant contributions from biomass

burning and sulfate/organic particles, and associated water, declining with increasing altitude. Two of the ATom deployments took place in winter and spring, when northern hemisphere pollution substantially affects the lower Arctic troposphere. The vertical profiles of extinction are consistent with the phenomenon of chronic, background "Arctic haze" (Brock et al., 2011). In sharp contrast, the Antarctic/Southern Ocean profiles (Fig. 9b) shows the dominance of sea salt and water, with minor contributions to extinction from biomass burning layers encountered in the upper troposphere.

In the Pacific northern midlatitudes (Fig. 9c), biomass burning and sulfate/organic particles also contribute significantly to extinction, and dominate above the MBL. These aerosol types are associated with plumes of pollution and biomass burning from Asia. Dust contributes as well, but to a lesser extent. Fewer such layers were encountered over the Atlantic at northern midlatitudes (Fig. 9d). Over the tropical and subtropical Atlantic (Fig. 9f), there is a significant contribution from Saharan dust in the lower troposphere along with biomass burning, sulfate/organic particles, sea salt, and absorption from rBC. In contrast, the Pacific tropical lower troposphere (Fig. 9e) shows the dominance of sea salt and lesser contributions from other components, similar to the Pacific and Atlantic southern midlatitudes (Figs. 9g, h).

Extinction in air classified as being in the MBL (Fig. 10a), in dust plumes (Fig. 10b) or in biomass burning plumes (Fig. 10c) may also be attributed to specific aerosol components using the ATom dataset. Unsurprisingly, sea salt and associated water dominate extinction in the MBL, followed by sulfate/organic mixtures, biomass burning aerosol, and dust. In dust plumes, mineral dust particles dominate, followed by water, sulfate/organic particles, and BB particles. In biomass burning plumes, particles containing biomass burning material dominate extinction, while sulfate/organic particles and water also contribute substantially to extinction. Absorption from rBC, which includes enhancement due to substantial non-absorbing coatings shown to be present by the SP2 measurements, is also a significant contributor to the extinction budget of these plumes.

### 3.3.2 Size-dependent composition

The PALMS single-particle mass spectrometer measures the composition and size of individual particles, which can then be mapped to high-resolution particle size distributions to provide a representation of the composition-based size distribution. Since many global models carry only the mass of different aerosol species and then prescribe their size distribution with modal or sectional representations (e.g., Chin et al., 2002; Liu et al., 2012; Mann et al., 2010), the high-resolution observations from ATom provide an important point of comparison. Aerosol-radiation and aerosol-cloud interactions flow directly from the size of the particles and their optical and hygroscopic properties; thus it is essential that models predict the right aerosol properties for the right reasons. In this section we present the average composition-dependent size distribution of the aerosol in the different air mass types, which is useful for evaluating how different compositions influence optical properties.

Two distinct volume (mass) modes are present in all air mass types; an accumulation mode between 0.08 and ~1 μm and a coarse mode at larger sizes (Fig. 11). Small peaks between ~0.6 and 2 μm (e.g., Fig. 11l) are likely due to ambiguous instrument response at particle sizes near the wavelength of the lasers, and to overlaps between the underwing CAS instrument and the in-cabin LAS instrument. Most of the other fine structure in the shape of these modes is due to averaging together different size distributions. These average size distributions do not properly represent the aerosol's modal characteristics. For example, averaging size distributions with two peaks might produce a mean distribution with an excessively broad, flat mode that does not accurately describe the characteristics of either–or any–atmospheric size distribution. However, these average size distributions usefully describe the contributions of different particle types to the different modes. In Section 3.3.3, we use modal representations of the measured size distributions to more accurately describe the shape of the aerosol size distributions and their statistics in different air mass types.

The composition-based size distributions with regional labels (i.e., the left two columns) are from the 3-4 km layer of the FT only and exclude data from strong BB and dust plumes and stratospheric intrusions. Size distributions from the MBL, stratosphere, and strong BB and dust plumes (right column) are not regionally separated; e.g., Fig. 11c is an average of all MBL size distributions in all regions.

Water is an important component in all of the regionally averaged size distributions (left two columns). In the Pacific and Atlantic northern midlatitudes (Figs. 11d, e) and the Atlantic tropics (Fig. 11h), the dry volume (ignoring water) associated with the coarse mode is substantially larger than that of the accumulation mode, primarily due to the contribution of mineral dust to the coarse mode. In the Arctic (Fig. 11a), dust is a major fraction of the dry coarse mode, but the accumulation mode is larger due mostly to the sulfate/organic and BB particles characteristic of Arctic haze (e.g., Brock et al., 2011). These regional-scale contributions of dust to the coarse mode are largely a result of averaging discrete layers or plumes of dust over the region, rather than the ubiquitous presence of dust throughout the FT (Froyd et al., submitted manuscript, 2021). In the southern midlatitude Pacific and the Antarctic/Southern oceans, which are more remote from continental sources (Fig. 11b, j), sea salt dominates the coarse mode of the FT when averaged over the region, while sulfate/organic particles contribute most to the accumulation mode. Biomass burning particles are substantial portions of the dry accumulation mode in all

regions except the Antarctic/Southern Ocean (Fig. 11b), and to a lesser extent over the tropical and South Pacific (Fig. 11g, j) and the South Atlantic (Fig. 11k). The biomass burning particles are found mostly in the upper end of the accumulation-mode volume, consistent with the larger diameters typically found near wildfire sources (Radke et al., 1977; Moore et al., 2021) compared to secondary particles from natural and anthropogenic sulfur and organic sources.

Size distributions measured in the southern hemisphere stratosphere (Fig. 11l) are unique from the tropospheric size distributions, with an accumulation-mode composition dominated by nearly pure sulfuric acid, meteoric materials mixed with sulfuric acid, and mixed sulfate/organic particles and sea salt from FT air mixed with the stratospheric air. During ATom, particles from three specific events–a volcanic eruption, a pyro-cumulus injection, and lofting of dust–strongly influenced the stratospheric aerosol during ATom; these cases are discussed in Murphy et al. (2021).

### 3.3.3 Modal parameters

Many global models use modal representations of the particle size distribution because sectional models are computationally expensive. As described briefly in Sect. 2.4, and in more detail in the Supplemental Materials, the measured size distributions were fitted using four lognormal functions, representing the nucleation, Aitken, accumulation, and coarse modes. After fitting, the integrated number, surface, and volume were compared with those from the raw size distributions. The number, surface, and volume from the four modes of the fitted and measured distributions were similar and highly correlated, with regression slopes between 0.94 and 1.08 and $r^2$ values >0.76 (Supplemental Material Tables S2–S4). The four-mode lognormal fits efficiently describe the measured size distributions, and provide measurement-based lognormal parameters for comparison with prescribed values used in many global models. Further, the modal fits provide a physically rational way to average size distributions together, since the average geometric mean diameter ($D_g$) and standard deviation ($\sigma_g$) for an air mass can be calculated directly. If one were to instead average all of the size distributions in an air mass together and then fit lognormal parameters, $\sigma_g$ would be too large because the average size distribution is broader than the individual size distributions contributing to that average. AOD and direct radiative forcing are sensitive to the value of $\sigma_g$ (Brock et al., 2016b).

As an example of the fitted lognormal parameters, vertical profiles for the tropics of the Pacific and Atlantic (Fig. 12) show several interesting features. It is important to note that there is considerable vertical and horizontal variability in aerosol properties in any given single profile, due to the effects of quasi-horizontal transport from continental sources in thin layers, near-surface wind speed, outflow from deep convection, removal in clouds, and other processes (e.g., Clarke et al., 2002;

Shinozuka et al., 2004). The average profiles presented in Fig. 12 are intended to highlight systematic features in the vertical distribution of aerosol properties that are robust when averaged across many profiles (Table S5) over four seasons. Low nucleation mode concentration ($<30$ cm$^{-3}$) were present at altitudes $<5$-6 km (Fig. 12a, e), and lognormal fits could not be made (although raw nucleation mode concentration data are still shown). Nucleation and Aitken mode concentrations decreased from values $>10^4$ cm$^{-3}$ and $>10^3$ cm$^{-3}$, respectively, at the top of the profile to values $\sim10$ cm$^{-3}$ and $\sim200$ cm$^{-3}$ at 2

km as a result of new particle formation in the UT and coagulational loss during slow descent (Fig. 12a; Clarke et al., 2002; Williamson et al., 2019). Growth due to condensation during this descent is evident in the slightly increasing modal diameter of the Aitken and accumulation modes with decreasing altitude (Fig. 12b), although this growth is somewhat obscured by the shift in growing particles from the nucleation mode to the Aitken mode. Of course, other processes such as cloud processing, wet scavenging, and loss of OA by chemical processing can also affect the variation in modal diameter with altitude.

However, we note that the $\sigma_g$ values of the accumulation and Aitken modes tend to decrease toward the surface in the troposphere (Figs. 12c, f), which is consistent with condensational growth, which leads to a narrowing of the size distribution (McMurry and Wilson, 1982). The new particle formation in the tropical UT is tightly coupled to the very low concentrations of accumulation-mode particles (Fig. 12a) due to scavenging during deep convection (Clarke et al., 2002; Williamson et al., 2019). Nucleation-mode concentrations are lower in the UT over the Atlantic (Fig. 12d) than over the

Pacific (Fig. 12a), although the same general trend of declining concentration towards the surface remains.
The number concentration of coarse-mode particles declines rapidly with increasing altitude above the MBL, while accumulation-mode concentrations do not fall consistently with increasing altitude (Fig. 12a,d). Coarse-mode particle concentrations in the lower troposphere are consistently higher over the Atlantic than over the Pacific due to smoke and Saharan dust. The $\sigma_g$ of the lognormal distribution is $>2$ in the lowest 2 km of the profile, where sea salt dominates, but $<2$ in

the middle and upper troposphere, where dust dominates the coarse mode. In general, the value of $\sigma_g$ ranges from ~1.5 to ~2

for all modes throughout the profiles except for the coarse sea-salt mode at altitudes <2 km.

Similar plots for the other regions measured during ATom are presented in the Supplemental Materials (Figs. S1–S2). The

modal parameters from the ATom data are compared with two previously published datasets in Sect. 4.1.1.

### 3.3.4 Single scatter albedo and absorption

Single scatter albedo $\omega_0$ is the ratio of light scattering to the sum of scattering and absorption. This parameter is key in

determining the direct radiative effect of aerosol (McComiskey et al., 2008). In most of the air masses encountered in ATom,

values of $\omega_0$ at both dry and ambient RH conditions tend to decrease from values >0.96 near the surface to a broad minimum

in the lower or middle FT, before increasing again in the UT (Fig. 13). These profiles result because extinction falls more

rapidly with increasing altitude from the boundary layer to the FT than does absorption due to rBC and BrC (Figs. 9, 13).

This decrease in $\omega_0$ in most of the profiles (Fig. 13) may be associated with the general shift of accumulation-mode particles

to smaller particle sizes with increasing altitude (Fig. 12), which would reduce their aerosol mass scattering efficiency, while

the mass absorption efficiency of absorbing rBC particles does not change much with increasing altitude. In other words,

shifts in aerosol size can change $\omega_0$ even if the relative mass of scattering and absorbing components is not changing

substantially.

### 3.3.5 Cloud condensation nuclei

The concentrations of CCN at STP conditions, determined from the size distributions and calculated hygroscopicity at five

values of supersaturation (Sect. 2.6), show substantial variations across the different regions sampled during ATom (Fig. 14).

In the midlatitudes of the Northern Hemisphere (Figs. 14c, d), concentrations are substantially higher in the middle and

lower FT than at similar latitudes in the Southern Hemisphere (Figs. 14g, h). For example, at supersaturations of 0.2%,

concentrations in the Southern Hemisphere FT are ~10–50 cm$^{-3}$ throughout the profile, while in the Northern Hemisphere the

concentrations fall with increasing altitude from >100 cm$^{-3}$ in the MBL to ~50 cm$^{-3}$ in the middle troposphere. In the tropics,

concentrations fall steadily from >200 cm$^{-3}$ near the surface to ~10 cm$^{-3}$ at 10 km altitude. The spread in CCN concentrations

for the different supersaturations increases with altitude in the tropics and northern midlatitudes due to the shift in modal diameter to smaller sizes (Fig. 12). In the Arctic, Antarctic/Southern Ocean, and southern midlatitude profiles the CCN

concentrations do not spread with increasing altitude as much because the aerosol size distributions in these regions do not shift to smaller sizes with increasing altitude (Supplemental Materials Figs. S1, S2).

## 4. Discussion

### 4.1 Comparisons with previously published work

It is far beyond the scope of this work to provide a comprehensive comparison of the ATom observations with the extensive

literature on global aerosol microphysical properties, which are derived from a panoply of in situ and remote sensing measurements and model simulations. However, it is useful to briefly compare the airborne data for a few parameters that are of special interest regarding the direct radiative effect. Here we compare modal fits to the ATom size distributions with model assumptions and a marine dataset, discuss the ATom observations in the context of an existing comprehensive airborne dataset, and evaluate the MAC values we calculate for the coated rBC relative to some recently published analyses.

#### 4.1.1 Lognormal size distribution parameters

Two frequently used datasets, the OPAC database (Hess et al., 1998; Koepke et al., 2015), which is commonly used by global models, and the ship-borne dataset reported by Quinn et al. (2017), provide useful comparisons. The measurements of Quinn et al. (2017; hereafter Q17) were made from 1993–2015 during multiple research cruises over the Arctic, Pacific, Southern, and Atlantic Oceans using a suite of instruments to obtain the particle size distribution from 0.02–10 µm diameter

at dry conditions. These observations are thus directly comparable to the dry Aitken, accumulation, and coarse-mode size distributions measured in the MBL during ATom. In addition, we can compare our observations with the modal aerosol model (MAM; Liu et al., 2012, 2020), which places various aerosol types into prescribed lognormal modes, usually using four or seven such modes.

Global models that use a modal description of aerosol size distributions often use the OPAC database to prescribe lognormal

parameters. The OPAC database provides lognormal parameters for several particle types, including "insoluble", "water-soluble", "soot", and mineral particles in three different size classes: Aitken (referred to as "nucleation" in OPAC), accumulation, and coarse modes, and sea salt in the latter two modes only. The OPAC database is meant to represent "average" atmospheric conditions, presumably including polluted air masses, while the ATom dataset focuses on remote marine air with aged aerosol from a mix of continental and marine sources.

The most direct comparisons between the ATom dataset and the OPAC database is between the "water-soluble", "sea-salt", and "mineral" OPAC components and the sulfate/organic, sea salt, and dust aerosols measured during ATom. The sulfate/organic particles are best described by modal fits to the Aitken and accumulation modes, while sea-salt and dust particles are best described by the coarse mode fits. The comparisons (Fig. 15; Table S7) show that, in general, $\sigma_g$ is wider in the OPAC database than in the ATom observations, except for coarse-mode sea salt (in which case OPAC is lower than the

observations) and accumulation mode dust (in which case they are comparable). In contrast to OPAC, several versions of the modal aerosol model (MAM), used in various earth system models (e.g., Liu et al., 2012, 2020), incorporate $\sigma_g$ values that range from 1.6 to 2.0 for the Aitken, accumulation, and coarse modes, which are much more aligned with the ATom and Q17 measurements, except for coarse mode sea salt. The larger $\sigma_g$ in the OPAC database for all aerosol types except sea salt (Fig. 15b) would tend to increase the amount of extinction and scattering per unit aerosol mass (Brock et al., 2016b),

potentially leading to an overprediction in AOD and direct radiative effect when the OPAC parameters are applied to the remote FT in global models. Additionally, the geometric mean diameters, both for number and for volume, differ considerably between the OPAC database and the ATom observations (Fig. 15a). For example, the accumulation-mode number geometric mean diameter $D_{g,n}$ in the observations is approximately twice that of the OPAC "water-soluble" fraction. This may be in part caused by the OPAC database including Aitken-mode particles in the "water-soluble" category.

The comparisons between the ship-borne measurements in Q17 and the ATom measurements are more direct, as both use similar modal fitting procedures and definitions for the modes, and are made primarily over the remote Pacific and Atlantic oceans. The modal fits from ATom and from Q17 are generally quite consistent. The Aitken and accumulation mode

parameters are similar between ATom and the ship-borne measurements, with the range of ATom parameters generally

narrower than the Q17 parameters, which span a longer time period and larger range of meteorological conditions than do

the airborne measurements. Both the Q17 and Atom data suggest values of $\sigma_g$ <1.9 in the MBL for both the accumulation

and coarse modes, while the OPAC database has a significantly larger value of $\sigma_g$ for these modes and all aerosol types. For

the coarse mode in the MBL, referred to as the "sea-spray" mode in Q17 and the "sea-salt" coarse mode in the OPAC

database, both the Q17 and ATom datasets report a value $D_{g,n}$ that is considerably smaller than that in OPAC and MAM7,

and a significantly larger $\sigma_g$. These differences are important, as sea salt is the single largest contributor to AOD over the

oceans (e.g., Haywood et al., 1999), and AOD (hence the direct radiative effect) is sensitive to these parameters (Brock et al.,

2016b).

### 4.1.2 Previous airborne campaigns

There have been a number of field programs that have made airborne aerosol measurements over many of the same regions

that ATom systematically sampled. One of most relevant analyses is that of Clarke and Kapustin (2010), who summarized

11 separate NASA airborne campaigns that included consistent aerosol measurements made by the same research group at

the University of Hawaii. These measurements include size distributions, $\sigma_s$, refractory and non-refractory particle number

concentrations, and proxies for CCN, as well as additional measurements specific to different campaigns. From these

measurements they have interpreted the abundances of sea-salt, dust, BC, and non-refractory (usually sulfate-organic)

particles. Most of their measurements were focused on the Pacific Ocean, but their analysis includes data measured over the

Southern, Arctic, and western Atlantic Oceans. The analysis of Clarke and Kapustin (2010) emphasizes the relationship

between refractory and non-refractory particle number concentration, CCN, light scattering, AOD, and carbon monoxide,

which is used as an indicator of combustion over the relevant time scales. Similar to Clarke and Kapustin (2010), the ATom

dataset shows very low values of scattering (<1 Mm$^{-1}$) in the FT of the remote Southern Hemisphere, and the clear influence

of combustion sources on aerosol abundance and properties throughout the troposphere in the northern midlatitudes. While

there are many opportunities to compare detailed aerosol properties, especially intensive values, between the ATom dataset

and that described by Clarke and Kapustin (2010), such detailed analysis is beyond the scope of this manuscript. In Sect. 4.2 we discuss some differences between the two datasets.

Another particularly relevant airborne field program, the HIPPO project, involved systematic profiling flights over the Pacific Ocean using an instrumented G-V business jet aircraft over all four seasons (Wofsy et al., 2011). The HIPPO project focused on gas-phase measurements, and contained only an SP2 instrument to measure rBC particles (Schwarz et al., 2013). An analysis using rBC data from both ATom-1 and HIPPO show very similar profiles over the Pacific Ocean (Katich et al., 2018). The additional data from ATom's Atlantic leg provides a marked contrast between the continentally influenced Atlantic and the more remote Pacific, with much higher rBC concentrations found over the Atlantic. Both data sets have proven useful in constraining global models that represent BC emissions and processes.

### 4.1.3 Mass absorption cross sections for rBC

Because light absorption by BC is a key uncertainty in global estimates of the direct radiative effect, it is useful to evaluate the assumptions we make in its calculation from the SP2 observations. We have calculated absorption assuming that the measured rBC particles are well-aged and compact with a density of $1.8\times10^3$ kg m$^{-3}$, and that core/shell Mie theory using the measured coating thickness, assumed to be a non-absorbing organic-sulfate mixture, provides a realistic approximation to their optical properties. Detailed consideration of different modeling approaches (Romshoo et al., 2021) suggests that core/shell Mie theory overestimates MAC values of coated BC by a factor of 1.1-1.5, with values increasing with increasing organic fraction (corresponding to coating thickness), for a fractal dimension for the BC core of 1.7, but with smaller discrepancies as fractal dimension increases toward 3 (a spherical core). Fierce et al. (2020) further report that core/shell Mie theory substantially overpredicts the absorption by BC in measurements in urban outflow, but that this discrepancy can be reduced by accounting for heterogeneity in particle composition and coating thickness. In contrast, Wu et al. (2021) found that core-shell Mie theory provided MAC values in agreement with, or even underestimating, directly measured MACs in aging biomass burning plumes downwind of West Africa. Zanatta et al. (2018) reported that core/shell Mie theory slightly underpredicted the measured MAC for aged, coated soot in the Arctic. China et al. (2015) found that aged soot particles

measured at a mountaintop site in the Azores had a compact morphology with thin coatings, and that radiative forcing

calculated using core/shell Mie theory was within 12% of that calculated using the discrete dipole approach.

Almost 30% by number of FT particles measured by PALMS during ATom were of BB origin (Schill et al., 2020). Thus we expect both compact core morphologies and substantial non-absorbing coatings in the rBC particles associated with the BB particles; these characteristics are supported by the coating thicknesses measured by the SP2 instrument (Supplemental Table S6) and by the small values of water-soluble BrC absorption measured in filter extracts (Zeng et al., 2020). The MAC values

for rBC we calculate from the ATom dataset using core/shell Mie theory were $14.4 \pm 1.4$ $m^2g^{-1}$ at a wavelength of 532 nm averaged over the free troposphere for all four ATom deployments (Supplemental Table S6). Values in identifiable BB plumes were $13.3 \pm 0.4$ $m^2g^{-1}$. These values are generally consistent with those measured at 514 nm in West African biomass burning plumes ranging from ~11.3 to ~14.2 $m^2g^{-1}$ for plume ages of ~1 to >9 h, respectively (Wu et al., 2021). Thus the MAC values we calculate using core/shell Mie theory appear to be reasonable given the likely BB source of most of the rBC.

Further, since no observations of soot morphology were made during ATom, we lack a basis for any additional refinement in our estimate of MAC values for coated rBC.

## 4.2 Considerations for using the ATom combined aerosol dataset

The ATom combined aerosol dataset presented here is unique in several ways. First, the aerosol composition measurements are not directly used; instead the relative abundance of the various species and aerosol types as a function of diameter are

mapped onto the size distribution. The integration of these composition-resolved size distributions provides the concentrations of the different aerosol components. Second, the optical properties and CCN concentrations were calculated from these size distributions, rather than directly measured. The exceptions to this process are the rBC and BrC concentrations and the light absorption associated with them. Third, ATom's strategy to make pre-planned, survey flight patterns (Thompson et al., submitted manuscript, 2021) means that the sampling was unbiased, with the exception of

deviations to avoid hazardous flight conditions (e.g., deep convection and low clouds) and to follow air traffic control instructions (e.g., sometimes staying below air traffic corridors over the North Atlantic Ocean). These features have resulted

in an aerosol dataset that is internally self-consistent (e.g., total scattering can be calculated by summing the scattering from the different composition-resolved size distributions) and absolutely unique in its representativeness and spatial coverage, ranging from 84 ºN to 86 ºS, from ~160 m to ~12 km over both the Pacific and Atlantic Ocean in four seasons.

One of the existing datasets most similar to the ATom aerosol dataset is that compiled by Clarke and Kapustin (2002, 2010). As with the ATom data, Clarke and Kapustin (2010) packaged their airborne data from multiple instruments, including several gas-phase species, into a dataset in netCDF format, and they have made it available through the Global Aerosol Synthesis and Science Project (GASSP) database (Reddington et al., 2017) and through the other data repositories indicated in Clarke and Kapustin (2010). However, there are several differences between the data analyzed by Clarke and Kapustin

(2010) and the ATom data presented here. Among the most significant are: 1) the ATom flights do not include any direct measurements of aerosol scattering or hygroscopicity (although dry extinction and absorption were directly measured on ATom-4); 2) the ATom flights included online measurements of aerosol composition and type from 0.05 to ~4 μm in diameter; 3) the ATom composition data are mapped to the size distributions rather than directly used; 4) many of the properties of the aerosol, including the optical properties, hygroscopicity, and CCN abundance, are calculated from these

composition-resolved size distributions rather than directly measured, and 5) the ATom flights used systematic survey sampling, while many of the flights analyzed by Clarke and Kapustin were focused on specific regions, events, or processes. The differences in instrumentation, sampling strategy, and spatial and temporal coverage provide opportunities to use both data sets to examine processes affecting aerosol properties and abundance and temporal and regional differences, and to constrain the global model simulations of aerosols. Additional data compiled in the GASSP archive, including extensive

measurements made at various locations around the world on the British Facility for Airborne Atmospheric Measurements BAE-146 aircraft, provide further information on detailed aerosol optical, microphysical, and chemical properties. Although the combined ATom aerosol dataset offers a comprehensive and detailed picture of global-scale aerosol properties, it is limited in important ways. Most significantly, the ATom measurements do not represent a climatology, although they are representative of seasonally typical values for a subset of measured parameters that have been compared to climatologies

(Strode et al., 2018; Bourgeois et al. 2020). The four circuits around the globe, once in each season, provide a snapshot of

aerosol conditions at those particular times without targeting specific phenomena, unlike most airborne projects. Comparisons between models and the ATom data will be most effective if meteorology and emissions are prescribed or nudged to match the times of the ATom flights, and if the model domain is sampled along the aircraft flight track. Similarly, comparisons with remote sensing measurements should overlap in space and time to the extent possible.

There are limitations to specific aspects of the data presented here, as well. The compositional data we consider in this combined dataset represent only a fraction of the richness of the data from the HR-TOF-AMS and PALMS spectrometers and of the filter-based bulk measurements. Data from these instruments include detailed information on molecular markers of specific sources and processes (e.g., *f57*, *f44*), elemental composition of OA (H/C and O/C ratios; Hodzic et al., 2020), ionic balance and acidity (Nault et al., 2021), speciation of inorganic ions, and the presence of rare particle types (e.g.,

Murphy et al., 2018). Potential users of the data are encouraged to communicate with the instrument teams to make full use of the available information in their analyses.

The particle size distributions are measured using a condensation technique (the NMASS battery of CPCs) which report a Kelvin (condensation) diameter. These data are combined with an optical particle spectrometer (the UHSAS) which measures an optical size. Thus discontinuities can occur at the boundary between the instruments, at about 60 nm (Brock et

al., 2019). Unfortunately, this is near the critical diameter for CCN activation at typical water supersaturations for stratocumulus and cumulus clouds. Smoothing is used to minimize potential discontinuities. At diameters from 0.6–2 µm, the laser optical particle spectrometers are in a regime of Mie oscillations, where particle sizing is relatively insensitive, or even ambiguous. This can cause spurious high-frequency features in the size distribution in this size range, as noted in Sect. 3.3.2. These features do not substantially affect the optical properties or modal parameters, but could be misinterpreted as

physical attributes.

The modal parameters fitted to the size distributions rely upon a priori assumptions regarding the number of modes and their characteristics (see Supplemental Materials). There are cases in the remote FT when the Aitken and accumulation modes are subjectively indistinguishable, yet the fitting procedure attempts to fit two modes. The user of the combined dataset is cautioned that there are times when the accumulation mode might actually be an extension of a single Aitken mode. In

addition, when there are very few coarse mode particles, the fitting algorithm may still attempt to describe the few counts present with lognormal parameters, leading to excessive noise in the modal parameters.

    The composition of the coarse mode is measured using the PALMS instrument sampling behind an inlet that removes particles with $D_p$>4.8 µm in the lower troposphere and >3.2 µm in the UT (McNaughton et al., 2007). The composition of larger particles measured by the underwing CAS instrument is assumed to be the same as those in the largest PALMS size

class (1.13 to ~4 µm). If the composition of particles with $D_p$ >4 µm measured by the CAS is different, this will produce a bias. This potential bias is likely to be small in the MBL since these larger particles are almost certainly sea-spray aerosol. In calculating optical properties, these coarse-mode particles are assumed to be spherical; no attempt has been made to simulate dust or sea salt properties using non-spherical approaches.

    We have not attempted to propagate uncertainties beyond the size distribution uncertainties described by Brock et al. (2019),

based on comprehensive instrument evaluations by Kupc (2018) and Williamson (2018). The final average uncertainty in integrated particle volume is estimated to be +13/-28% for the accumulation mode when counting statistics are not a limiting factor. Integrated aerosol volumes determined independently from the size distributions and from the AMS instrument are highly consistent (Guo et al., 2021), which lends confidence to the measurements. Determining uncertainties associated with applying composition data to the size distributions, with calculating hygroscopic growth, or with determining the resulting

optical properties and CCN concentrations would require Monte Carlo simulations over a large number of parameters for each of >$2.4\times10^4$ measurements, which is impractical. Comparisons of calculated dry extinction and absorption with directly measured values during ATom-4 (Fig. 4) suggest errors in dry extinction and absorption of <20%, while comparisons of the derived AOD with directly measured values from nearby AERONET sites (Fig. 6) show no substantial biases. While the normalized mean bias was only -7%, there was considerable scatter in the comparison and it is not possible to disentangle

atmospheric inhomogeneity from measurement uncertainty given the spatial mismatch between the slantwise aircraft profiles and the AERONET locations. We hope to gain a better understanding of errors in the ATom AOD product through ongoing comparisons with satellite observations.

Finally, we note that we anticipate continued evolution of this publicly available dataset. Despite our best efforts, there are undoubtedly errors or inconsistencies that will need to be corrected, as well as newly calculated parameters that could

enhance its usefulness. We encourage users of the data to report any issues or suggestions for improvement to the lead author.

## 5. Conclusions

The ATom project made four surveys, once in each season, of the composition of the remote, oceanic troposphere and portions of the lower stratosphere at high latitudes. The aircraft repeatedly profiled between ~160 m and ~12 km, mapping

out the vertical and horizontal variation in aerosol and gas-phase properties. We have combined dry aerosol composition and size distribution measurements made over the remote Pacific and Atlantic Oceans, as well as over portions of the Arctic and Antarctic, to comprehensively describe the chemical, microphysical, and optical characteristics of the aerosol. Inorganic electrolyte composition was determined using an algebraic composition model, and aerosol water was then estimated using $\kappa$-Köhler theory. From the hydrated, composition-resolved size distributions, we have calculated a number of intrinsic and

extrinsic parameters that are related to the climate effects of the aerosol. These parameters include various optical properties at 10 wavelengths, cloud condensation nuclei concentrations at 5 supersaturations, and lognormal fits to 4 modes of the particle size distribution. Mid-visible dry extinction and absorption coefficients calculated from the composition-resolved size distributions were in excellent agreement with directly measured dry extinction and absorption coefficients made with independent instruments during the ATom-4 deployment. Mid-visible AOD was calculated by vertically integrating ambient

extinction values during profiles, and showed little bias compared with values directly measured with AERONET sunphotometers, despite substantial scatter due to the distances between the slantwise profiles and the AERONET sites. Initial findings from the combined dataset show that the remote Northern Hemisphere lower and free troposphere has considerably more aerosol from continental sources than does the Southern Hemisphere, consistent with understanding gained from past in situ studies (e.g., Clarke and Kapustin, 2010). Dust and sulfate/organic mixtures contribute substantially

to AOD in the middle troposphere over the midlatitude northern Pacific Ocean and the lower and middle troposphere over

the tropical Atlantic Oceans. Unsurprisingly, sea-salt particles and associated water dominate AOD over most of the remote oceans, especially in the Southern Hemisphere, while BB particles contribute over the subtropical and tropical Atlantic Ocean and to a lesser extent over the North Pacific. Single scatter albedo was found to vary substantially with altitude due to changes in both composition and size. The geometric standard deviation of lognormal fits to the Aitken and accumulation modes generally lay between 1.5 and 2.0, narrower than values in some modal representations used in global models. Within the MBL, the lognormal parameters for these modes and for the coarse mode are generally consistent with values from extensive shipboard measurements in the remote oceans.

The ATom aerosol dataset presented here is unique in that on-line, size-resolved aerosol composition measurements have been mapped to aerosol size distributions, thus providing separate size distributions for several different aerosol constituents. These data products more closely match the way aerosol components are treated in global models than is typical for other airborne datasets, and is, to our knowledge, unique. From these composition-resolved size distributions, hygroscopicity, CCN concentrations, and optical properties have been calculated, resulting in a single, self-consistent, global-scale dataset for use by the scientific community. The global-scale mapping of atmospheric composition provided by ATom's representative, profiling survey flights, while not a climatology with statistical information on time-varying properties, provides unique information that can help constrain model representations of aerosol emissions, transport, removal, and processing, as well as a priori assumptions used in retrievals of aerosol properties from remote sensing measurements. The data are accessible for public scientific use as described in the data availability statement below.

*Author Contributions.* All authors contributed to the observations described in this manuscript. CAB, KDF, MD, CJW, GS, DMM, NLW, SW, PCJ, LZ, and JMK performed the key analyses. CAB wrote the manuscript with significant contributions from KDF, MD, CJW, GS, DMM, NLW, PCJ, JLJ, JMK., JD, and JP.

*Competing Interests.* The authors declare that they have no conflicts of interest.

*Disclaimer*. The contents do not necessarily represent the official views of NOAA or of the respective granting agencies. The use or mention of commercial products or services does not represent an endorsement by the authors or by any agency.

*Acknowledgements*. This work was supported by NASA under awards NNH15AB12I, NNX15AJ23G, NNX15AH33A,

NNH13ZDA001N, NNX15AT90G, 80NSSC19K0124, and 80NSSC18K0630, and by the U.S. National Oceanic and

Atmospheric Administration (NOAA) Atmospheric Chemistry, Carbon Cycle, and Climate Program. AK is supported by the

Austrian Science Fund FWF's Erwin Schrodinger Fellowship J-3613. BW and MD have received funding from the

European Research Council (ERC) under the European Union's Horizon 2020 Research and Innovation Programme under

grant agreement No. 640458 (A-LIFE) and from the University of Vienna. We thank the following investigators for their

effort in establishing and maintaining the AERONET sites used in this analysis: Resolute Bay–Ihab Abboud and Vitali

Fioletov; ARM-Oliktok, ARM-Graciosa, and ARM-Southern Great Plains–Rick Wagener and Lynn Ma; ARM Macquarie

Island–Lynn Ma; Mauna Loa, Ascension Island, and Kangerlussuaq–Brent Holben; CEILAP RG–Brent Holben, Eduardo

Quel, Lidia Otero, and Jacobo Salvador. The GLOSSAC stratospheric AOD data were obtained from the NASA Langley

Research Center Atmospheric Science Data Center.

*Data Availability.* The ATom data products described in this manuscript are publicly available at the Oak Ridge National Laboratory Distributed Active Archive Center (https://doi.org/10.3334/ORNLDAAC/1908). Additional ATom data are available at https://doi.org/10.3334/ORNLDAAC/1925. AERONET data are available at https://aeronet.gsfc.nasa.gov.

Table 1. Aerosol properties calculated from the combined aerosol dataset and archived in files.

| Parameter | Parameter identifier[1] | Method | wavelengths | Comments |
|---|---|---|---|---|
| Dry scattering | scat_dry_ambpt | Mie theory from composition-resolved size distribution using refractive indices in Table 2 | all[2] | Calculated at ambient pressure and temperature; Sect. 2.7.1 |
| Dry absorption from rBC | BC_abs_ambPT | Core-shell Mie theory using air mass-averaged MAC multiplied by 60s rBC mass concentration | all[2] | Calculated at ambient pressure and temperature; Sect. 2.7.1 |
| Dry absorption from BrC | BrC_abs_ambPT | Bivariate fit between BrC absorption from filter extracts and PALMS biomass burning particles and rBC mass concentrations | all[2] | Calculated at ambient pressure and temperature; Sect. 2.7.1; estimated factor of 3 uncertainty |
| Dry extinction | ext_dry_ambPT | Sum of dry scattering and absorption from rBC and BrC | all[2] | Calculated at ambient pressure and temperature |
| Ambient scattering | scat_ambRHPT | $\kappa$-Köhler theory to estimate water content; Mie theory to calculate scattering | all[2] | Calculated at ambient pressure and temperature |
| Ambient extinction | ext_ambRHPT | Ambient scattering + dry absorption from rBC and BrC | all[2] | Calculated at ambient pressure and temperature |
| Dry single scatter albedo | SSA_dry | Dry scattering and extinction | all[2] | Ratio of scattering to extinction |
| Ambient single scatter albedo | SSA_ambRH | Ambient scattering and extinction | all[2] | Ratio of scattering to extinction |
| Dry extinction Ångström exponent | ext_Angstrom_dry | Fit to dry extinction across all wavelengths[1] | all[2] | least squares regression to Eq. 11 |
| Ambient extinction Ångström exponent | ext_Angstrom_ambRH | Fit to ambient extinction at all wavelengths[1] | all[2] | least squares regression to Eq. 11 |
| UV-Vis absorption Ångström exponent | abs_Angstrom_UV_Vis | Fit to sum of dry absorption from rBC and BrC | 340, 380, 405, 440, 532 nm | least squares regression to Eq. 11 |
| Vis-IR absorption Ångström exponent | abs_Angstrom_Vis_IR | Fit to dry absorption from rBC | 532, 550, 670, 940, 1020 nm | least squares regression to Eq. 11 |
| Dry mass extinction efficiency | MEE_dry | Dry extinction and dry aerosol mass from composition resolved size distributions and densities in Table 2 | all[2] | ratio of dry extinction to dry aerosol mass |
| Ambient mass extinction efficiency | MEE_ambRH | Ambient extinction and dry aerosol mass | all[2] | ratio of ambient extinction to dry aerosol mass |
| Mass absorption cross-section | MAC | Core-shell Mie theory applied to coated rBC particles | all[2] | Ratio of coated rBC absorption to rBC mass; calculated for air mass averages only; Table S6 |
| Dry asymmetry parameter | asymmetry_dry | Mie theory at dry conditions, not including absorbers | all[2] | Eq. 9 |
| Ambient asymmetry parameter | asymmetry_ambRH | Mie theory at ambient conditions, not including absorbers | all[2] | Eq. 9 |
| Ambient lidar backscatter ratio | backscat_ratio_ambRH | $\kappa$-Köhler theory to estimate water content; Mie theory to calculate backscattering and scattering | all[2] | Ratio of backscatter to extinction at ambient RH |
| Ambient lidar backscatter cross-section | backscat_ambRH | $\kappa$-Köhler theory to estimate water content; Mie theory to calculate backscattering, dry aerosol mass | all[2] | Ratio of backscatter at ambient RH to dry particle mass |
| Effective radius | eff_radius | Integration of size distribution | — | Ratio of 3rd moment of the size distribution to the 2nd moment |
| Hygroscopicity parameter $\kappa$ | kappa_ams | Volume-weighted sum of $\kappa$ values from AMS in Table 2 | — | Algebraic calculation of electrolytic composition; literature values |
| $f(RH)_{85\%}$ | f_rh_85 | Ratio of calculated extinction at 85% Rh to that at dry conditions | 532 nm only | |

| $\kappa_{ext}$ | *kappa_ext* | Fit to calculated extinction at 0, 70, 80, and 85% RH | 532 nm only | Fit to Eq. 10 |
| --- | --- | --- | --- | --- |
| CCN concentration | *CCN_005, CCN_010, CCN_020, CCN_050, CCN_100* | Integration of particle size distribution for $D_p > D_{crit, dry}$ | — | Eq. 2, Sect. 2.6; calculated for supersaturations of 0.05%, 0.1%, 0.2%, 0.5%, and 1.0% |
| Lognormal parameters $D_g$, $\sigma_g$, $N$ | *lognorm_coefs_nucl, lognorm_coefs_Aitken, lognorm_coefs_accum, lognorm_coefs_coarse* | Fits to volume for coarse and accumulation mode and to number for Aitken and nucleation modes. | — | Supplemental materials, Tables S2-S4. |
| Mass concentration of sulfate, organics, dust, rBC, BrC, aerosol water | *sulfate, organics, nitrate, ammonium, sea_salt, dust, BC, BrC_est, aerosol_H2O, mass_fine, mass_coarse* | Integration of volume size distribution for each component multiplied by density from Table 2, separated into coarse ($D_p \geq 1\ \mu m$) and fine ($D_p < 1\ \mu m$). | — | Ammonium and nitrate from AMS applied to sulfate/organic class across all sizes |
| Ambient fine mode extinction fraction $\eta$ | *FMF* | Coarse and accumulation-mode compositions applied to lognormal fits to those modes, then Mie theory used to calculate extinctions for each | all[2] | Aerosol water calculated using $\kappa$-Köhler theory & values from Table 2 |

[1]Identifier of variable (short name) in netCDF file
[2]340, 380, 405, 440, 532, 550, 670, 870, 940, 1020 nm


Table 2. Assumed values of hygroscopicity parameter $\kappa$, density $\rho$, and refractive index.

| Instrument: Parameter | Hygroscopicity parameter $\kappa$[A] | Reference | Density $\rho$ (kg m$^{-3}$) | Reference | Refractive Index | Reference |
|---|---|---|---|---|---|---|
| PALMS: sulfate/organic particles | $(1-F_{org})$[B]$\times 0.73$ $+F_{org}\times 0.17$[C] | Froyd et al. (2019) | $(1-F_{org})\times 1770$ $+F_{org}\times 1350$[C] | Froyd et al. (2019) | $((1-F_{org})\times 1.44$ $+F_{org}\times 1.48)+0i$ | Froyd et al. (2019) |
| PALMS: biomass burning, heavy fuel oil combustion, and meteoric | $(1-F_{org})\times 0.73$ $+F_{org}\times 0.17$ | Froyd et al. (2019) | $(1-F_{org})\times 1770$ $+F_{org}\times 1350$[C] | Froyd et al. (2019) | $((1-F_{org})\times 1.44$ $+F_{org}\times 1.48)+0i$ | Froyd et al. (2019) |
| PALMS: Soot (assumes small soot core with thick coating) | $(1-F_{org})\times 0.73$ $+F_{org}\times 0.17$ | Froyd et al. (2019) | $(1-F_{org})\times 1770$ $+F_{org}\times 1350$[C] | Froyd et al. (2019) | $((1-F_{org})\times 1.44$ $+F_{org}\times 1.48)+0i$ | Froyd et al. (2019) |
| PALMS: Sea salt | 1.1 | Zieger et al. (2017) | 1800[C] | Froyd et al. (2019) | $1.447+0i$[D] | Froyd et al. (2019) |
| PALMS: Mineral dust | 0.03 | Froyd et al. (2019) | 2500 | Froyd et al. (2019) | 530 nm: $1.55+0.002i$[E] | Weinzierl et al. (2009) |
| PALMS: Alkali salts | 0.5 | Froyd et al. (2019) | 1500 | Froyd et al. (2019) | $1.52+0i$ | Froyd et al. (2019) |
| SP2: Black Carbon | N/A[F] | | 1800 | Park et al. (2004) | $2.26+1.26i$ | Moteki et al. (2010) |
| SP2: Coating | N/A[F] | | N/A[F] | | $1.44+0i$ | mean of AMS for all of ATom |
| Calculated: H$_2$O | N/A[F] | | 1000 | | $1.33+0i$ | Hale and Querry (1973) |
| AMS: (NH$_4$)$_2$SO$_4$ | 0.483 | Good et al. (2010) | 1760 | Hand and Kreidenweis (2002) | $1.527+0i$ | Hand and Kreidenweis (2002) |
| AMS: (NH$_4$)HSO$_4$ | 0.543 | Good et al. (2010) | 1780 | Hand and Kreidenweis (2002) | $1.479+0i$ | Hand and Kreidenweis (2002) |
| AMS: (NH$_4$)$_3$H(SO$_4$)$_2$ | 0.579 | Good et al. (2010) | 1830 | Hand and Kreidenweis (2002) | $1.53+0i$ | Hand and Kreidenweis (2002) |
| AMS: H$_2$SO$_4$ | 0.87 | Petters and Kreidenweis (2007) | 1800 | Hand and Kreidenweis (2002) | $1.408+0i$ | Hand and Kreidenweis (2002) |
| AMS: NH$_4$NO$_3$ | 0.597 | Good et al. (2010) | 1725 | Tang, 1996 | $1.553+0i$ | Tang (1996) |
| AMS: NH$_4$Cl | 0.5 | assumed[G] | 1519 | Haynes et al. (2014) | $1.64+0i$ | Haynes et al. (2014) |
| AMS: HNO$_3$ | 0.999 | Good et al. (2010) | 1513 | Haynes et al. (2014) | $1.393+0i$ | Haynes et al. (2014) |
| AMS: HCl | 0.5 | assumed | 1490 | Haynes et al. (2014) | $1.329+0i$ | Haynes et al. (2014) |
| AMS: OA | $0.19\times$(O/C)$-$ $0.0048$[H] Mean=0.179 | Rickards et al. (2013) | 1550[C] | Guo et al. (2021) average from ATom-1 and -2 | $1.48+0i$ | Varma et al. (2013) |

[A]PALMS $\kappa$ values are applied to refractory and non-refractory components for all $D_p>0.25$ µm. AMS values are applied to all non-refractory components for $D_p\leq 0.25$µm. The Zaveri et al. (200x) composition model provides speciation of AMS components.

[B]$F_{org}$ is the ratio of organic to organic+sulfate mass in that size class determined by the PALMS instrument.

[C]Organic density applied to PALMS is chosen from Froyd et al. (2019) for consistency with other PALMS data products, but is inconsistent with AMS-derived density from Guo et al. (2021) applied here to AMS data.

[D]Assumes 27% residual water by mass (Froyd et al., 2019).

[E]Imaginary component of refractive index for mineral dust assumed to vary with wavelength using Eq. 11. with Ångström exponent of 3.

[F]Not applicable: this parameter not used in any calculations.

[G]Assumed value is not critical because these species are an insignificant part (<0.5%) of the total fine aerosol mass.

[H]O/C is the O:C ratio from the HR-ToF-AMS measurements. The O:C ratios are smoothed with a running 10-point binomial filter (across ~10 minutes of data) before this equation is applied.

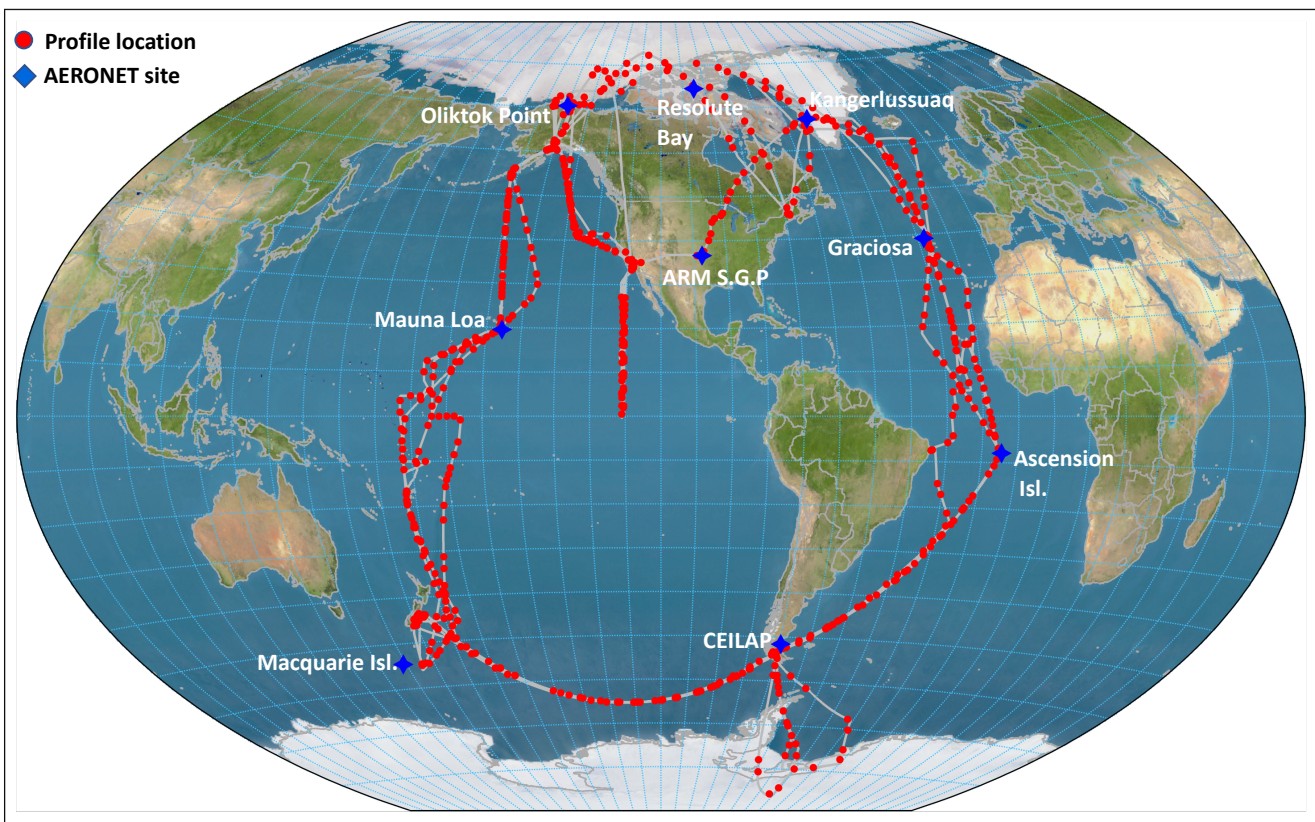

**Figure 1. Map showing the flight track of the DC-8 aircraft (grey lines) and midpoint location of each vertical profile (ascent or descent; red circles). Locations and names of AERONET sites against which calculated AOD is compared are shown by blue diamonds and labels. Custom map produced using 1 km digital elevation model data from NOAA (https://www.ngdc.noaa.gov/mgg/topo/globe.html; last accessed 3 February 2016).**

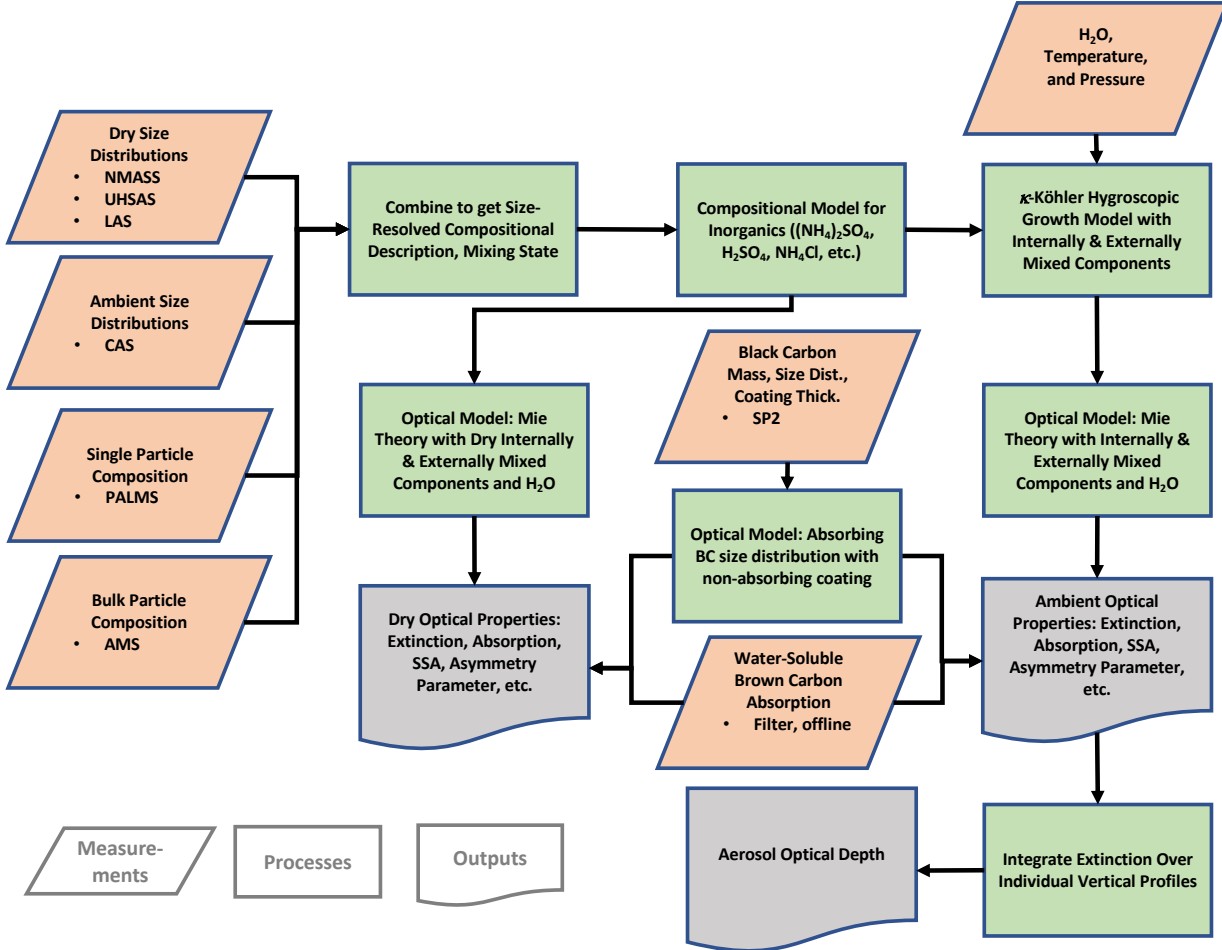

**Figure 2. Schematic showing how data from instruments that measure size distribution, particle composition and meteorological parameters are combined to form a self-consistent description of the composition-dependent size distribution. Compositional, hygroscopic growth, and optical models are combined to determine dry and ambient aerosol optical properties and AOD.**


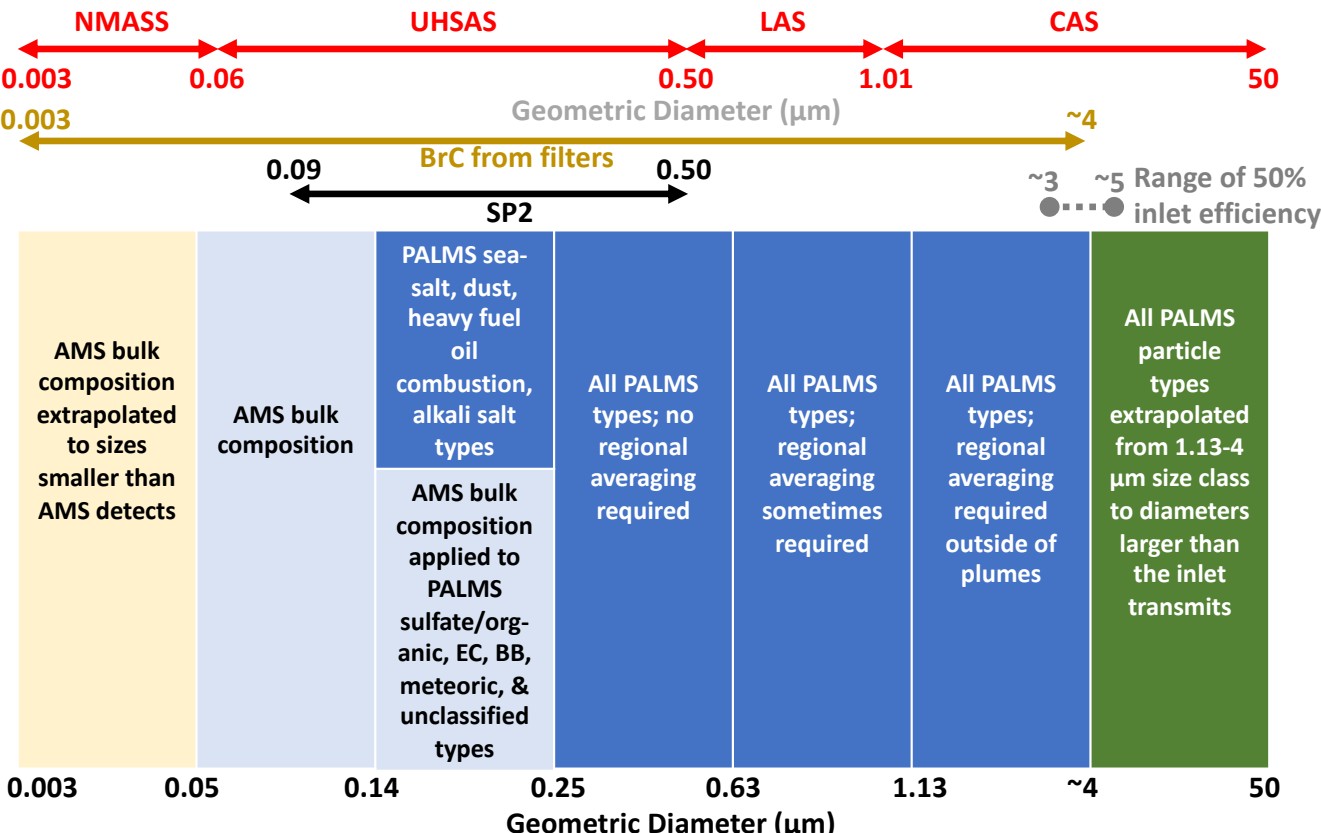

Figure 3. Schematic showing the portions of the size ranges of the particle size distribution instruments that are used, and the size ranges over which the composition measurements from the filter measurements, the SP2, the AMS, and the PALMS are applied. The approximate range of 50% inlet transmission efficiency is shown. The needs to average PALMS data to achieve statistically significant descriptions of particle composition are shown, as are the extrapolations of AMS and PALMS data to sizes where no compositional information is available. The diameter ranges of instrument detection are presented in detail in Guo et al. (2021).

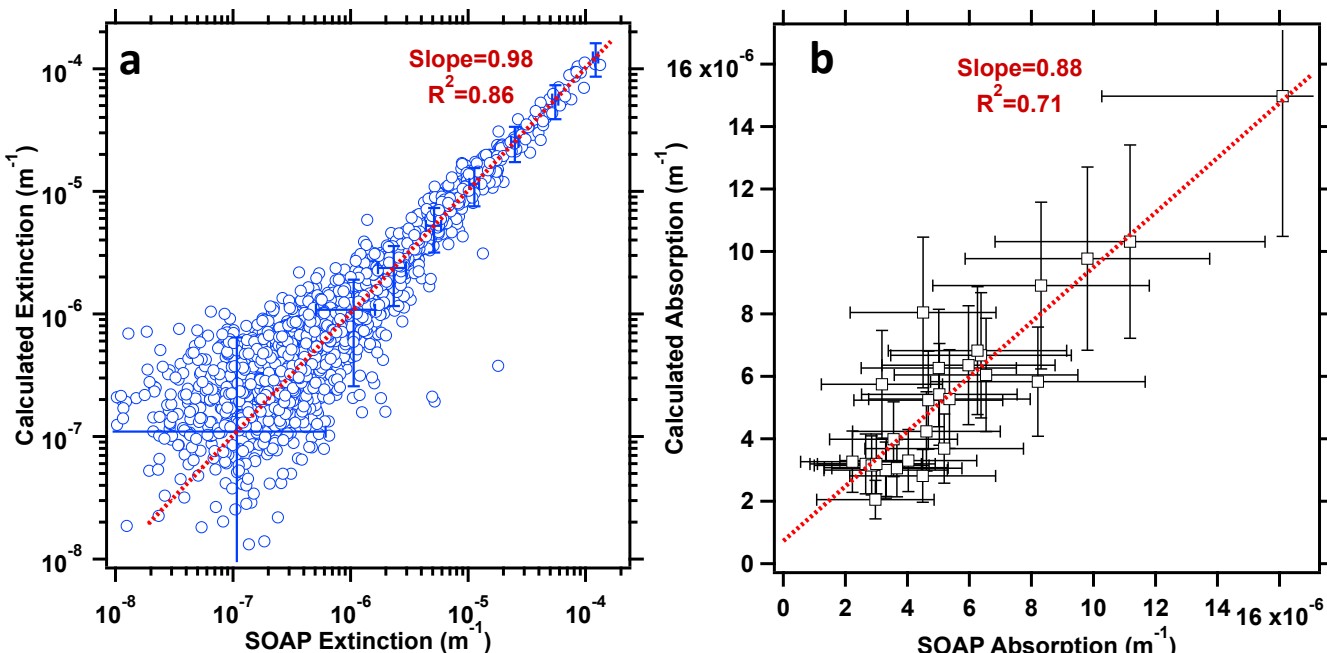

**Figure 4. a) Calculated aerosol extinction as a function of measured extinction from the SOAP cavity ringdown spectrometer during ATom-4, showing representative error bars. b) As in (a), but for calculated aerosol absorption and measured absorption from the SOAP photoacoustic spectrometer for cases when absorption $>2\times10^{-6}$ m$^{-1}$ (2× the detection limit). Lines and slopes are from two-sided (orthogonal distance) linear regressions accounting for uncertainties; r$^2$ values are from one-sided fits. The fitted line in (a) was determined from logarithmically transformed data (log(y) vs log(x) regression).**

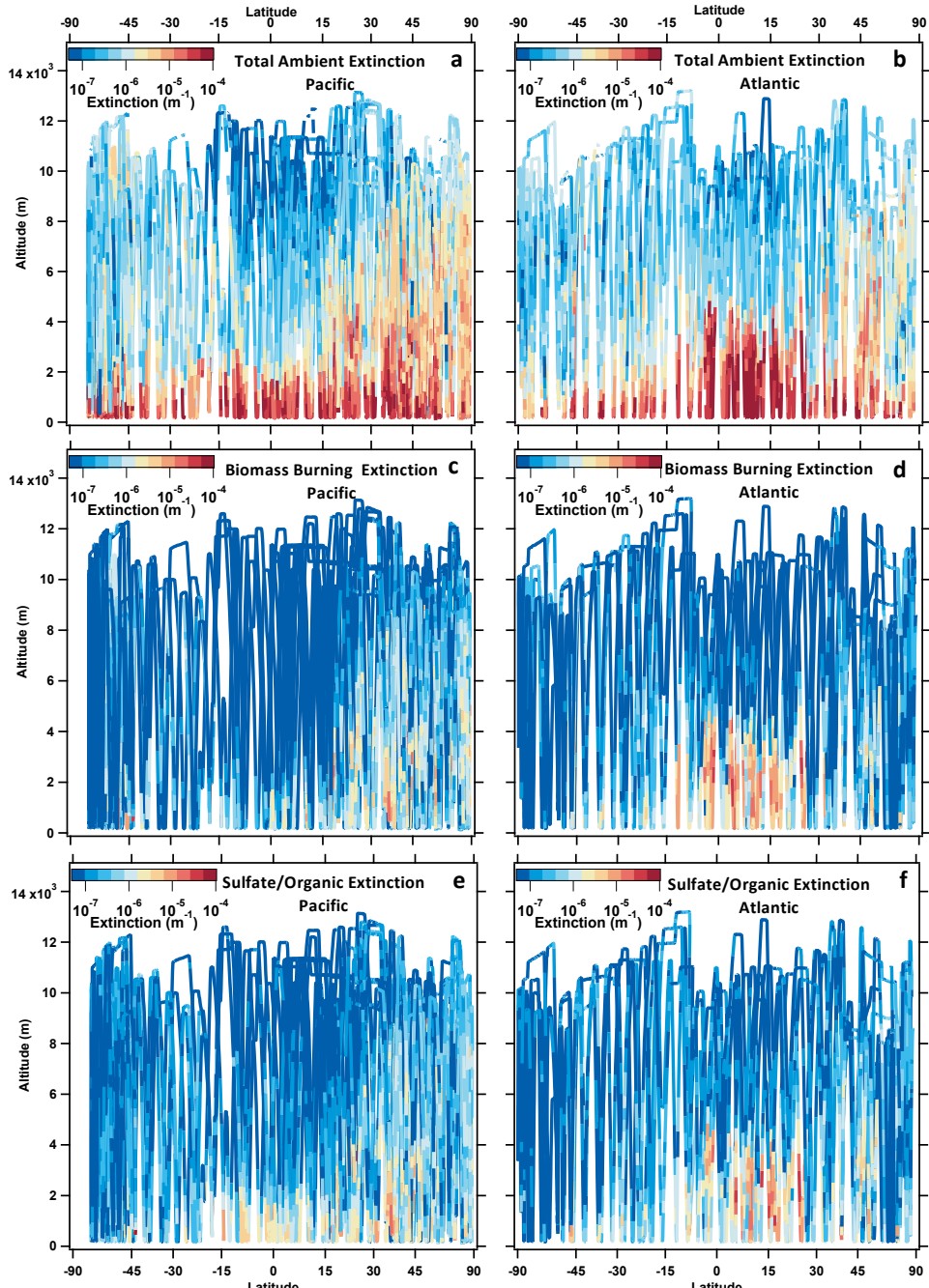

**Figure 5. Altitude as a function of latitude color coded by extinction for all ATom deployments. X-axis is scaled to be proportional the Earth's surface area. Left column shows measurements made over the Pacific Ocean, western Arctic, and Southern Ocean; right column over the Atlantic, eastern Arctic, and Antarctic Peninsula (see Fig. 1). (a) and (b), total ambient extinction; (c) and (d), dry extinction from biomass burning particles; (e) and (f), dry extinction from mixed sulfate/organic particles.**

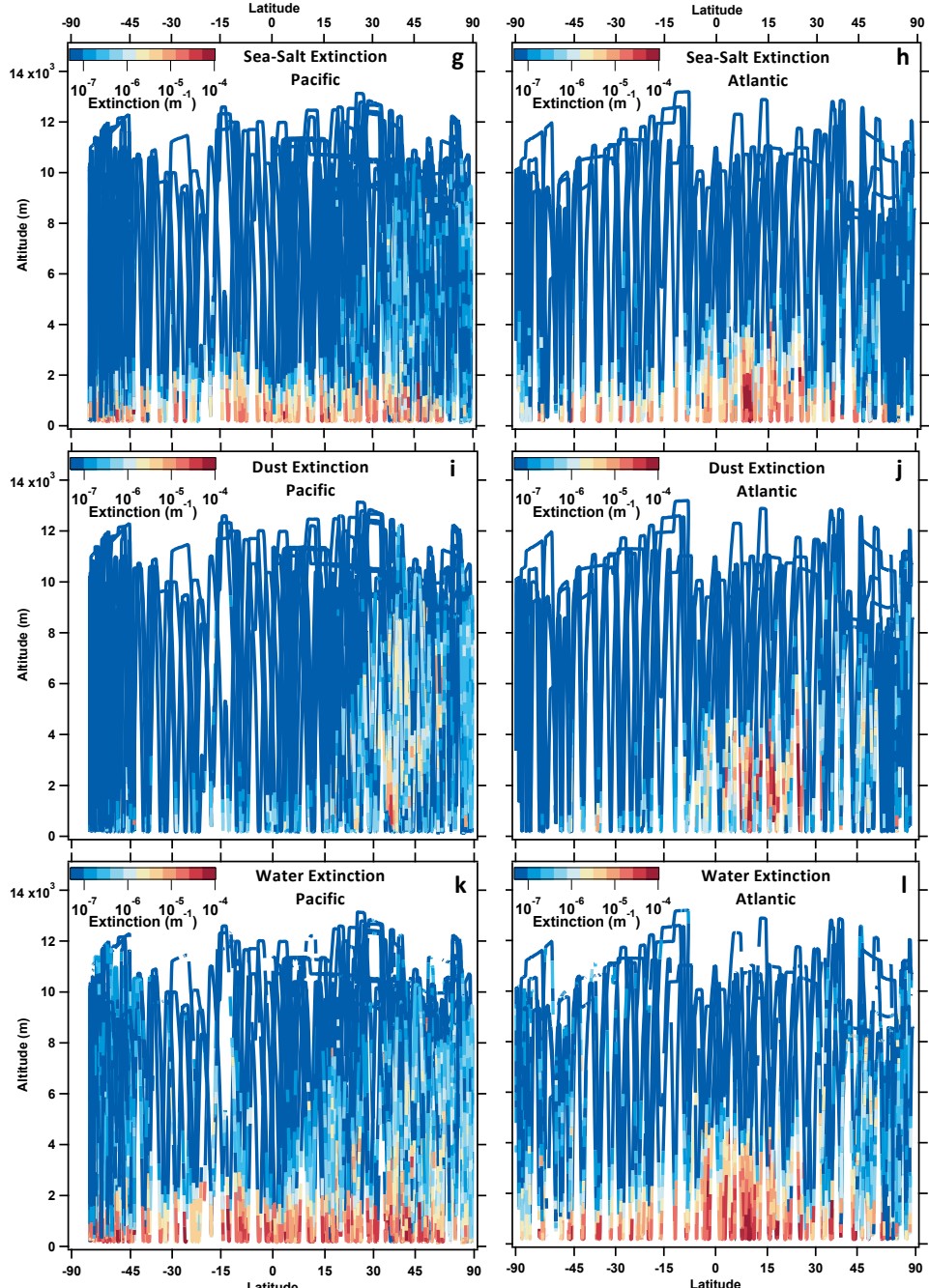

**Figure 5 (continued). (g) and (h), dry extinction from sea-salt particles; (i) and (j), dry extinction from dust particles; (k) and (l), extinction from water associated with all particle types, based on κ-Köhler hygroscopic growth model.**

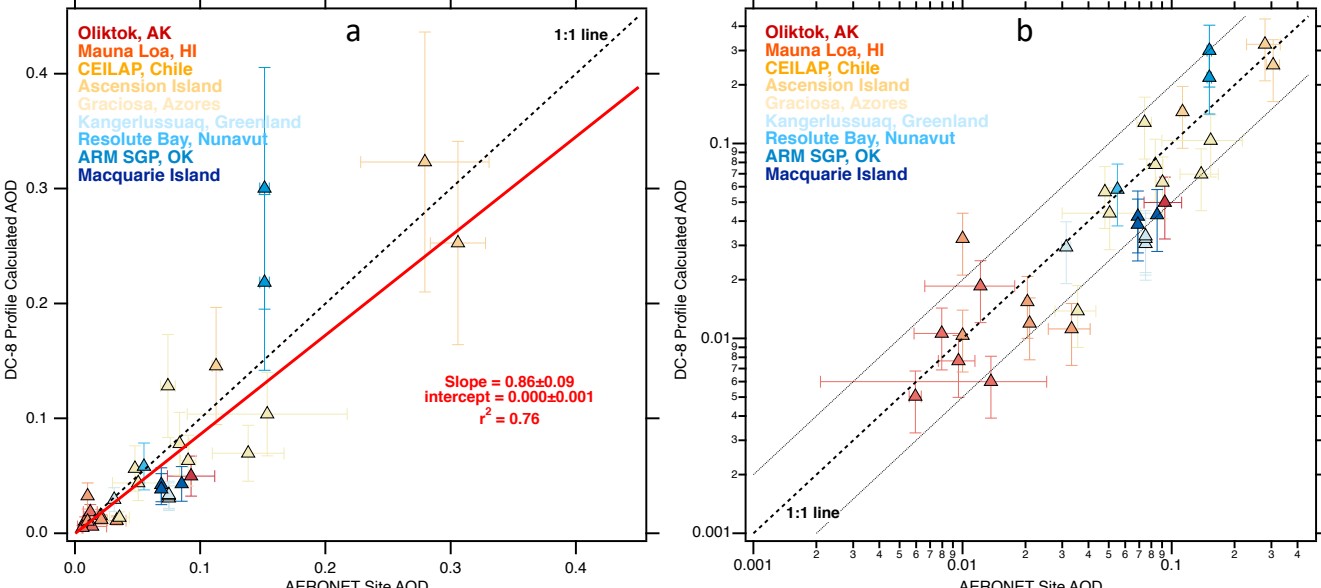

**Figure 6. Aerosol optical depth (AOD) at 532 nm calculated from the in situ aerosol measurements on the DC-8 as a function of AOD measured by AERONET sites within 300 km and ±4 hours of the profile. a) Linear plot. Two-sided linear regression (red line) accounts for x and y uncertainties. b) As in a, but a log-log plot. Dashed line is the 1:1 line and dotted lines are a factor of two higher and lower. AERONET AOD at 532 nm is interpolated from measurements at 500 and 670 nm following Eq. 10. One outlier data point has been removed. Horizontal error bars indicate the variability in the AERONET AOD in ±4 hours surrounding the measurement time. Vertical error bars indicate an approximate ±30% uncertainty in the AOD derived from in situ measurements. Locations of the AERONET sites are given in Fig. 1.**

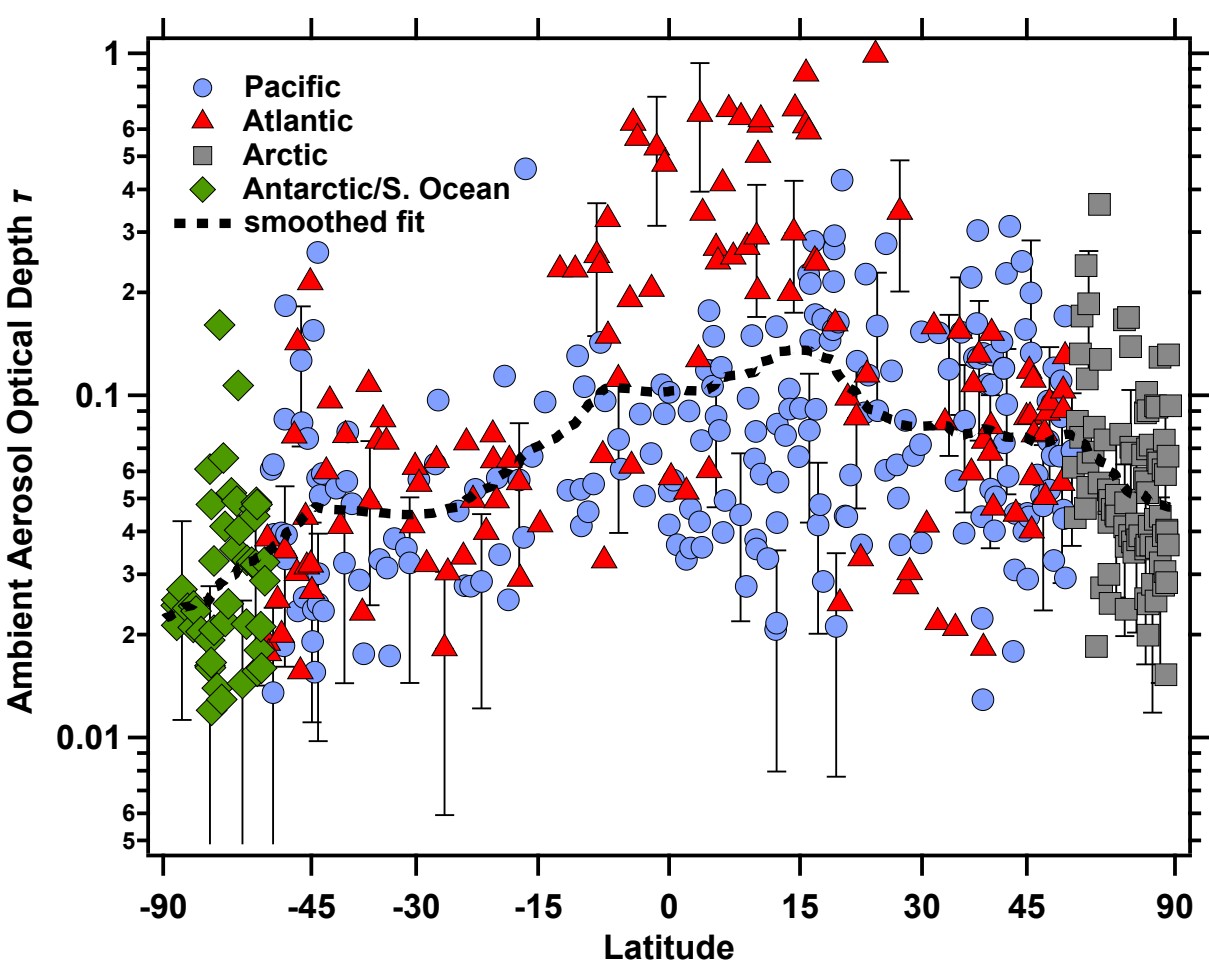

155     Figure 7. Ambient AOD calculated from in situ measurements as a function of latitude. Symbols indicate data taken over the
Atlantic, Pacific, Southern Ocean and Antarctica, and the Arctic, with these regions described in Table S1. The smoothed dashed
line is calculated using a locally weighted linear (LOWESS) regression to the logarithm of the AOD values.

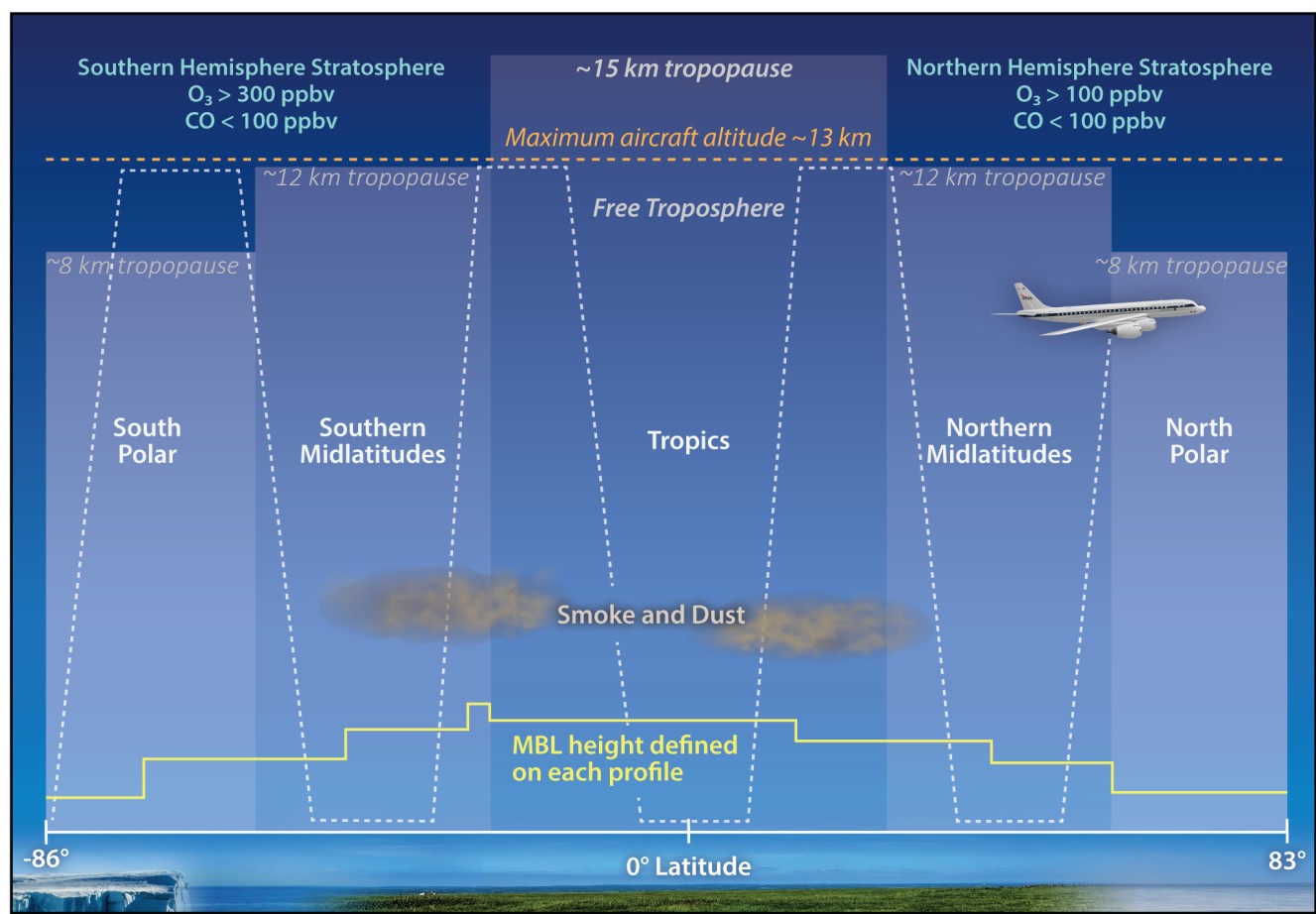

**Figure 8. Schematic representation of the air mass classification scheme. The boundaries between the polar, midlatitude, and tropical air masses vary for each ATom deployment and ocean basin, and are listed in Table S1 in the Supplemental Materials. Data taken in biomass burning smoke ("smoke") and mineral dust ("dust") plumes are combined when concentration criteria are met (Sect. 2.3) regardless of latitude, while stratospheric regions are separated into northern or southern hemispheres because of different aerosol characteristics in each (Murphy et al., 2021).**

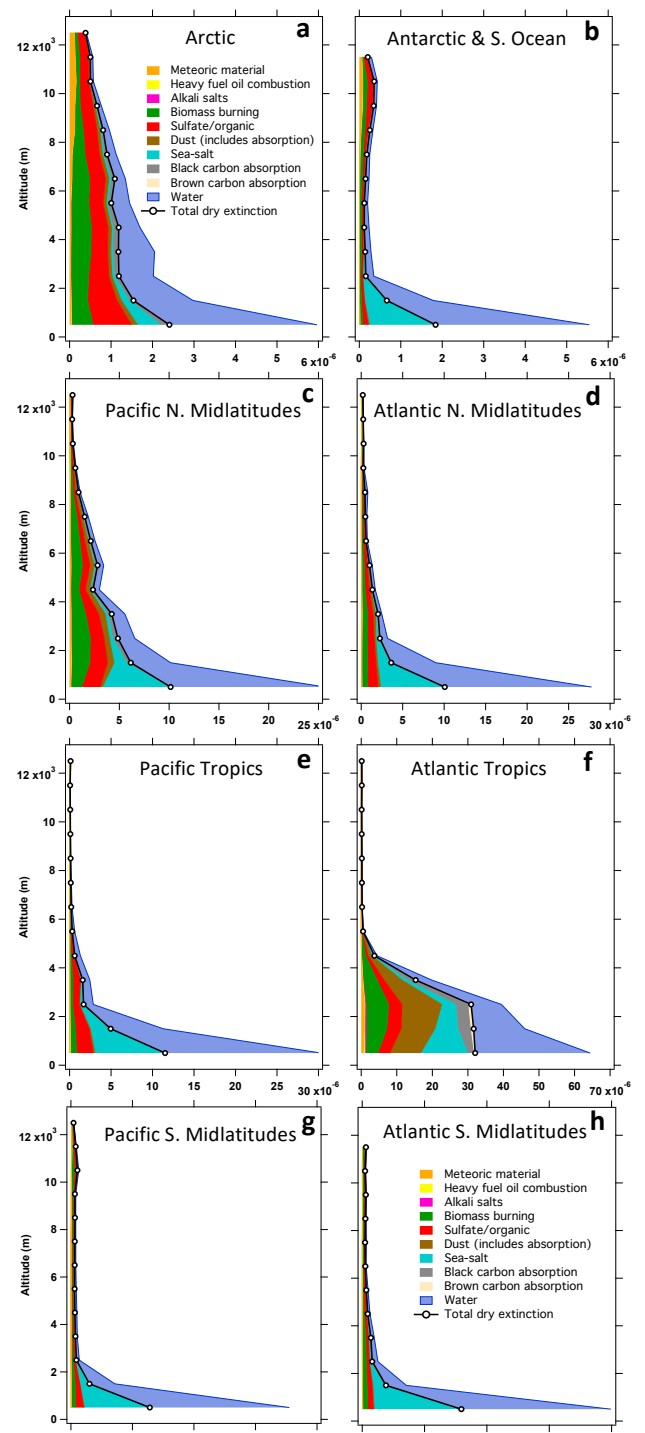

**Figure 9. Mean vertical profiles of extinction from each of the aerosol types, for different regions, across all of the ATom deployments. Note that scales on the x-axes vary. Descriptions of the regions are given in Fig. 8 and Table S1. The fractional contribution of the aerosol types to total extinction and the number of data points in each 1-km altitude bin are given in Supplemental Fig. S5.**

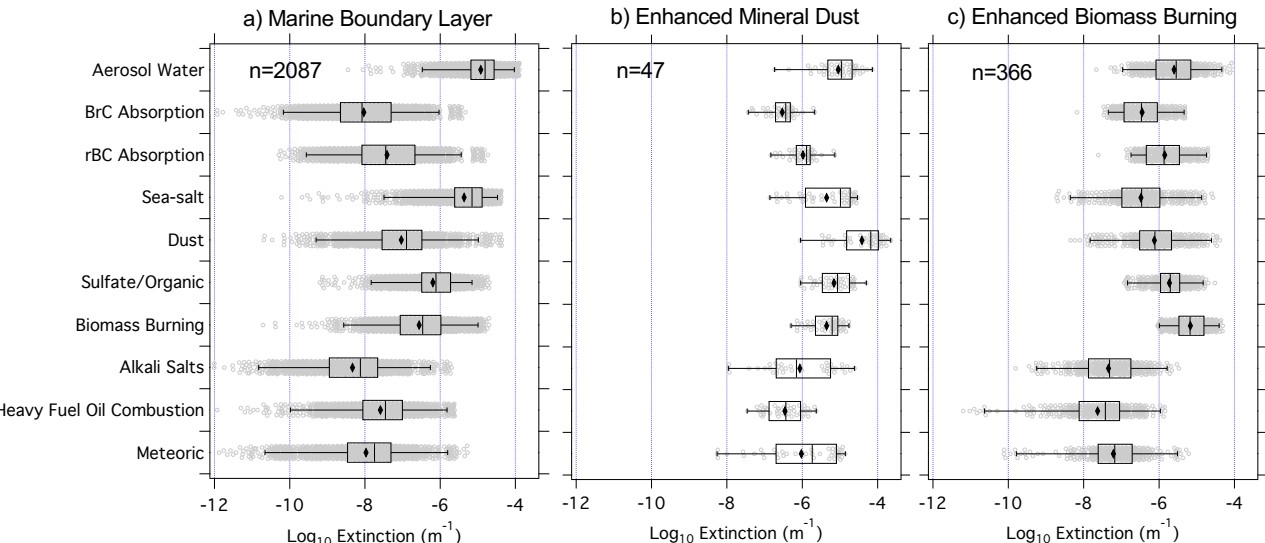

**Figure 10. Contributions of different aerosol components to logarithm of aerosol extinction (a) in the MBL, (b) in regions of enhanced mineral dust particle concentrations, and (c) in regions of enhanced biomass burning particle concentrations. Absorption only is shown for the BrC and rBC components. Each gray point is calculated from a single 60 s measurement. Boxes indicate the interquartile range, the central line represents the median, the diamond symbol the mean of the logarithm, and the whiskers are at the 2nd and 98th percentiles. The number of 60 s data points in each airmass type is indicated.**

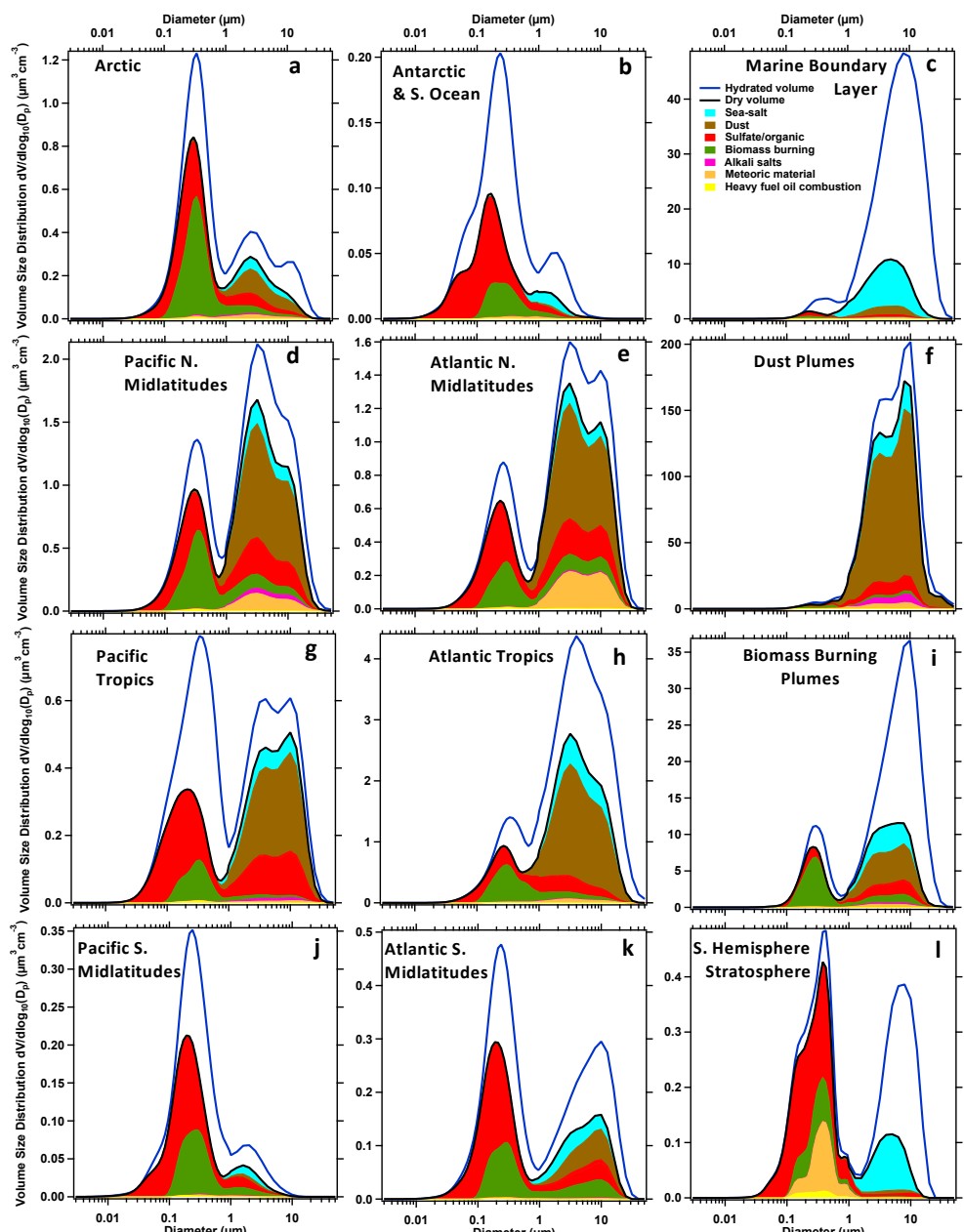

**Figure 11. Volume of particles of different aerosol types as a function of diameter, averaged over all data in different regions and air mass types across all of the ATom deployments. Note that scales on the y-axes vary. Descriptions of the regions are given in Fig. 7 and Table S1. Regional data (left two columns) are from the FT only and exclude data from BB and dust plumes and stratospheric intrusions. Size distributions from the MBL, stratosphere, and BB and dust plumes (right column) are not separated by ocean basin or latitude range. One pass of a binomial smoothing filter (Marchand and Marmet, 1983) has been applied to the data; PALMS particle types shown below 0.14 μm are extrapolated for smoothness.**

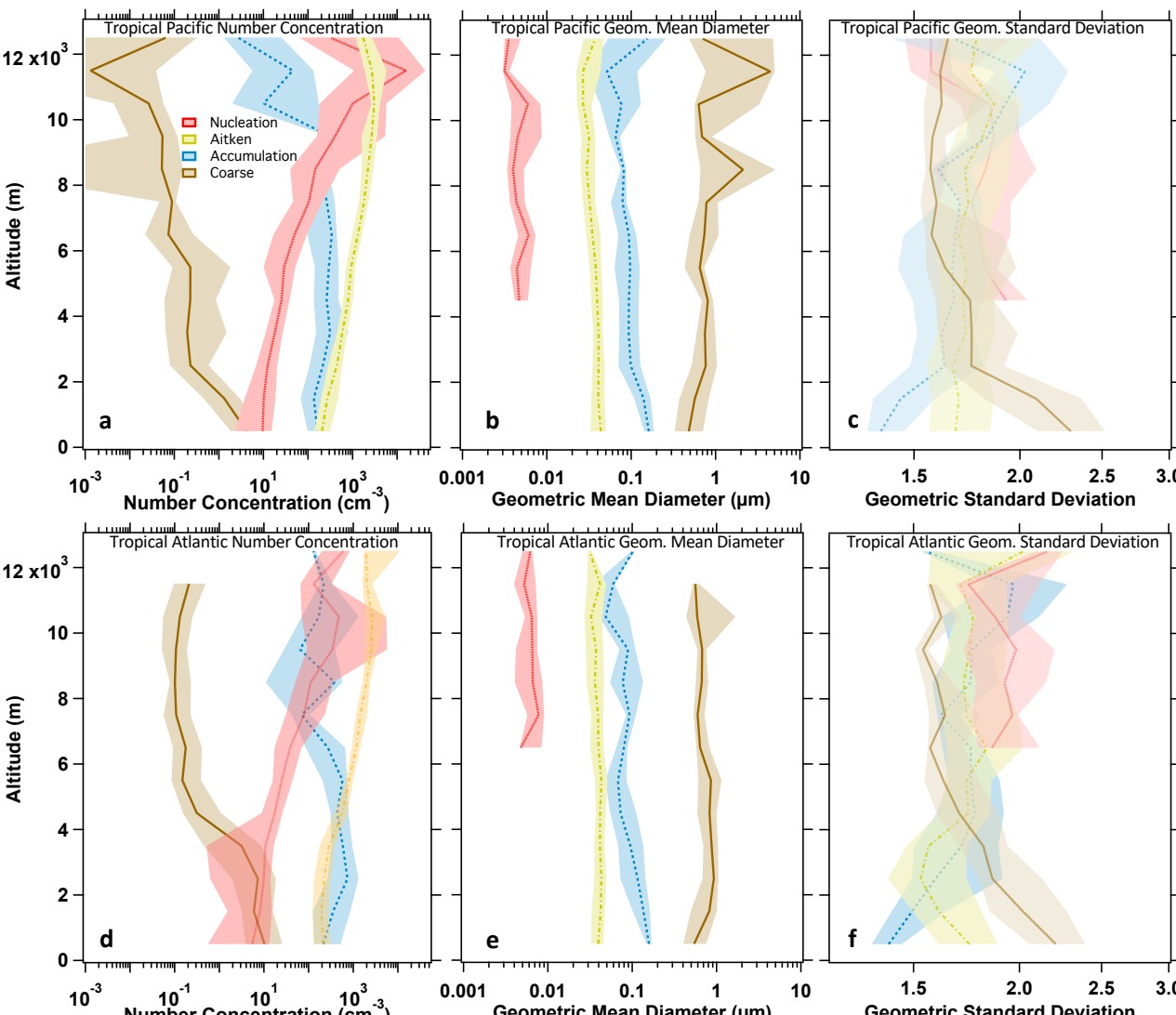

Figure 12. Vertical profiles of fitted lognormal parameters for the nucleation, Aitken, accumulation and coarse modes for the Pacific tropics (a, b, and c) and the Atlantic tropics (d, e, and f) for the entire ATom project. Lines are median values and shaded regions show the interquartile range. Number concentrations for the nucleation mode extend to lower altitudes than do the geometric mean diameter and standard deviation because samples with very low or zero concentrations could not be fitted, yet still provide valid concentration data that should be averaged. Similar vertical profiles for other regions sampled during ATom are in Supplemental Materials Figs. S1-2. The number of 60 s samples contributing to the values in each altitude bin are provided in Supplemental Fig. S7.

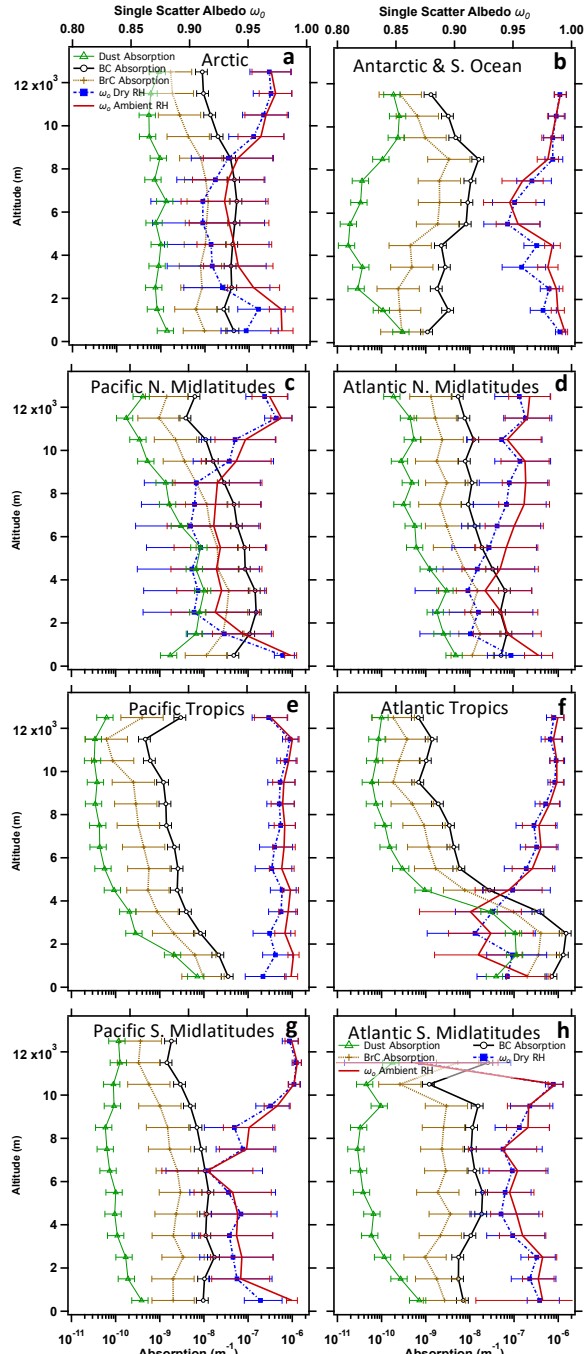

**Figure 13. Median vertical profiles of absorption at 532 nm wavelength from rBC, BrC, and dust (bottom axis) and single scatter albedo $\omega_0$ at 532 nm wavelength dry and ambient RH conditions (top axis) for different regions sampled during the entire ATom project. a) Arctic. b) Antarctic and Southern Ocean. c) Pacific northern midlatitudes. d) Atlantic northern midlatitudes. e) Pacific tropics. f) Atlantic tropics. g) Pacific southern midlatitudes. h) Atlantic southern midlatitudes.**

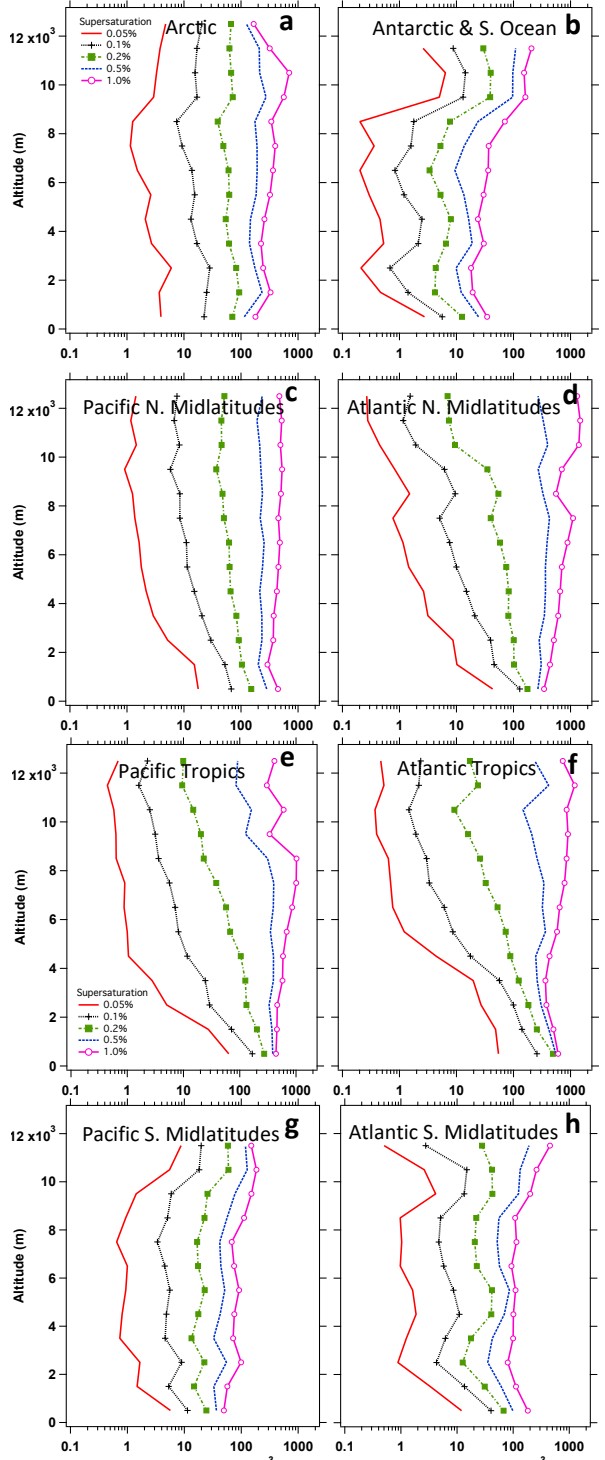

**Figure 14. Median vertical profiles of calculated CCN concentration at STP for supersaturations of 0.05, 0.1, 0.2, 0.5, and 1% for different regions sampled during the entire ATom project. a) Arctic. b) Antarctic and Southern Ocean. c) Pacific northern midlatitudes. d) Atlantic northern midlatitudes. e) Pacific tropics. f) Atlantic tropics. g) Pacific southern midlatitudes. h) Atlantic southern midlatitudes.**

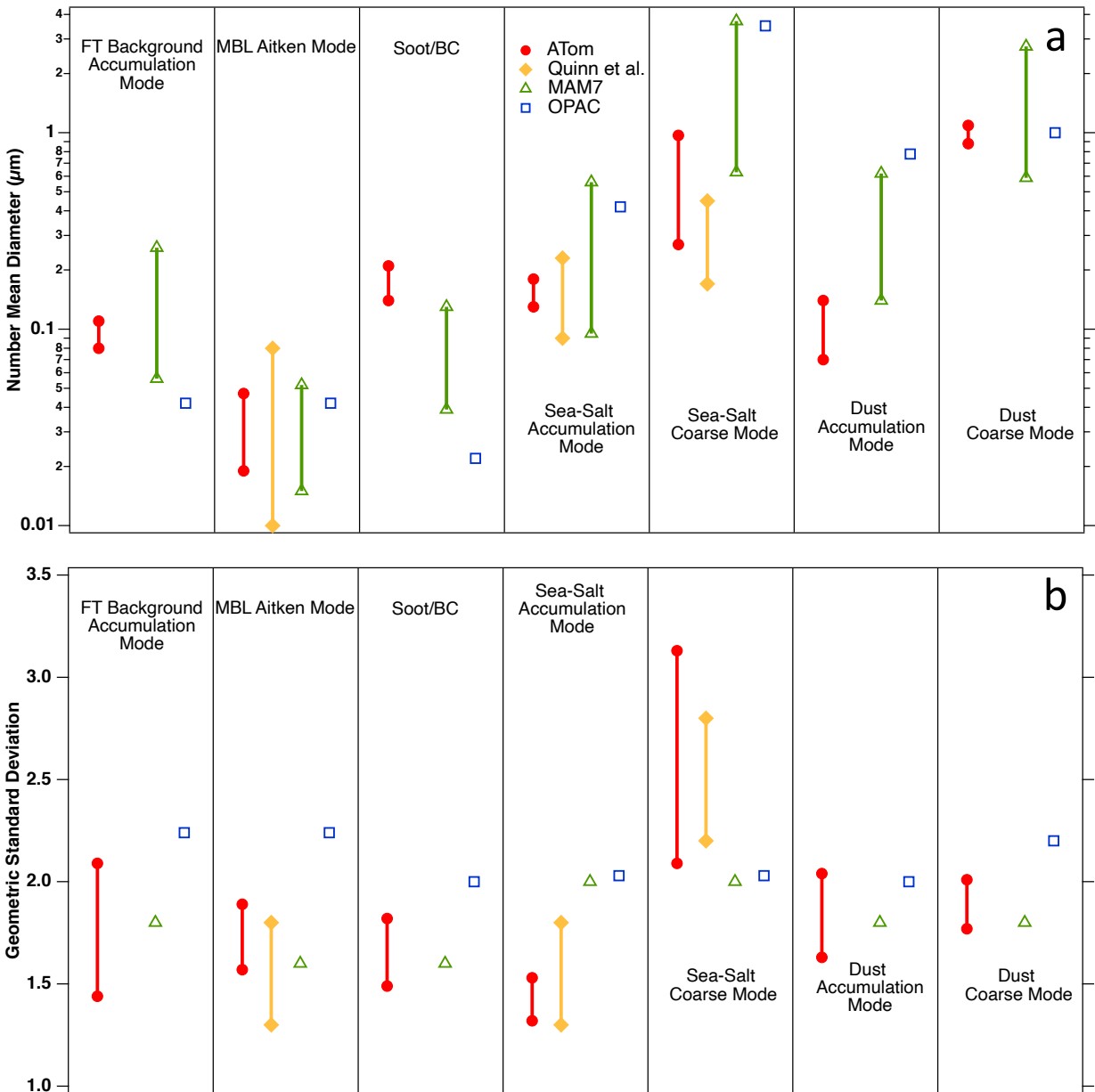

**Figure 15. Lognormal parameters for different aerosol types from fits to the ATom dataset (showing interquartile range), from fits to shipboard size distribution measurements on the remote oceans (Quinn et al., 2017, showing full range), from the MAM7 modal aerosol model (Liu et al., 2012; 2016, showing interdecile range), and from the OPAC parameterization (Hess et al., 1998). a) Number geometric mean diameter; b) geometric standard deviation. The MAM7 parameterization provides a single fixed value of geometric standard deviation but a range of diameters for each aerosol type.**

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
