# Peer review of "Ambient aerosol properties in the remote atmosphere from globalscale in-situ measurements"

_Atmospheric Chemistry and Physics, 2021_

## Community Comment (CC2)

Some responses from A. Clarke to "Reply to Comments" from C. Brock et al..

I appreciate the authors consideration of my earlier comments and indications of changes to be made in the final version. I will just clarify a few points further here with reference to original line numbers identified. New comments in ***Bold Italic*** follow earlier comment (**Bold**) and authors response (*italics*).

**L98   Please specify that "….an inlet.." is actually the " shrouded solid diffusor inlet designed by Clarke (University of Hawaii) and evaluated by McNaughton et al., 2007."**

*We will make this change. The origin of the inlet and sampling using it is described in more detail in Brock et al. (2019), which focuses on the aerosol sampling methodology*

   ***OK.  As an inlet affects all the aerosol data collected I think it needs to be clearly identified in a paper of this scope.  Moreover, it should be noted that this inlet must be operated under isokinetic conditions as failure to do so will particularly impact the relative collection of aerosol components associated with the larger sizes.***
* * *
**L220  The treatment of all components as externally mixed sizes would benefit from additional discussion of when this may or may not be a representative approach.**

*We will amplify this point, but in the interest of space will largely direct the reader to Froyd et al. (2019), who provide both the methodology and an examination of the mixing state of the aerosol during ATom. The PALMS data show that there is always an external mixture present; for example, most of the coarse mode particles are always different in composition than the accumulation mode. There are particles with dust in them and particles with no dust; the same with sea salt, biomass burning, BC-containing particles, etc. There are many internally mixed organic/sulfate particles as well, but they are present externally from particles of these other types.*

   ***Yes, I think all these points are well understood. However, perhaps some clarification is needed as Froyd (section 3.2) states "***The most abundant classes under most tropospheric environments are the sulfate–organic–nitrate internal mixtures and biomass burning (BB) particles…….." ***and goes on to say*** "All particle types acquire secondary material such as sulfate, ammonium, organics, and nitrate during atmospheric transport and aging. This secondary accumulation does not change particle assignments, except that heavy coatings may partially obscure unique signatures, resulting in a particle classified as "Other". For example, a mineral dust particle that contains secondary sulfate, nitrate, and organic material

will still be classified as mineral dust, and the derived dust mass includes the secondary material……………

**Such mixtures of aerosol components can alter the optical, humidification response, nucleating ability etc. relative to treating components as being in external mixtures.**
* * *
**L522    Far more robust comparisons with ambient extinction and AOD exist in the literature.  Given the numerous and sometimes subtle considerations (Fig 2) for calculated extinction discussed here, I do not see how the agreement or lack thereof  in Fig. 6  actually "……..indicates the methodology to calculate ambient aerosol optical properties is sound."  It may be sound but better agreements with simpler assumptions exist.  This data set is not designed to get AOD closure or even challenge many sources of uncertainty.  One worthwhile objective would be to determine what are the most important measurements needed to characterize AOD within a specified uncertainty.  Or how well do we need to know all properties to  reduce uncertainties to an acceptable level. Assessing the global role of intensive aerosol properties measured would appear better suited to the ATom measurement strategy.**

*We agree that there are much better ways to perform AOD closure; that was not a goal of the ATom measurements. We believe that our comparisons with the AERONET observations have value nonetheless. Certainly if there were no correlation between the AOD calculated from the ATom slantwise profiles and the AOD measured by "nearby" AERONET sites, this would be a major cause for concern. Only when we calculate the aerosol hygroscopicity and add the coarse mode measurements from the underwing probe does the AOD from the profiles show consistency with the AERONET observations. Again, the purpose of the AOD comparison is to demonstrate that we've properly accounted for the key features of the aerosol that contribute to ambient extinction. We certainly agree that the ATom dataset provides measurements that can be used to evaluate the sensitivity of climate to aerosol properties; that is the intent of providing this dataset for broader use by the community. We are currently working with modeling and remote sensing groups to diagnose discrepancies between remote sensing and in situ measurements at low AOD values (see below) and to evaluate assumptions underlying retrievals and models.*

**The main purpose is stated above …**"*Again, the purpose of the AOD comparison is to demonstrate that we've properly accounted for the key features of the aerosol that contribute to ambient extinction.*"
**Simply getting "ballpark" agreement of AOD with Aeronet in higher AOD cases does not indicate what has been properly accounted for.**
**What are the "key features" referred to and how are they "properly accounted for"?**

*It would be helpful to take a profile and show the contributions of the various measured aerosol components (and associated water) to the calculated aircraft AOD so the reader has a sense of what is being "properly accounted" for and which data is most important.*
*Does the calculated spectral AOD agree with Aeronet spectral AOD? – a key feature that is not extensive.*
*Regarding low AOD comparisons, it is probably worth noting that AOD contribution from above aircraft max altitude can often be 0.01 (or more).*
* * *
**L800-803 "To our knowledge this is the first….."----- This claim is not correct! …Etc.**

**The authors include the following in their response.**
*This dataset gives constraints for global models that have not previously existed. For example, we provide a size distribution for dust particles that can be directly compared with that carried in models (in most models dust mass is predicted and the size distribution prescribed). We provide an estimate of the contribution of these dust particles to ambient extinction and AOD; again, this can be compared directly with models. We do the same for sea salt, biomass burning particles, sulfate/organic mixtures, and even meteoric particles of stratospheric origin. No other data set does this; the comprehensive, self-consistent description of the size-and-composition-resolved aerosol properties in ATom is absolutely unique.*

**I believe these assertions still appear to be an overstatement.**
**For example, the size distributions of dust and black carbon are similar to those described previously for use by modelers along with their optical properties and mixing (***See Clarke et al., Size distributions and mixtures of dust and black carbon aerosol in Asian outflow: Physio-chemistry and optical properties. JGR, 109, 2004 and references therein)*

**Also, sea salt size distributions and size-resolved fluxes have been characterized for modelers in Clarke et al.** *"An ultrafine sea-salt flux from breaking waves: Implications for cloud condensation nuclei in the remote marine atmosphere [https://doi.org/10.1029/2005JD006565]* **and references therein as well as other papers by various authors.**

*Internal mixing is also common and well documented for most species with varying contributions as a function of size. The Froyd 2019 paper discusses a lot of assocated features of the data. However, some direct comparisons that reveal the differences and or improvements afforded by the ATom data sets over previous data should be presented that support the claims above. A proper comparison and assessment would probably require a separate paper focused on this topic.*

---

## Author Response (AR1)

Response to Reviewer Comments

Reviewer #1

The authors thank the reviewer for the positive overview response to the manuscript and for the extremely thorough and constructive comments, which we take as an encouraging sign of interest. We agree that getting the details right matters for this important dataset, and have attempted to address almost all of the reviewer's concerns and comments, which have substantively improved the manuscript. The reviewer's comments are in **bold**; our responses are in *italics*.

**The short assessment of this study is: impressive spatial coverage, comprehensive and state of the art set of experimental methods, and appropriate approaches to merge and integrate these. All this leads to a great data set on chemical, physical and optical properties of atmospheric aerosols. Definitely, suitable and important to be published in this journal.**

**The length of this review is by no means in contradiction with above positive assessment. By contrast, the expectation that this data set is going to be a benchmark that will most likely be used for a long time and in many future studies. Therefore, I consider it valuable to clarify several items and maybe adjust one or two assumptions in the calculations. It is one of several strengths of this manuscript that the entire chain of assumptions and calculations required to in infer optical aerosol properties from primary observations is quite completely and transparently presented. Addressing at least some of the comments below, which are virtually all of minor or even technical nature, could help in putting additional emphasis on the basic starting point assumptions, and to clarify a few things. There may be one or two items where I question an assumption (e.g. BC refractive index or "F_org"; see specific comments). Even if adjusting would likely lead to little change only, I encourage to consider it all the same. There is considerable chance that any assumption made in this study will be pickup up in many follow-up studies, given the quality of this work, also for calculations where it might have more impact than in this work. Keeping this in mind should help in gauging the effort put in addressing or discarding below comments.**

**More general and more important comments**

**Sect. 2.3, L262-280: As far as I can judge, the approach for approximating composition resolved size distributions to handle light scattering and light absorption is well thought. However, the underlying basic motivation / physics behind the approach could be communicated more clearly to the readers less experienced in aerosol optics. I suggest to introduce the distinct concepts behind handling scattering and absorption, possibly before dwelling on where composition and mixing state data are taken from (see also comment below, which addresses visualization in Fig. 2). For example, when it comes to light scattering, there is no way around describing aerosol size distribution and hygroscopic growth with sufficient accuracy, whereas replacing the volume associated with BC by the same volume of "NR-PM" as measured by the AMS introduces very limited error (BTW: I do not think that the approach chosen for this study leads to "double counting" of BC (line 275); instead it simply is material substitution for the light scattering calculations plus some volume error from sizing**

errors associated with BC particles). By contrast, first order approximation for calculating absorption simply is: summation over all absorbing components of the product "component specific mass absorption cross section times mass concentration" (with 2nd order corrections for size mixing state effects which lead to deviation from volume-based absorption and hamper additivity of light absorption for internal mixtures), while volume/size of externally mixed non-absorbing particles is absolutely irrelevant.

Section 2.3: The approach to obtain a good approximation of size-resolved aerosol composition and mixing state appears to be appropriate, though certain aspects are not perfectly clear. Given that different methods are combined in different manner for different size bins and particle types (in order to account for size and composition dependent detection efficiency of each method), I suggest to base the discussion on Figure 3 and to start with the basic assumptions before providing details. E.g.:

- Purpose: Prepare the ground for calculating RH dependence of size distribution and aerosol light scattering (as a function of RH) → step1: assign approximate mixing state and composition to each size bin in order to infer hygroscopic growth factors and refractive index (and density) in next steps. (If I understand correctly, this composition information is not used for inferring light absorption (otherwise, it would be inappropriate to substitute e.g. EC with different species).

- All size ranges: aerosol volume taken from AMP

- 0.05-0.14 um: Internal mixture. Composition exclusively based on AMS measurement. This means that the aerosol volume associated with refractory components that remain undetected by the AMS (BC, dust, NaCl, …) is substituted with AMS measured bulk composition.

- The three size bins in 0.25 to 4 um range: Measured volume is split into contributions by 9 particle types based on PALMS particle type classification. Then explain for each particle class how composition of respective volume is approximated, only including approximations common to calculating kappa and refractive index (here an SI figure or SI table resolved by particle type may be very useful and much clearer than a linear text block).

- 0.14 to 0.25 um size bin: measured volume is split to contributions from two sub-groups of PALMS-derived particle types. One sub-group is treated as internal mixture with AMS composition imposed, the other sub-group is retained as external mixture with PALMS-derived particle type specific composition imposed.

- extrapolation for small and large particles…

*It's challenging to describe all the steps taken to combine these disparate measurements in a way that is easily understandable, but we agree with the reviewer that walking the reader through the overall process, as represented by Figs. 2 and 3, would improve understanding of*

*the processes involved. To that end, we have modified Section 2.3 by adding a new section, as follows:*

[revised manuscript text omitted]

**Inferring kappa:**

**- Equation 4: this is an explicit variant of the ZSR mixing rule, as applied for inferring the kappa of some PALMS-derived particle classes. It is a very basic implementation which parametrizes kappa based on "inorganic to organic" ratio. Such a simplification can perform very well, given that the major contributors to inorganic volume are measured and lumped together, and alike for all major contributors to organic volume. However, footnote "B" in Table 2 suggests that sulfate is the only species considered for calculating F_org. This will lead to systematic bias if other inorganic ions such as nitrate make a substantial contribution to inorganic volume. It remains unclear how/whether nitrate volume is appropriately accounted for in size classes relying on PALMS composition data. Please clarify.**

*The PALMS instrument does not quantify the mass fraction of nitrate in the individual particles it analyzes, although it does for organic and sulfate components, from which F_org is determined. We do ignore the potential role of nitrate when calculating the hygroscopicity and the refractive index for particles with diameters >0.25 µm. For submicron sizes, the median AMS nitrate mass fraction in particles with diameters <0.25 µm was 2.4%, with 25th and 75th percentiles of 0.9% and 4.6%, respectively, when total AMS concentrations were positive. For more information on aerosol composition derived from the AMS measurements, see Nault et al. (2021). For particles <0.25 µm, nitrate is accounted for when calculating hygroscopicity and refractive index, but is quite a small contributor to these properties. For the coarse mode, nitrate concentrations are available from the filter measurements (minus the small amount of submicron AMS nitrate); there was ~0.1-1 µg m$^{-3}$ of nitrate in these measurements, which appears to be in the form of NaNO3 and Ca(NO3)2. The contributions are quite minor except in dust and BB plumes and in the MBL. Given the relative crudeness with which coarse-mode optical and hygroscopic properties are estimated, we don't feel that it's worthwhile attempting to vary the properties of the coarse mode refractory particle types to accommodate variations in coarse-mode nitrate abundance determined from relatively infrequent filter samples.*

Nault, B. A., Campuzano-Jost, P., Day, D. A., Jo, D. S., Schroder, J. C., Allen, H. M., Bahreini, R., Bian, H., Blake, D. R., Chin, M., Clegg, S. L., Colarco, P. R., Crounse, J. D., Cubison, M. J., DeCarlo, P. F., Dibb, J. E., Diskin, G. S., Hodzic, A., Hu, W., Katich, J. M., Kim, M. J., Kodros, J. K., Kupc, A., Lopez-Hilfiker, F. D., Marais, E. A., Middlebrook, A. M., Neuman, J. A., Nowak, J. B., Palm, B. B., Paulot, F., Pierce, J. R., Schill, G. P., Scheuer, E., Thornton, J. A., Tsigaridis, K., Wennberg, P. O., Williamson, C. J., and Jimenez, J. L.: Models underestimate the increase of acidity with remoteness biasing radiative impact calculations, Commun. Earth Environ. 2, 93, https://doi.org/10.1038/s43247-021-00164-0, 2021.

- The hygroscopic growth has discontinuities at size bin boundaries, particularly where switching from AMS to PALMS for composition constraints. Does this cause any problems with wet size distribution shapes, or is this unimportant because discontinuity is small or because final optical parameters are integrated over all sizes?

*These are small effects. We occasionally see small discontinuities, which are hard to distinguish from OPC response, but don't believe they have any substantive impact on any derived properties.*

- Minor: Sulfuric acid or nitric acid contain considerable residual water at "dry RH" (in cabin conditions), such that the effective kappa value would be considerably smaller (more comparable to e.g. corresponding ammonium salts). Anyway, volume fractions of these acids are likely low, such that propagated uncertainties are unimportant.

*The ambient aerosol in the background FT was actually quite acidic, with sulfuric acid often present (see Nault et al., 2021). Generally the RH during sampling was <8% in the FT due to warming of the sample stream in addition to deliberate drying. Resulting volume errors are <5%. Sulfuric acid is also important in the lower stratosphere, where conditions are extremely dry and residual water, though always present for sulfuric acid, is at a minimum.*

**Inferring refractive index (Sect. 2.7 and Table 2):**

- The equation composition dependence applied to some PALMS particle classes (1-F_org) * 1.479 + F_org * 1.480 + 0i appears to be a precision overkill given considerably larger absolute uncertainties. Furthermore, is it important to consider nitrate salts, which does not appear to be the case, for refractive index estimates (see related comment on hygroscopic growth)?

*As mentioned above, nitrate was generally a very small component of the aerosol mass as measured by the AMS, but would be accounted for in calculating the hygroscopicity of particles smaller than 0.25 μm. The reviewer is correct that the formulation for PALMS compositional dependence of refractive index is excessively precise, but it is just a product of using the "(1-F_org)" methodology and applying generic ammonium bisulfate (known pretty precisely) and organic (which could be a fairly broad range of values) refractive index values to the organic and inorganic fractions. We're now applying a more acidic assumption to the inorganic fraction*

*(based on AMS observations), and will keep this format so the reader can see how we calculated this refractive index, but are using a value of 1.44 rather than 1.479.*

**- "SP2: Black Carbon" and "SP2: Coating": please clarify whether these refractive indices feed into general calculation of scattering coefficient and/or absorption coefficient, or whether they are exclusively used for inferring BC particle mixing state from SP2 raw data. The Moteki 2010 value does not appear to be appropriate for absorption calculations and questionable for general applicability to light scattering calculations because it is only based on a single light scattering cross section measurement at 1'064 nm for BC heated to sublimation temperature by a strong laser ("single" in the sense of one parameter rather than single data point).**

*These entries in Table 2 are used only to calculate the MAC values that are then applied to the SP2 mass concentrations to get an absorption coefficient. This absorption coefficient is then added to the scattering coefficient from the non-absorbing species and to absorption from BrC (and now dust, in the revised manuscript) to get total extinction. Regarding the refractive index of rBC, there is a wide range of values from BC and BC surrogates in the literature (e.g., Bond et al., 2013). There are two reasons we wish to stick with the Moteki et al. values. First, using similar values has produced agreement within 20% between SP2-calculated (using core-shell) and measured (with a PSAP) absorption in both biomass burning and urban plumes (Schwarz et al., 2008). Second, this value and core-shell Mie theory produces a visible absorption coefficient that agrees very well with the independent observations made by photoacoustic spectroscopy during the fourth ATom deployment, admittedly limited to plumes containing significant levels of absorption (Fig. 4). The seminal Bond et al. (2013) review suggests a mid-visible value of 1.95+0.79i, but give unqualified mention of the Moteki et al. value with no clearly stated preference between the values. Given the relative lack of observations of well-aged soot particle refractive indices in the literature, but most importantly the agreement between calculated and measured absorption during ATom-4, we prefer to stay with the stated refractive index for rBC.*

*Schwarz, J., Gao, R., Spackman, J.R., Watts, Laurence, Thomson, D.S., Fahey, D., Ryerson, T., Peischl, Jeff, Holloway, John, Trainer, M., Frost, Gregory, Baynard, T., Lack, Daniel, de Gouw, Joost, Warneke, Carsten, and Del Negro, Lori: Measurement of the mixing state, mass, and optical size of individual black carbon particles in urban and biomass burning emissions. Geophysical Research Letters, 35, L13810, https://doi.org/10.1029/2008GL033968, 2008.*

*Bond, T. C., et al. (2013), Bounding the role of black carbon in the climate system: A scientific assessment, J. Geophys. Res. Atmos., 118, 5380– 5552, doi:10.1002/jgrd.50171.*

**Line 354ff: Treatment of e.g. the "PALMS-derived sulfate/organic" particles with respect to optical calculations remains somewhat unclear. The equation in the "refractive index" column of Table 2 is a simplified two-component volume mixing rule (only distinguishing "inorganics" and "organics" with refractive indices of ammonium bisulfate and OA assigned, respectively). This brings back the question: are inorganic salts other than sulfate salts considered for calculating "F_org"?**

*No, no additional inorganic salts are considered for particles >0.25 μm; instead this simple "organic or sulfate" framework is used.*

**Furthermore, is this 2-component mixing rule applied to big and small sulfate/organic particles (and equally treated types) or is the full AMS-composition considered for the small ones as implied by the compositional model?**

*The AMS composition, speciated by the Zaveri et al. inorganic composition model, is used for all refractory particles smaller than 0.25 μm. This is now clarified in the text.*

**Besides clarifications, I suggest some reordering and rewording along the line (depending how calculations were done actually): "Scattering was calculated for the wavelengths of 340, 380, 405, 440, 532, 550, 670, 870, 940, and 1020 nm, which match common wavelengths for the AERONET sunphotometers and satellite measurements of AOD. The refractive indices in Table 2 are not adjusted for wavelength; this is a small potential bias in the context of other assumptions and approximations in the calculation. All particle types were treated as spherical in shape and internally homogeneous for optical calculations. For particles that are a multi-component mixture based on the simplified composition and mixing state representation introduced in Sect. 2.3, the dry particle refractive index is calculated as the volume-weighted mean refractive index of contributing components. This calculation is further simplified for this and that particle type using this and that equation/approach….". – Note: this volume-based mixing rule is also applied to the small particles for which composition is constrained with AMS only (if I understand correctly; this is not clearly stated in the manuscript).**

*Agreed this could be clearer. The manuscript wording has been changed along the lines the reviewer suggests, starting at line 260 in the revised manuscript:*

*"Scattering was calculated for the wavelengths of 340, 380, 405, 440, 532, 550, 670, 870, 940, and 1020 nm, which match common wavelengths for the AERONET sunphotometers and satellite measurements of AOD. The refractive indices in Table 2 are not adjusted for wavelength; this is a small potential bias in the context of other assumptions and approximations in the calculation. All particle types are treated as purely scattering, spherical in shape and internally homogeneous for optical calculations, with the exception of the absorbing components rBC, BrC, and mineral dust, which are described in Sect. 2.7.2. Non-refractory particles with Dp<0.25 μm, and all particles with Dp<0.14 μm, are treated as fully mixed, multi-component mixtures based on the AMS-derived composition and the ZSR mixing state representation introduced in Sect. 2.3.1. The dry particle refractive index is calculated as the volume-weighted mean refractive index of contributing components. This calculation is further simplified for non-refractory particles with Dp>0.25 μm using just the PALMS organic and sulfate mass fractions (Froyd et al., 2019), and applying organic and ammonium sulfate real refractive indices of ~1.48 to both of these components. Total scattering is the sum of the scattering from the individual composition-based size distributions.*

To calculate scattering coefficient of the aerosol at ambient RH, the effects of hygroscopic growth were considered. The diameter of every particle was adjusted based on growth factors for that aerosol type calculated as described in Sect. 2.5, and the refractive index was adjusted to the volume weighted mean of dry particle and water refractive indices. Scattering was calculated for the wavelengths of 340, 380, 405, 440, 532, 550, 670, 870, 940, and 1020 nm, which match common wavelengths for the AERONET sunphotometers and satellite measurements of AOD. The refractive indices in Table 2 are not adjusted for wavelength; this is a small potential bias in the context of other assumptions and approximations in the calculation. The refractive indices for the sulfate/organic, biomass burning, meteoritic, and EC compositional classes are calculated from the volume-weighted components from the inorganic compositional model and the organic mass. Additionally, the refractive indices for the hydrated ambient aerosol are adjusted on a volume-weighted basis for the water content. Total scattering is the sum of the scattering from the individual composition-based size distributions."*

**Suggestions for some additions/reorganization of Figure 2:**

*Most of the proposed edits (i-vi below) focus on adding more specific detail on the quite complex process of combining data from 8 instruments. In fact, the purpose of Fig. 2 is to provide a schematic, high-level overview so that the reader is not overwhelmed by the detail involved in this. The more complex steps would not be adequately described in a schematic diagram, yet would add a lot of clutter that would make it much less readable. So we prefer to keep the schematic largely as is, and will rely on the interested reader to probe in more detail into the more complete descriptions in the various sub-sections of Sect. 2.*

**i) Explicitly indicate which subsets of the flow chart are used to compute light scattering and light absorption, respectively. Maybe even split in two separate panels as inputs hardly have any overlap.**

**ii) How is contribution of BC particles to asymmetry parameter handled? Is BC particle contribution completely ignored? Is this expected to have a significant effect? If yes, rather systematic positive or negative bias (depending on value of calculated asymmetry parameter)?**

*Correct; the contribution of BC particles to the asymmetry parameter is ignored. As BC particles represent a few percent of particle number in the 90-550 nm size range, this is a small effect. This is now explicitly stated in the revised manuscript in Sect. 2.7.2.*

**iii) What optical model is used for water-soluble brown carbon absorption? (See also separate comment.)**

*None (see Sect. 2.7.2). Absorption from BrC is just parameterized from direct measurements.*

**iv) How is hygroscopic growth effect on BrC and BC absorption treated? (RH also as input for absorption calculations? Explicitly include in the figure that treated independent of RH.**

We now clearly state the hygroscopic growth effects on BrC and rBC absorption is ignored.

**v) How is dust absorption treated? Indicated even if set to zero, as this would also be an important piece of information.**

*We have now calculated light absorption due to mineral dust. Mineral dust is given a real refractive index of 1.55 (which can vary with water uptake), with a wavelength-dependent imaginary component of 0.002i at 530 nm, scaled to other wavelengths with an Angstrom exponent of 3 based on Saharan dust measurements by Weinzierl et al. (2011).*

*Weinzierl, B., Sauer, D., Esselborn, M., Petzold, A., Veira, A., Rose, M., Mund, S., Wirth, M., Ansmann, A., Tesche, M., Gross, S. and Freudenthaler, V., Microphysical and optical properties of dust and tropical biomass burning aerosol layers in the Cape Verde region—an overview of the airborne in situ and lidar measurements during SAMUM-2, Tellus B, 63: 589-618, https://doi.org/10.1111/j.1600-0889.2011.00566.x, 2011.*

**vi) Top right box: Why/how is pressure required? "H2O" likely stands for "water vapour partial pressure (as opposed to total liquid water content or liquid water associated with non-activated aerosol particles; see e.g. "H2O" label in Fig. 10). I assume that temperature is only required to infer RH from water vapour partial pressure, whereas nothing else is treated as temperature dependent? It might be worthwhile to emphasize the top right box is exclusively required to deliver RH to hygroscopic growth calculations, i.e. simplify it to "(ambient) RH".**

*Water vapor concentration (ppmv) was directly measured by open-path absorption. Pressure is needed to convert to mass mixing ratio and then RH. We wish to show which instruments provided input, and so explicitly show the water vapor and temperature measurements as separately measured components leading into the calculations.*

**Line 371: Any peculiar reason for using the idealistic core-shell morphology assumption as opposed to fractal-like shapes? Size dependence of MAC tends to be stronger for compact spheres than for loose compact spheres (e.g. Romshoo 2021). Additionally, the refractive index used for BC is inappropriate as it is a value to get light scattering by BC at 1064 nm and at 4'000 K within the SP2 instrument right, whereas it wasn't determined to get light absorption by BC right (which is not accessible to standard SP2 measurements). What matters in the end are the resulting coated BC MAC values (or the alternatively the product bare BC core MAC value times absorption enhancement factor due to lensing). Values resulting from the calculations made in this study should be reported and be put in context of previous values in the literature, even if it simply remains on the level of confirming plausibility of the result.**

*Because most of the ATom dataset involves measurements in the very remote troposphere, we feel that core/shell morphology, while simplistic, is the most authentic way we can simulate BC MAC values based on the measurements we have. We did not collect and examine particles with microscopy, and the SP2 (and the photoacoustic spectrometer on ATom-4) is the only tool–and a*

*powerful one–that we have available. Following long distant transport, others have noted that soot exhibits a compact morphology (e.g., China et al., 2014). Very recently, Wu et al. (2021) used core-shell Mie theory to obtain satisfactory agreement between measured and modeled MAC values at several wavelengths. Finally, our calculations using core/shell Mie theory agree very well with the photoacoustic measurements in absorbing plumes in ATom-4.*

*We agree that the MAC values should be placed in the context of existing literature, including Romshoo et al., Fierce et al. (2020), and Wu et al. (2021). We are adding the following text to the manuscript in Sect. 4.1.3:*

*"Because light absorption by BC is a key uncertainty in global estimates of the direct radiative effect, it is useful to evaluate the assumptions we make in its calculation from the SP2 observations. We have calculated absorption assuming that the measured rBC particles are well-aged and compact with a density of $1.8 \times 10^3$ kg m-3, and that core/shell Mie theory using the measured coating thickness, assumed to be a non-absorbing organic-sulfate mixture, provides a realistic approximation to their optical properties. Detailed consideration of different modeling approaches (Romshoo et al., 2021) suggests that core/shell Mie theory overestimates MAC values of coated BC by a factor of 1.1-1.5, with values increasing with increasing organic fraction (corresponding to coating thickness), for a fractal dimension for the BC core of 1.7, but with smaller discrepancies as fractal dimension increases toward 3 (a spherical core). Fierce et al. (2020) further report that core/shell Mie theory substantially overpredicts the absorption by BC in measurements in urban outflow, but that this discrepancy can be reduced by accounting for heterogeneity in particle composition and coating thickness. In contrast, Wu et al. (2021) found that core-shell Mie theory provided MAC values in agreement with, or even underestimating, directly measured MACs in aging biomass burning plumes downwind of West Africa. Zanatta et al. (2018) reported that core/shell Mie theory slightly underpredicted the measured MAC for aged, coated soot in the Arctic. China et al. (2015) found that aged soot particles measured at a mountaintop site in the Azores had a compact morphology with thin coatings, and that radiative forcing calculated using core/shell Mie theory was within 12% of that calculated using the discrete dipole approach.*

*Almost 30% by number of FT particles measured by PALMS during ATom were of BB origin (Schill et al., 2020). Thus we expect both compact core morphologies and a substantial non-absorbing coatings in the rBC particles associated with the BB particles; these characteristics are supported by the coating thicknesses measured by the SP2 instrument (Supplemental Table S6) and by the small values of water-soluble BrC absorption measured in filter extracts (Zeng et al., 2020). The MAC values for rBC we calculate from the ATom dataset using core/shell Mie theory were 14.4 ± 1.4 m2g-1 at a wavelength of 532 nm averaged over the free troposphere for all four ATom deployments (Supplemental Table S6). Values in identifiable BB plumes were 13.3 ± 0.4 m2g-1. These values are generally consistent with those measured at 514 nm in West African biomass burning plumes ranging from ~11.3 to ~14.2 m2g-1 for plume ages of ~1 to >9 hr, respectively (Wu et al., 2021). Thus the MAC values we calculate using core/shell Mie theory appear to be reasonable given the likely BB source of most of the rBC. Further, since no*

*observations of soot morphology were made during ATom, we lack a basis for any additional refinement in our estimate of MAC values for coated rBC.."*

*China, S., et al. (2015), Morphology and mixing state of aged soot particles at a remote marine free troposphere site: Implications for optical properties, Geophys. Res. Lett., 42, 1243–1250, https://doi.org/10.1002/2014GL062404.*

*Fierce, L., Onasch, T. B., Cappa, C. d., Mazzoleni, C., China, S., Bhandari, J., Davidovits, P., Fischer, D. A., Helgestad, T., Lambe, A. T., Sedlacek, A. J., Smith, G. D., and Wolff, L.: Radiative absorption enhancements by black carbon controlled by particle-to-particle heterogeneity in composition, Proc. Nat. Acad. Sci., 117, 5196-5203; https://doi.org/10.1073/pnas.1919723117, 2020.*

*Wu, H., Taylor, J. W., Langridge, J. M., Yu, C., Allan, J. D., Szpek, K., Cotterell, M. I., Williams, P. I., Flynn, M., Barker, P., Fox, C., Allen, G., Lee, J., and Coe, H.: Rapid transformation of ambient absorbing aerosols from West African biomass burning, Atmos. Chem. Phys., 21, 9417–9440, https://doi.org/10.5194/acp-21-9417-2021, 2021.*

*Zanatta, M., Laj, P., Gysel, M., Baltensperger, U., Vratolis, S., Eleftheriadis, K., Kondo, Y., Dubuisson, P., Winiarek, V., Kazadzis, S., Tunved, P., and Jacobi, H.-W.: Effects of mixing state on optical and radiative properties of black carbon in the European Arctic, Atmos. Chem. Phys., 18, 14037–14057, https://doi.org/10.5194/acp-18-14037-2018, 2018.*

**Romshoo, B., Müller, T., Pfeifer, S., Saturno, J., Nowak, A., Ciupek, K., Quincey, P., and Wiedensohler, A.: Radiative properties of coated black carbon aggregates: numerical simulations and radiative forcing estimates, Atmos. Chem. Phys. Discuss. [preprint], https://doi.org/10.5194/acp-2020-1290, in review, 2021.**

**Lines 383 to 397: BrC data are only available on 5 – 15 minutes time resolution, while the aerosol variability occurred on shorter time scales. Furthermore, a time resolution of 60s is chosen for the reported data set. Therefore, the BrC absorption data are "resampled" to higher time resolution with making use of covariance between BrC and BC and between BrC and biomass burning aerosol mass concentrations. While I fully support such an approach, I do not consider the actual implementation appropriate because it does not conserve the mean BrC mass measured on the original time intervals. I suggest to choose an approach in which covariance with one or both aforementioned parameters is used to introduce variation of BrC absorption around the measured mean value over the original sampling interval in such a manner that mean BrC absorption is conserved for each original interval.**

*This is an interesting idea, essentially scaling the parameterized BrC to match the mean of the observations around the filter sampling interval, then allowing the BrC to vary with the parameterization at shorter time intervals. Unfortunately, this is impractical because the filters were short in duration and relatively irregular, covering a small fraction of flight. In addition,*

*filters with cloud contamination were discarded, further limiting the dataset. We don't feel that the substantial effort of this adjustment would reduce uncertainties much below the already stated approximate factor of three.*

**Minor comments**

**Figure 3: This figure is useful. Some minor suggestion:**

**i) The inlet size range could additionally be indicated with arrow at top.**

*The figure has been modified as suggested.*

**ii) It might be worthwhile to indicate the size ranges across which AMS and PALMS provide composition information (currently only shown across which size range information of these instruments is used).**

*This topic is covered in considerable detail in Guo et al., AMT, 2021, and would make the current figure too complex, in our judgement.*

**iii) Maybe "AMS bulk composition" because no size-resolve data are used (in contrast to PALMS for which size-resolved information is retained).**

*Agreed, and the figure has been changed.*

**iv) For the 0.14 to 0.25 size class: "meteoric" appears twice, oil combustion appears to be missing?**

*Yes, thanks for catching this error! We also inconsistently used "meteoritic" and "meteoric" and have fixed that throughout the manuscript.*

**L57-59: Just a side remark: interaction between different molecules in gas and liquid condensed phase is of chemical and physical nature: Raoult's law shifts the phase partitioning for species that are miscible with a liquid aerosol phase present in the system without involving chemical reactions, i.e. also for non-reactive vapours. See volatility basis set approach for phase partitioning in e.g. Donahue et al. (2006).**

*Agreed; we have changed to "interacting chemically and physically".*

**Donahue, N. M., Robinson, A. L., Stanier, C. O., and Pandis, S. N.: Coupled partitioning, dilution, and chemical aging of semivolatile organics. Environ. Sci. Technol., 40, 2635-2643, doi:10.1021/es052297c, 2006.**

**L63: "dilution" could be added to this comprehensive list (as it can cause evaporation feedback for semi-volatile species through the physical effect mentioned in the previous**

comment). Distinction between solid and liquid condensed phase is also important in this context.

*Agreed, we added "dilution" and "mass, composition, and phase" to this section.*

**L86: Recent overview article on polarimetric retrievals:**

**Dubovik, O., Li, Z., Mishchenko, M. I., Tanré, D., Karol, Y., Bojkov, B., Cairns, B., Diner, D. J., Espinosa, W. R., Goloub, P., Gu, X., Hasekamp, O., Hong, J., Hou, W., Knobelspiesse, K. D., Landgraf, J., Li, L., Litvinov, P., Liu, Y., Lopatin, A., Marbach, T., Maring, H., Martins, V., Meijer, Y., Milinevsky, G., Mukai, S., Parol, F., Qiao, Y., Remer, L., Rietjens, J., Sano, I., Stammes, P., Stamnes, S., Sun, X., Tabary, P., Travis, L. D., Waquet, F., Xu, F., Yan, C., and Yin, D.: Polarimetric remote sensing of atmospheric aerosols: Instruments, methodologies, results, and perspectives. J. Quant. Spectrosc. Radiat. Transf., 224, 474-511, doi:10.1016/j.jqsrt.2018.11.024, 2019.**

*We have added a mention of polarimetric retrievals and cited Dubovik et al.*

**L164: Petzold et al. (2013) recommend using "rBC" when reporting BC mass quantified using laser-induced incandescence. For the purpose of this manuscript, it is probably more useful to stay with "BC mass", except for dropping a remark in the methods section that BC mass is obtained through measurement of operationally defined "rBC mass".**

*For consistency with Petzold et al.'s recommendations, we have changed "BC" to "rBC" throughout the text when referring to BC reported by the SP2 technique.*

**Petzold, A., Ogren, J. A., Fiebig, M., Laj, P., Li, S. M., Baltensperger, U., Holzer-Popp, T., Kinne, S., Pappalardo, G., Sugimoto, N., Wehrli, C., Wiedensohler, A., and Zhang, X. Y.: Recommendations for reporting "black carbon" measurements. Atmos. Chem. Phys., 13, 8365-8379, doi:10.5194/acp-13-8365-2013, 2013.**

**L167-168: "BC mass concentration data, corrected to reflect accumulation mode BC outside of the detection-range of the instrument" – How was this done? Extrapolation via lognormal fit? At large diameter tail only, or also for small BC cores? Is it important to state time resolution of the correction factor?**

*Extrapolation to smaller and larger sizes was performed on a 1s time basis using an average rBC size distribution calculated for each flight, eliminating time periods near takeoff and landing. That single scaling factor to account for undetected mass (almost all of which is smaller than detected, not larger) for each flight was applied to the 1s data for that flight. We now state this in the manuscript:*

*"This instrument uses laser-induced incandescence to measure the rBC mass within individual particles in the accumulation mode size range on a 1 s time basis (with frequent null detections*

*at this rate at the concentrations found in ATom). The rBC mass concentration data were corrected to reflect accumulation mode rBC particles outside of the detection range of the instrument by using a lognormal distribution fitted for the average rBC size distribution for each flight, eliminating time periods near takeoff and landing, to calculate a scaling factor. That single correction factor per flight, which increased rBC concentrations less than a factor of 1.1 (Katich et al., 2018), was applied to the 1s data for that particular flight. The rBC data were then averaged, with zeros, to the 60 s AMS sampling times, with an uncertainty of ~30%. Information on the size distribution of the rBC and on the thickness of non-refractory coatings on the rBC particles, which are used to calculate optical properties of the rBC, were obtained by accumulating data over longer time periods (Supplemental Materials Table S6)."*

**L166-168: "BC mass concentration data, […], are reported on a 1 s time basis (with frequent null detections at this rate at the concentrations found in ATom)" – All okay with this. However, important to provide good instructions in the meta data to users of the data base on how (not to) handle zero entries when aggregating 1s data to lower time resolution.**

*The values in the dataset represented by this manuscript are aggregated to 60s, so this is not an issue. We now describe this properly.*

**L182-187: only "cloud free" conditions are reported, whereas "haze" category is included. Is this going to be taken up in the discussion section when it comes to the question of comparing these in-situ data with remote sensing data? I.e., are remote sending data typically "cloud free only" inclusive or exclusive "haze"?**

*Different remote sensors have different cloud screening algorithms. We do not report data in identifiable clouds, but in the moist marine boundary layer, often at very high RH values, which are categorized as the "aerosol-cloud transition regime" by the instrument investigators for the cloud probes (U. of Vienna). We provide this information for data users wishing to compare or combine this dataset with the full ATom dataset that contains many more gas-phase, meteorological, and radiative parameters, so that they can exclude the same data when averaging. We will address the issue of satellite/in situ comparisons in a subsequent manuscript, Wang et al., in prep. See also the response to Antony Clarke's extensive comments on in situ/AERONET AOD comparisons in the online discussion.*

**L208: PALMS uses laser ablation to desorb and ionize, correct? Can all particle types be vaporized (i.e. also transparent particles)? Some hint is given on L213 that detection efficiency might be composition dependent (or was this only about size dependence), while no statement is made about resulting impacts on averaged composition in a size bin.**

*Yes, PALMS is a laser ablation/ionization mass spectrometer. Details of the PALMS detection method and quantification, and of the mapping of aerodynamic size to optical size are described exhaustively in Froyd et al., (2019). Briefly, the PALMS has a high detection efficiencies for the particle types we describe here, with the notable exception of "EC". There is a strong and time-varying size dependence to detection efficiency and the PALMS data rate is limited to ~4*

*particles/s; this is why the PALMS cannot provide quantitative component concentrations, and instead fractional abundances are mapped to the independently measured size distributions. For consistency, we then do the same with the AMS data, using it to more quantitatively describe the composition of the non-refractory component of the aerosol.*

**L218ff: "Thus, to calculate optical and hygroscopic properties, we do not assume a weighted internal mixture of the chemical components, but rather treat the total aerosol as an externally mixed collection of independent size distributions, each composed of one PALMS compositional type mapped onto the particle size distributions." – Important piece of information. Shouldn't this better be moved to where Figure 2 is discussed?**

*The new Section 2.3.1 should take care of this. We will leave this sentence here to remind readers of the methodology.*

**L225: "Further, the AMS composition is applied to the sulfate/organic, biomass burning, EC, and unclassified particle types for the 0.14–0.25 μm PALMS size range". – How is this to be understood? Are they all thrown into one bucket and composition is assigned based on AMS data (which does not exclude that composition of these aerosol types still differs after this process because they are prevalent at different times)? What happens to e.g. EC, which is not detected by the AMS, i.e. are the "EC particles ending up as "EC-free" (which may not necessarily be an issue)?**

*Yes, as clarified in the Section 2.3.1, all these types are thrown into the AMS "bucket", since the AMS is providing bulk compositional information on these particle types together. The EC particles are just treated as non-refractory (i.e., in the AMS "bucket") but separately tracked because a) they are a very small fraction of the total; b) we have the SP2 to provide more quantitative rBC measurements, and c) we still want to keep track of them in the event it's somehow useful in some future analysis.*

**L230: "As noted by Hodzic et al. (2020), in background conditions during ATom a substantial fraction of the AMS organic aerosol (OA) concentrations were below detection limit, and included negative values. We substitute negative AMS values with zeros when calculating optical or hygroscopic properties (Sect. 2.5)" – One should first pre-average the data to intervals sufficiently long to reduce negative organic readings to a minor fraction such that replacing negatives by zeros does not really make a change in the average composition inferred from these pre-averaged data. If replacing the negatives by zeros is done before any averaging, then this will introduce a systematic high bias in average organic fraction and bias calculated hygroscopic growth through its dependence on organic to inorganic ratio.**

*Yes, this is true. However, the aircraft was continually profiling at 500 m/min and we needed to make a tradeoff between vertical resolution and averaging of data. Indeed, for the PALMS, the statistics were sufficiently poor for larger particles at 60 s (or even 3 min) in the free troposphere that we resorted to regional average values of composition for the refractory particles in the FT (Fig. 3). For the AMS, a quick check can be made of the effect of substituting*

*negative values with zeros vs. including them: For all data with a positive AMS total concentration (a requirement to calculate hygroscopicity or refractive index; otherwise the data were discarded), the median ratio of OA to total AMS mass was 0.35 including negative values and 0.37 if negative values were substituted with zeros. We conclude that this does not have a substantial effect on parameters derived from the AMS composition.*

**L234 – L238: This descriptions sounds as if PALMS data were aggregated by "air mass or plume" type most of the time, with few exceptions only, when truly time-resolved composition is provided? Please clarify. I additionally suggest to refer to Fig. 3 where regional averaging is also addressed.**

*We now refer to Fig. 3 in this section. Yes, for most of the FT outside of discrete BB or dust plumes, we made use of regional averages for the two largest PALMS size classes. In the MBL there were always enough particles for the PALMS to provide good counting statistics. We now cite the fraction of time the raw data could be applied for each size class, from smallest to largest (number of 60s intervals with >5 particles per size class/total number of 60s intervals with valid PALMS data):*

*SD1: 17827/19921=0.89*

*SD2: 19263/19921=0.97*

*SD3: 7749/19921=0.39*

*SD4: 2103/19921=0.11*

*New lines 344-345: "For the four PALMS size ranges, from smallest to largest, the regionally averaged compositions were applied to 11%, 3%, 61%, and 89% of the 19,921 60-s samples, respectively. "*

**L263: "$D_p$" appears to be reserved for total particles geometric diameter throughout most of the manuscript. Here it is used for a BC mass equivalent core diameter, which differs from total particle diameter for internally mixed BC (in which case optical sizing by the SP2 provides something like $D_p$). I suggest using a distinct symbol for BC core mass equivalent diameter to avoid ambiguity. Strictly speaking, the SP2 arrow in Fig. 3 also lives on a different diameter axis.**

*Agreed, we will use the word "diameter" for the core mass equivalent diameter. Fig. 3 is very schematic in nature and we won't create a new diameter axis to minimize confusion for most readers.*

**L279: This should read: "for the purpose of calculating aerosol hygroscopic growth and light scattering" (I doubt that PALMS data go into BC light absorption calculations.)**

*No, but that information goes into asymmetry parameter, mass scattering efficiency, single scatter albedo, etc., which qualify generically as "optical properties".*

**Equations 3: This equation appears to be quite randomly picked among those equations used to come up with hygroscopic growth (most of which are provided in Table 2, except for the ZSR mixing rule approach applied to AMS data on top level, which is quite hidden on lines 307/308). The ZSR mixing rule deserves at least as much emphasis as Equation 3. (PS: Equation 4 is the explicit variant of ZSR mixing rule applied to some PALMS-derived particle classes).**

*The equation for calculating the kappa of OA was chosen based on experience from the SENEX/SEAC4RS campaigns in the southeastern U.S. (Brock et al., 2016a) where, in a heavily OA-dominated region, it successfully replicated the observed hygroscopicity of the aerosol. It also relies on the AMS O/C ratio, which is provided at the 60s resolution needed to construct vertical profiles. Other approaches are of course possible, and as noted the Nakao (2017) paper points out that volatility and solubility--information we don't have--are probably the key properties governing kappa for OA.*

*Nakao, S.: Why would apparent κ linearly change with O/C? Assessing the role of volatility, solubility, and surface activity of organic aerosols, Aerosol Sci. Technol., 51, 1377-1388,https://doi.org/10.1080/02786826.2017.1352082, 2017.*

**Line 318: "The overall project-mean value of κ from the AMS measurements was 0.53±0.19". This value appears to be at the high end of AMS-based literature data (given that potential sea salt contribution is not considered in this value). Is this a result of small organic fraction and/or high fraction of acids, which got a high kappa assigned in pure form (on average)?**

*The organic has a relatively high kappa, to account for its aged nature, and in the remote troposphere continual condensation of small quantities of acidic sulfate from SO2 oxidation in the absence of ammonia leads to the high average hygroscopicity value.*

**Line 323: "For a pure organic aerosol (Forg=1), this yields κorg=0.17, close to the AMS project-wide value of κorg=0.18 from Eq. 3"; good to put approaches chosen for different size ranges in context to each other. However, while consistency for pure organic aerosol is given (just skipping O/C dependence in PALMS size range), Equation 4 based kappa have a low bias compared with AMS-kappa for pure inorganic aerosol (F_org = 0), as the lowest among all sulfate and inorganic kappa values is chosen for kappa_inorg in Equation 4. Is this of relevance or anyway within uncertainties? (The potential bias in F_inorg, which I addressed in a previous comment, may be of greater impact.)**

*Note that there was an error in the entry in Table 2 (and the associated calculations), for the hygroscopicities of the PALMS non-refractory particle types. The kappa for ammonium sulfate had been entered, it should have been that of ammonium bisulfate (see refractive index column, for example). However, we agree that the value chosen for the kappa of the inorganic fraction*

*of the PALMS non-refractory particle types should be more consistent with that of the AMS. We have calculated the project-average kappa for the inorganic fraction from the AMS data; this value is 0.73, which lies between the ammonium bisulfate and sulfuric acid kappas. We have applied this value to the dataset and recalculated the hygroscopic growth and resultant optical properties. The text and Table 2 are modified appropriately (line 540):*

*"For a pure organic aerosol (Forg=1), this yields κorg =0.17, close to the AMS project-wide value of κorg=0.18 from Eq. 3. Using Eq. 4, the project-wide mean value of κ for non-refractory PALMS particle types with Dp>0.25 μm was 0.52±0.09, similar to the AMS value of 0.54 for smaller particles. The κ values for each aerosol type in the largest PALMS size class (1.13<Dp<= 4.8 μm) were applied to particles with Dp > 4.8 μm."*

**Line 324: Statements such as "The project-wide mean value of κ from Eq. 4 for particles with Dp>0.25 μm was 0.36±0.05." are potentially misleading, because this "kappa value" only applies for a subset of particles in this size class. The grand average kappa over all particles within a size class would be more relevant as a most basic parameter for comparison with other studies.**

*The intent was to compare the kappa for non-refractory particle types derived from the PALMS kappa formulation with that from the AMS/composition model calculations. This is made more clear in the revised text. Calculating a grand average kappa across all types may not be too useful, as models typically carry dust, sea-salt, and other aerosol types as separate species with parameterized hygroscopicity. See response to preceding comment for text.*

**Treatment of the coarse tail of the size distribution taken from the CAS on Lines 255-260 and Lines 330-338: Some duplication of the approach on how water content correction is applied to infer dry size distribution from the measurement at ambient RH. I suggest blend these blocks together. Furthermore, it is not quite clear whether data reduction is done in a single or two step process, i.e. where and how compositional constraints are used. Single step would be directly relating dry size distribution to scattering signal using a forward kernel constrained with observed composition to approximate hygroscopic growth and refractive index as a function of RH (wet size distribution would then be an intermediate side product of the forward kernel applied to the inverted dry size distribution). Two step approach would be: first inferring wet size distribution from scatting signal (done with or without using composition constraints for water content and refractive index estimates?) and then dividing wet diameters by hygroscopic growth factors based on composition constraints (no more need of refractive indices in second step).**

*The two-step data processing for the CAS instrument is the subject of a separate manuscript now in preparation (Dollner et al., 2021), which is part of a student's thesis, so we strongly prefer not to pre-empt this work-in-progress with a detailed description of the process. And it would be a very detailed description, involving Monte Carlo simulations that produce a range of possible wet size distributions that are consistent with the CAS scattering signal, and then using*

*the optical and kappa values in Table 2 to infer the wet size distribution. We have consolidated these sections as requested.*

**Section 2.6: I suggest adding a remark that AMS-derived kappa values are chosen to infer critical dry diameters as these fall into the size range where composition is best constrained with the AMS (Fig. 3).**

*Agreed, and the manuscript is so modified.*

**Line 353: the scattering efficiency also depends on the imaginary part of the refractive index. Why is it not considered? Could possibly be important for the larger and absorbing particles, e.g. dust.**

*In the original manuscript, we treated all particle types as scattering only, and handled rBC as a completely separate component that contributed only absorption (its very small contribution to scattering is ignored). In the revised manuscript, we include an imaginary component for the mineral dust component, so the manuscript is changed to say "refractive index" rather than "real refractive index".*

**Line 360: This sentence belongs into the next paragraph. And I suggest something along the line: "In order to calculate scattering coefficient of the aerosol at fixed RH values of 70, 80, and 85% RH, the effects of hygroscopic growth were considered. The diameter of every particle was adjusted based on the growth factor calculated as described in Sect. 2.5, and the refractive index was adjusted to the volume weighted mean of dry particle and water refractive indices.**

*Agreed. Modified as suggested.*

**Line 366 onwards: I suggest a separate sub-section for the method to calculate absorption including a few introductory sentences on the basic approach behind it, also commenting on which parts of Table 2 and Figure 2 are used for it and which parts are not required.**

*We have created new sub-sections 2.7.1 ("Scattering"), 2.7.2 ("Absorption"), 2.7.3 ("(Extinction) and 2.7.4 "Intensive optical properties" to more clearly separate the topics. We introduce 2.7.2 with the following new text:*

*"The aerosol absorption coefficient (sigma_a, in m-1) is determined for three aerosol components: refractory black carbon as measured by laser-induced incandescence by the SP2 instrument (rBC), brown carbon (BrC) extrapolated from measurements of liquid absorption in aqueous filter extracts, and absorption due to mineral dust particles identified by the PALMS instrument. The absorption for each of these components is calculated differently. Absorption due to rBC is determined using core/shell Mie theory to calculate regionally representative values of absorption per unit mass (mass absorption cross-sections, or MACs) in different airmass types based on the observed size distribution of absorbing cores and the thickness of*

*non-absorbing coatings. These MAC values are then multiplied by the observed 60s-average rBC concentrations to get sigma_a,rBC values. Absorption due to BrC is only roughly approximated, using the liquid absorption measured in aqueous extracts from infrequent filter samples, correcting these values for assumed non-soluble BrC and for aerosolization, and developing a proxy relationship between sigma_a,BrC and measured rBC and BB particle concentrations. Neither rBC nor BrC absorbing components are considered in the calculation any other optical properties for any of the other particle types, for which we use Mie theory assuming homogeneous, uncoated spheres. For mineral dust, a refractive index with a wavelength-dependent imaginary component is applied to the measured, 60s dust size distributions, and sigma_a,dust is explicitly calculated using Mie theory assuming homogeneous spherical particles. Details of the calculations of sigma_a for these three absorbing components follows."*

**Line 374: "We assume that hygroscopic growth on coated BC particles does not appreciably change the absorption coefficient through additional lensing effects, since substantial coatings on the aged BC particles already existed." – This statement implicitly includes expert knowledge that absorption enhancement saturates for thick coatings. Might be useful to state this explicitly. I would support this assumption even outside the saturation range with the following two arguments. Uncoated BC does not undergo hygroscopic growth hence no absorption enhancement. Moderately coated BC will undergo hygroscopic growth, however, opposite effects of increasing shell thickness and decreasing shell refractive index will approximately compensate each other, thereby leaving a small net effect. PS: These assumptions are not perfect, but reliable experimental characterization of humidity dependence of aerosol absorption unfortunately remains an open challenge to the best of my knowledge.**

*The tradeoff between coating thickness and water refractive index is quite involved and would require some additional research and explanation that is probably too involved for this manuscript. We agree that more clearly stating that we are assuming a coating saturation effect is warranted. We now state, "We assume that hygroscopic growth on coated rBC particles does not appreciably change the absorption coefficient through additional lensing effects, since substantial coatings on the aged rBC particles already existed. This assumption is supported by studies that have modeled of the effects of coating thicknesses on BC cores that show a saturation effect as coating thickness increases (e.g., Zanatta et al., 2018)."*

**Line 377: Why is it considered unimportant to treat absorption accurately in dust plumes (where dust contributes substantially to absorption)? E.g. line 46 does not read like dust is not generally unimportant for the ATom data set. (More comments on dust absorption are provided below.)**

*This is a valid point, and we now incorporate estimated dust absorption in the revised dataset and manuscript:*

*"Absorption due to dust particles was calculated simultaneously with the dust scattering calculation using the complex refractive indices at three visible wavelengths for Saharan dust*

*provided by Weinzierl et al. (2011). Based on these measurements we use a refractive index of 1.55+0.002i at a wavelength of 530 nm, with an Ångström coefficient of 3. We assume that water uptake by dust particles does not change the imaginary component of the refractive index; i.e., the absorbing minerals are insoluble, and we assume no lensing effects due to coatings or water uptake. Dust absorption is expected to be significant in the ATom dataset only in discrete dust plumes."*

**Line 389: "[…] approximately account for unmeasured BrC that is extractable in organic solvents […]": this statement is imprecise and should be reformulated to: "[…] approximately account for unmeasured BrC that is not extractable in water […]". The unmeasured BrC may include material that is exclusively extractable in organic solvents but insoluble in water (while it does not include BrC that is soluble in both water and organic solvents). The unmeasured BrC may also include amorphous carbon "tar BrC" that is insoluble, i.e. neither soluble in organic nor polar solvents (e.g. Corbin et al., 2019).**

*Agreed and manuscript modified.*

**For the same reason, "organic-soluble BrC" on line 399 should be replaced by "water-insoluble BrC".**

*Agreed and manuscript modified.*

**Corbin, J. C., Czech, H., Massabò, D., de Mongeot, F. B., Jakobi, G., Liu, F., Lobo, P., Mennucci, C., Mensah, A. A., Orasche, J., Pieber, S. M., Prévôt, A. S. H., Stengel, B., Tay, L. L., Zanatta, M., Zimmermann, R., El Haddad, I., and Gysel, M.: Infrared-absorbing carbonaceous tar can dominate light absorption by marine-engine exhaust. npj Clim. Atmos. Sci., 2, 12, doi:10.1038/s41612-019-0069-5, 2019.**

**Lines 388-390: It is good that first order approximations are made to account for unmeasured BrC and for the difference between mass specific absorption in bulk solution versus airborne particulate form. The conversion factor to infer particulate absorption from bulk solution data implicitly includes particle morphology assumptions and an optical model, likely homogeneous spheres and Mie theory, respectively. This should be reflected in Figure 2 (see separate comments made on Figure 2).**

*The factor of 2 conversion derives from such calculations, but we feel this is appropriately covered in the cited literature (Zeng et al., 2020). We've argued that Fig. 2 should remain a high-level schematic overview of the data processes, and prefer to keep this level of detail out of it.*

**Lines 400-404: Did I understood correctly, that the water extracted absorbance measurement is only used to quantify absorption by BrC at 365 nm? Or was it also used to constrain the AAE of 5, which is used to extrapolate BrC absorption to longer wavelength? If assumed, then provide suitable references, if constrained by measurements, then don't forget to state this (also updating at line 462). Furthermore, absorption by soluble BrC typically vanishes at**

**visible red and NIR wavelength. Is it justified to extrapolate BrC absorption with an AAE of 5 all the way up to NIR wavelength?**

*Yes, the water absorbance measurement was conducted from 300-700 nm wavelengths (line 382), which provided the AAE of 5. The manuscript is modified to state this more clearly. The extrapolation is applied to all wavelengths, but the absorption is vanishingly small above the mid-visible.*

**Figure 4a: Logarithmic axis scaling could possibly provide a better visualization of the level of agreement for lower concentrations.**

*Agreed, and so modified.*

**Figures 4a and 4b: It would be useful to have error bars on the data points (for both calculated and measured values).**

*Agreed, and for 4a applied for a subset of the data points and for 4b all of the data points. The fitted lines have changed slightly because uncertainties in the data have been accounted for in the regression.*

**Line 438: Shouldn't total scattering per component rather than total extinction per component be used as weighting factor for phase function averaging? It may be worthwhile to drop a remark here on how BC-particle contribution to phase function is treated (or neglected). See also above comment.**

*The contribution of rBC to the phase function is ignored, and this is now stated. Now that we are treating dust as an absorbing component, we agree that the weighting should be by scattering rather than extinction, and the manuscript and data are so modified.*

**Line 440-446: Is total scattering in the approach used to calculate the fine mode fraction identical to total scattering calculated with the standard approach? I suspect it comes out slightly different due to different "effective refractive index"? I'm just curious, while I don't see need to adjust anything. Even if slightly inconsistent, it is in the end irrelevant as reported fine mode fraction is a normalized quantity, and the major uncertainty comes from splitting the two size ranges.**

*Yes, they differ because the fitted lognormal does not exactly match the raw data to which it is fit. The composition in the fine mode and coarse mode, which govern the hygroscopicity and the refractive index, are dictated by the composition associated with the modal peak in the fitted lognormal size distribution. We now say, "The refractive index and hygroscopicity of the coarse and fine modes used to calculate [fine mode fraction] is calculated from the volume-weighted mean contribution of each composition class within one geometric standard deviation of the volume modal diameter of that mode."*

**Line 460: Is the approach used to determine the extinction Angström exponent also applied to determine the scattering Angström exponent in equivalent manner?**

*Yes, and now clearly stated. We have also decided to split the Ångström exponent into UV-vis (340-550 nm) and vis-IR (670-1020 nm) fractions. We also note that the end user can calculate the Ångström exponent easily using any wavelength pairs in the dataset.*

**Line 460: Approximating spectral dependence with a power law typically works very well over limited wavelength ranges. However, a rather wide range from 340 nm to 1020 nm is used here, in which case the power law approximation loses out in performance to precisely describe the spectral dependence. As a consequence of this, Angström exponent values become increasingly dependent on the specific approach chosen to infer it from spectral measurements: which wavelength range is covered, which discrete wavelength values are included in the fit, are data log-transformed before fitting, etc.? One peculiar approach to assess sensitivity could be to additionally determine the Angström exponent for two wavelength pairs (e.g. for 340/670 and for 550/1020). If the result is insensitive to choosing different approaches, then it is worthwhile to say so. If not, then it should be stated how exactly the least square fit was done and, more importantly, whether this peculiar approach was on purpose chosen to be identical with standard approaches in e.g. AERONET date processing routines (or some other standard data products or model outputs). BTW: Angström exponent results are not presented or discussed in the main text of the manuscript, correct?**

*We state clearly how we calculate the Ångström exponent over two wavelength ranges, (see answer to the previous question) and don't feel it is necessary to go into more detail. Our choice to do a fit rather than calculating using a specific wavelength pair is just that there are many wavelengths available on different sensors, and picking a specific wavelength pair implies specificity of the results. Fitting over a broader wavelength range implies generality to that portion of the spectrum. The user of the data can of course use the extinction, absorption, and scattering coefficients for the ten wavelengths provided to calculate values using any combination of wavelength pairs or fits, as they wish.*

**Line 461-464: AAE BrC is kept fixed and AAE BC likely isn't too variable either. Thus, overall AAE essentially just reflects the relative amounts of BrC compared to BC, correct? Or does calculated AAE BC exhibit considerable regional variation? In any case, calculated AAE BC should be reported to confirm plausibility of calculated values (after updating the refractive index of BC as requested elsewhere). Actually, I actually wonder whether one should just assume a value of around 1±0.2 for AAE BC. Most importantly, it should be quantified how input uncertainties (e.g. a factor of 3 for BrC absorption; see line 400) propagate through to AAE uncertainty (this should be quite straight forward).**

**Line 474: Here it is argued that dust plumes must also be considered for (A)AOD calculations. Does this go together with the approach to neglect absorption by dust in the calculations?**

*In answer to this and the previous comment, we agree that not including dust absorption was a mistake, and we now explicitly calculate dust absorption using the wavelength-dependent absorption values from Weinzierl et al. (2011) and the PALMS-derived dust size distribution. As a result, AAE will vary more. We still use a BC refractive index that you may consider inappropriate, but which generates good agreement with the photoacoustic absorption measurements at 532 nm (Fig. 4b) and MAC values that are aligned with measurements downwind of African biomass burning plumes (Wu et al., 2021). We calculate new AAEs using these three absorbing components, and can estimate an uncertainty using the very broad BrC uncertainty (factor of 3), the reported uncertainty for the rBC measurement (~30%), and an estimated uncertainty for the dust absorption of ~30%. (A more detailed analysis of the dust absorption uncertainty would require Monte Carlo simulations for a range of possible refractive indices, coupled with size distribution uncertainty, which varies with each 60-s time interval. This is extremely daunting.)*

**Line 478: Interpolation is applied if no more than two vertical layers are filtered due to cloud screening. This seems fine, in particular in the interest of increasing data coverage, except for one potential caveat. Such conditions would be removed via cloud screening from columnar remote sensing measurements. I would expect above-average RH in the cloud-free layers of profiles with clouds in some layers. Could this lead to a systematic RH bias between the data set of this study and remote sensing data sets that may be used for future comparison, or is it likely a minor effect?**

*Yes, we expect that there will be differences between the slantwise, cloud-screened AODs calculated from the in situ data and the scene-filtered AODs from remote sensing measurements. We are reporting slantwise AOD values even in partly cloudy conditions. This makes the interesting point that our AOD estimates are not affected by the clear-sky bias of remote sensors, and so there might be some systematic differences because of this. The ATom AOD estimates are actively being compared with values from the NOAA VIIRS satellite instrument and will be the subject of a future manuscript (S. Wang et al., in prep.).*

*Also, please note that we originally neglected to consider the contribution of the stratospheric aerosol layer to our calculated AOD, which was pointed out in A. Clarke's reply to our response to his initial comments. We have now incorporated satellite-derived AOD at 525 and 1020 nm provided through the comprehensive GLOSSAC dataset. We sample this dataset along the flight track of the DC-8, calculating a mean AOD between the start and end point of each slantwise profile. Typical values for stratospheric AOD were ~ 0.01 at 525 nm, which substantially reduces the bias between the DC-8 profile and AERONET AODs for the lowest AOD values (NASA/LARC/SD/ASDC, 2018; Kovilakam et al., 2020; Yang et al., 2017). There is still a low bias in the DC-8 data, but the normalized mean error over all AERONET comparisons is about -7%. We have added an author, S. Wang, to recognize the contribution of the stratospheric AOD product.*

*Kovilakam, M., Thomason, L. W., Ernest, N., Rieger, L., Bourassa, A., and Millán, L.: The Global Space-based Stratospheric Aerosol Climatology (version 2.0): 1979–2018, Earth Syst. Sci. Data, 12, 2607–2634, https://doi.org/10.5194/essd-12-2607-2020, 2020.*

*NASA/LARC/SD/ASDC: Global Space-based Stratospheric Aerosol Climatology Version 2.0 [Data set]. NASA Langley Atmospheric Science Data Center DAAC, https://doi.org/10.5067/GLOSSAC-L3-V2.0 (accessed: 13 July 2021), 2018.*

*Yang, J.-M.: Towards a combined SAGE II and SCIAMACHY aerosol dataset and implications for stratospheric aerosol trend analysis over East Asia, Atmos. Ocean. Sci. Let., 10, 343-347, https://doi.org/0.1080/16742834.2017.1341812, 2017.*

**Figure 5: When comparing panel d with panel f and panel c with panel e, it looks like considerable covariance of biomass burning and sulfate/organic extinction. Does this suggest that true biomass burning particles do mostly show up in the biomass burning class but also bleed over significantly into the sulfate/organic particle class? Would that affect the interpretation given on lines 495 to 498? More generally, the dust particle class also has quite some co-variance with above two classes, whereas sea salt class exhibits a very distinct spatial pattern. Are source and transport patterns of true biomass burning, sulfate/organic and dust particles more correlated with each other than with sea salt source patterns, or is the applied methodology very good in isolating sea salt particles, while distinction between biomass burning, sulfate/organic and dust is more ambiguous? The authors have much greater experience in strength and limitations of the particle typing approach they applied than the general reader of this manuscript has. Therefore, they should convey their expert interpretation relating to above questions.**

*The ability of the PALMS instrument to clearly discriminate particle types is excellent and examined in considerable detail in Froyd et al. (2019). We are very hesitant to add to an already long manuscript with a qualitative digression that ultimately replicates what is already published.*

**Figure 6:**

**i) Fitting a slope with axis intercept is not a suitable means to assess "overall" closure performance. This type of regression analysis heavily up-weighs higher concentration values and heavily down-weighs lower concentration values. Hence, the regression slope only represents the performance for the higher AOD values. It might be useful to add a sub-panel with a histogram of ratios, possibly even segregated by upper and lower half of AERONET (or DC-8) AOD values.**

*We're not trying to assess closure, we're just performing a "sanity check" to see if our calculated AODs are in the vicinity of the AERONET AOD. We feel that the log-log plot we added (see response to next comment) clearly shows the differences across all AOD values, and does a good*

*job at displaying the unsurprising scatter in the comparison, as well as the overall reasonable agreement.*

**ii) There is always a question whether using linear or logarithmic axis scaling (or both). Linear axis scaling puts visual emphasis on absolute values and errors. In its current form the graph nicely shows that the two methods both agree will in terms of distinguishing low AOD from high AOD. Logarithmic axis scaling puts emphasis on relative errors (and helps for visualizing values varying by several order in magnitude). In this example, choosing logarithmic axis scaling (and adding further grid lines in parallel to the 1:1-line corresponding to fixed ratios) would likely allow for a better visual assessment of closure performance in relative terms at low AOD values, where the fit and 1:1 lines both appear to lay systematically off the data points (little in absolute terms, a lot in relative terms). In the end, axis-scaling type is always a subjective decision. However, discussion of closure results should always clearly distinguish between the two basics questions: "ability to distinguish low from high values" and "level of relative agreement across the full range (or defined sub-ranges) of observed absolute values".**

*We have modified the figure in response to this comment and those of Antony Clarke. Figure 6b now incorporates log-log scaling and lines at 0.5 and 2x the 1:1 line. We also calculate a normalized mean error, which is -7% (DC-8 lower than AERONET).*

**Figure 7: The data points appear to scatter quite symmetrically about the LOWESS fit, in a visualization with logarithmic axis scaling. I have no experience with LOWESS but this fit result looks counter intuitive as "linear" sounds like "putting emphasis on absolute deviation" (unless data were log-transformed before applying LOWESS). Anyway, it looks like the LOWESS provides local modal/median values rather than local mean values. Depending on atmospheric process or on what is to be emphasized with the fit curve, one or the other type of value can be more relevant. It could be considered to provide two types "smoothed fit" to show both local mode and mean (local median and local averaging are simple ways to get such curves). Providing two different curves has the advantage that potential future users of the fit curve, which may want to compare it with their own data, will have to assess which one to choose and how to fit their data to ensure consistency of the comparison.**

*The LOWESS fit is performed just guide to the eye toward the centroid of the highly scattered data, and is a locally-weighted polynomial least-squares regression. It was fitted to the log-transformed data. We now state in the caption that the LOWESS fit is made to the log-transformed data. We think there is not too much value added by providing other smoothing or averaging methods, and, since this dataset will be freely available there is an opportunity for users to apply different averaging that matches their analysis goals. Please note also that the addition of a stratospheric AOD estimate has reduced some of the scatter, bringing the lowest values up.*

**Lines 553 to 556 / thresholds for "in plume" conditions: choosing rather high thresholds, as done here is perfectly suitable to separate conditions where bulk aerosol properties represent the plume aerosol type. However, the "free troposphere" conditions contain, as a**

consequence, all more dilute biomass and dust plumes. It would be interesting to know whether "free tropospheric conditions without BB/dust plume", could be unambiguously isolated with suitable low thresholds. However, this may not be possible as transition between "dilute plume" to "no plume" may be continuous or due to limited sample number. Based on this comment, line 604 should read "[…] exclude data from strong BB and dust plumes […]" (and equivalently in caption of Fig. 11).

*We've slightly revised the definition of BB plumes to require AMS OA concentrations >1 µg m-3, (rather than volume >1 µm3 cm-3), which is quite a bit of OA for oceanic airmasses. We agree on the wording change. We've long conceptualized the atmosphere--even much of the "remote" atmosphere--as composed of plumes/laminae from various sources at different levels of dilution and mixing; the idea of a "background" does not apply except in very specific circumstances, such as regions of the Southern Ocean furthest from land (although even there we encountered BB plumes in the UT and regions of relatively high concentrations of gas-phase pollutants). It's interesting to note that Hodzic et al. (2020) applied a quite rigorous filter (PALMS BB number fraction <0.15) to help identify the most remote air, so there is a lot of "grey area" between "plumes" and "pristine".*

Figure 9: A duplicate of this figure with showing percentage contribution of each particle type (and water) to total extinction shown on the abscissa could be added to the SI. This would visualize relative contributions of different aerosol types at higher altitudes, which is not accessible in the current graph.

*This is a good idea. A new figure is added to the SI.*

Fig. 10b: Scattering by dust is one or two orders in magnitude greater than absorption by BC for dust plumes. This means that absorption by dust could exceed absorption by BC in these plumes, unless dust SSA is really high. This brings me back to earlier comments on the role of dust absorption. Would it be possible to approximately consider dust absorption in the optical model by simply applying literature values of dust SSA to calculated dust scattering (if dust SSA isn't excessively size or source area dependent)?

*See response above about adding dust absorption.*

First paragraph in Sect. 3.3.2: I perfectly agree with the arguments made for need to get composition and size right. Only one small caveat: number size distribution instead of volume size distribution, as presented in Fig. 11, would provide better insight when it comes to CCN and aerosol-cloud interactions. Would it be possible to add particle-type resolved number size distributions to the SI?

*This is a good idea. A new figure is added to the SI. Not surprisingly, the AMS sulfate-organic composition dominates the CCN size range.*

**Lines 640 to 645: I would refrain from interpreting altitude dependence of accumulation mode modal diameter. Cloud processing and wet removal may affect size in addition to condensation.**

*In the tropics and subtropics outside of convective clouds, the net motion is downward, and there has been extensive discussion in the literature regarding condensational growth (see Clarke et al. references, Williamson et al. 2020). While slight, the increase in diameter with decreasing altitude is consistent with this picture and we feel it deserves being mentioned. We will also mention that other effects, including cloud processing/removal and OA removal through chemical aging could affect the large-scale averages shown here.*

**Figures 12 c & f: The GSD of the accumulation mode decreases at altitudes below ~2 km with a concurrent increase of the coarse mode GSD. The latter trend is explained with SSA versus dust. I wonder whether the former is real, or to some extent a fitting artefact which leads to reduced accumulation mode GSD when coarse SSA is present?**

*Inspection of raw (rather than fitted) size distributions shows that this is real. It's probably associated with moist processing in the MBL. It would be interesting to compare with detailed LES models to pinpoint the mechanisms involved.*

**Figure 13: A fair estimate of uncertainties should be added to calculated values shown in this figure. For example, a factor of 3 uncertainty is state for BrC absorption in the methods section, whereas I expect lower uncertainty for absorption by BC. Therefore, the ratio of BC to BrC absorption seen in this figure must not be over-interpreted, whereas single scattering albedo may have relatively small error with little contribution from BrC absorption uncertainty. The authors have a discussion section on limitation, caveats and uncertainties, which is very good to have, and in which they argue against feasibility of error propagation with reasonable effort. This is fair enough; however, some additional guidance of the reader on how to interpret or not to interpret results in one or the other figure could be helpful.**

*A fair point, and we have now added uncertainty bars to Fig. 13 that should guide the reader regarding interpretation of relatively small changes in single scatter albedo.*

**Line 665 to 668: This statement almost motivates an SI figure showing vertical profiles of MAC and MSC.**

*Good suggestion, but the MAC values are derived from the airmass-average rBC Mie calculations (Supplemental Table S6) and therefore don't vary with altitude above the MBL. Also, we are working on a manuscript that looks at the variation with altitude of intensive optical properties and plan to discuss the causes of the variations shown here, so will defer this presentation for the upcoming paper.*

**Technical comments**

**L95: Define acronyms (ATom) at first incident in the text (excluding abstract).**

*Corrected.*

**L111: "ATom"**

*Changed to "Atmospheric Tomography Mission" in the section title.*

**L141-142: Avoid exclusive use of "size" when reporting quantitative numbers. Instead explicitly state "radius" or "diameter".**

*Corrected.*

**L174: please add: […] was then converted to aerosol absorption as described in Section XY.**

*Done.*

**L194: refer forward to Sects. 2.5 & 2.6 for hygroscopic growth and CCN activity and to Sect. 2.7 for optical properties (i.e. for additional simplifications made to infer kappa and refractive index from composition).**

*In response to your earlier comments, we added section 2.3.1, which points to these sections.*

**L208 & 215: "particle volume size distributions"**

*In fact, the mapping of PALMS to the size distribution takes place in number space. PALMS measures the number fraction of particle types in each size class and these are then mapped to the number size distribution. Volume size distributions are calculated from this.*

**L352: use common term for the quantity calculated - "scattering" alone is quite undefined - and provide units (to further minimize potential risk of ambiguity).**

*Changed to "scattering coefficient" (m^-1) and added concentrations units (m^3).*

**Lines 401-403: Calculation of extinction is a bit hidden. I suggest a separate paragraph and including equation number, just to give it a little more weight.**

*Extinction is now described in Sect. 2.7.3, and equation (7) provides its definition.*

**Lines 419-422: This belongs into the previous subsection.**

*This would be redundant there. We changed the introductory clause of the sentence to read, "As described in Sect. 2.7.1, . . . ".*

**Equation 7: "I" on the right hand side of the equation also requires a subscript "i". And, strictly speaking, all θ in the denominator should be replaced by e.g. θ' in order to disambiguate the integration variable, which runs over the range from 0 and π, from the θ in numerator and on the left hand side, which has a fixed value between 0 and π.**

*Done.*

**Equation 10: I suggest to explicitly include wavelength and reference wavelength on the left hand side of the equation.**

*Done.*

**Equation 11: Using "i" as index for layer here and as index for chemical component elsewhere in the manuscript, bears a (small) risk of causing confusion. I suggest using a different index for the layers. With "AOD" on the left hand side, "x" on the right hand side can exclusively be a placeholder for extinction. I suggest to provide the AOD variant of the equation only and comment that AAOD is obtained with substituting extinction by absorption.**

*Changed as suggested.*

**Figure 5: Color scale font size is really at the lower limit.**

*There's really not much room for a larger legend/font without obscuring the data, but we boosted the size as much as feasible.*

**Line 562: "water dominates"**

*Fixed.*

**Line 576: also refer to Fig. 10a**

*Changed as suggested.*

**Figure 10 caption:**

**i) Add a "combustion=HFO-combustion".**

**ii) t should be stated that all rows except BrC Abs and BC Abs represent scattering only (based on the basic assumptions behind the optical calculations).**

**iii) "H2O" could also be expanded to "contribution of light scattering enhancement by particulate water relative to dry particle properties". (Elsewhere, "H2O" is used for water vapour.)**

**iv) Where has the "unclassified" PALMS class gone?**

*We have changed the captions and labels to consistently refer to heavy fuel oil combustion, and have changed "H2O" to "aerosol water". We now state the absorption only is shown for the BrC and rBC components. Unclassified particles are subsumed into the sulfate/organic class and not tracked any further, because sulfate/organic particles are the majority of the particles sensed by PALMS, and most unclassified particles have significant sulfate and organic content. This is now explicitly stated on line 253:*

*"Unclassified particles are combined with and treated as sulfate/organic particles in all further processing." Unclassified particles are no longer mentioned anywhere else in the text.*

**Line 581: I suggest: "[…] absorption from BC, which includes the enhancement by substantial coating as shown to be present by the SP2, is also a significant contributor […]"**

*Done.*

**Figs. 9 and 11 and line ~200: Which PALMS classes are included in "industrial combustion"? Generally, labelling should be harmonized across figures and throughout the manuscript.**

*This refers to heavy fuel oil combustion, which is primarily shipping. Fixed and harmonized.*

**Fig. 12d: Fix Aitken mode color.**

*Good catch! Done.*

**Line 654: I suggest: "The sigma_g of the lognormal distribution is >2 in the lowest 2 km of the profile, where sea salt dominates, but <2 in the middle […]"**

*Changed.*

**Figure 13: Please put emphasis on the wavelength!**

*Done.*

**Line 676: Maybe: "[…] due to the shift of modal diameter to smaller sizes […]"**
*Done.*
* * *
Response to Reviewer #2 Comments

The authors thank the reviewer for the positive overview response to the manuscript and for the constructive comments. The reviewer's comments are in **bold**; our responses are in *italics*.

**This is an excellent manuscript. They have extremely unique measurements in the ATOM campaign with a comprehensive list of state-of-the-art instruments and conducted highly detailed data analysis to derive ambient aerosol properties including composition-resolved size distributions, CCN concentrations, and various optical properties. Understandably, the analysis involves some key assumptions and assumed parameters, which appear to be mostly reasonable to this referee. The manuscript is overall very well written and I do not have major comments. I applaud huge efforts by the authors. Given there are two very detailed comments, I have only a few comments as below. I strongly support the publication of this manuscript in ACP.**

**- For calculations of optical properties of black carbon particles, core-shell Mie theory was applied leading to absorption enhancements by coatings. Some studies indicated that coatings may not enhance absorption as expected by core-shell Mie theory. Refractive index of BC is assumed (Table 2), but this may be subject to uncertainty. Given absorption from coated BC contributes significantly to extinction (L580), can you estimate uncertainties associated with assumed refractive index and morphology on your calculations?**

*In response to this comment and that of Reviewer 1, we have added the following content to the manuscript to explain our choice of core-shell Mie theory and BC refractive index. The bottom line: using this method we get MAC values that result in excellent agreement with the photoacoustic spectrometer that flew in ATom-4, at least in highly absorbing plumes (due to limited sensitivity of the instrument). The estimated uncertainty in the SP2 measurement is ~30% (Fig. 4b), which dominates the uncertainty in the absorption.*

*"Because light absorption by BC is a key uncertainty in global estimates of the direct radiative effect, it is useful to evaluate the assumptions we make in its calculation from SP2 observations. We have calculated absorption assuming that the measured rBC particles are well-aged and compact with a density of $1.8{\times}10^3$ kg m$^{-3}$, and that core/shell Mie theory using the measured coating thickness, assumed to be a non-absorbing organic-sulfate mixture, provides a realistic approximation to their optical properties. Detailed consideration of different modeling approaches (Romshoo et al., 2021) suggests that core/shell Mie theory overestimates MAC values of coated BC by a factor of 1.1-1.5, with values increasing with increasing organic fraction (corresponding to coating thickness), for a fractal dimension for the BC core of 1.7, but with smaller discrepancies as fractal dimension increases toward 3 (a spherical core). Fierce et al. (2020) further report that core/shell Mie theory overpredicts the absorption by BC in measurements in urban outflow, but that this discrepancy can be reduced by accounting for heterogeneity in particle composition and coating thickness. In contrast, Wu et al. (2021) found that core-shell Mie theory provided MAC values in agreement with, or even underestimating, directly measured MACs in aging biomass burning plumes downwind of West Africa. Zanatta et al. (2018) report that core/shell Mie theory slightly underpredicted the measured MAC for aged, coated soot in the Arctic. China et al. (2015) report that aged soot particles measured at a mountaintop site in the Azores had a compact morphology with thin coatings, and that*

*radiative forcing calculated using Mie theory was within 12% of that calculated using the discrete dipole approach.*

*Almost 30% by number of FT particles measured by PALMS during ATom were of BB origin (Schill et al., 2020). Thus we expect both compact core morphologies and a substantial non-absorbing coatings in the rBC particles associated with the BB particles; these characteristics are confirmed by the coating thicknesses measured by the SP2 instrument and by the small values of water-soluble BrC absorption measured in filter extracts (Zeng et al., 2020). The MAC values for rBC we calculate from the ATom dataset using core/shell Mie theory were $14.4 \pm 1.4\ m^2g^{-1}$ at a wavelength of 532 nm averaged over the free troposphere for all four ATom deployments (Supplemental Table S6). Values in identifiable BB plumes were $13.3 \pm 0.4\ m^2g^{-1}$. These values are generally consistent with those measured at 514 nm in West African biomass burning plumes ranging from ~11.3 to ~14.2 $m^2g^{-1}$ for plume ages of ~1 to >9 hr, respectively (Wu et al., 2021). Thus the MAC values we calculate using core/shell Mie theory appear to be reasonable given the likely BB source of most of the rBC. Further, since no observations of soot morphology were made during ATom, we lack a basis for any additional refinement in our estimate of MAC values for coated rBC."*

*China, S., et al. (2015), Morphology and mixing state of aged soot particles at a remote marine free troposphere site: Implications for optical properties, Geophys. Res. Lett., 42, 1243–1250, https://doi.org/10.1002/2014GL062404.*

*Fierce, L., Onasch, T. B., Cappa, C. d., Mazzoleni, C., China, S., Bhandari, J., Davidovits, P., Fischer, D. A., Helgestad, T., Lambe, A. T., Sedlacek, A. J., Smith, G. D., and Wolff, L.: Radiative absorption enhancements by black carbon controlled by particle-to-particle heterogeneity in composition, Proc. Nat. Acad. Sci., 117, 5196-5203; https://doi.org/10.1073/pnas.1919723117, 2020.*

*Wu, H., Taylor, J. W., Langridge, J. M., Yu, C., Allan, J. D., Szpek, K., Cotterell, M. I., Williams, P. I., Flynn, M., Barker, P., Fox, C., Allen, G., Lee, J., and Coe, H.: Rapid transformation of ambient absorbing aerosols from West African biomass burning, Atmos. Chem. Phys., 21, 9417–9440, https://doi.org/10.5194/acp-21-9417-2021, 2021.*

*Zanatta, M., Laj, P., Gysel, M., Baltensperger, U., Vratolis, S., Eleftheriadis, K., Kondo, Y., Dubuisson, P., Winiarek, V., Kazadzis, S., Tunved, P., and Jacobi, H.-W.: Effects of mixing state on optical and radiative properties of black carbon in the European Arctic, Atmos. Chem. Phys., 18, 14037–14057, https://doi.org/10.5194/acp-18-14037-2018, 2018.*

**- Average Kappa_org of 0.18 appears to be a bit higher than previous ambient measurements and modeling (e.g., Gunthe et al., ACP, 9, 7551, 2009; Pringle et al., ACP, 10, 5241, 2010, etc.). You used parameterizations of Rickards et al., which is based on lab measurements of model organic compounds. I wonder the impact and uncertainty**

**associated with this application.**

*The equation for calculating the kappa of OA was chosen based on experience from the SENEX/SEAC4RS campaigns in the southeastern U.S. (Brock et al., 2016a) where, in a heavily OA-dominated region, it successfully replicated the observed hygroscopicity of the aerosol. It also relies on the AMS O/C ratio, which is provided at the 60s resolution needed to construct vertical profiles. Other approaches are of course possible, and as noted the Nakao (2017) paper points out that volatility and solubility--information we don't have--are probably the key properties governing kappa for OA. The relatively high value of OA kappa is conceptually consistent with a very aged organic aerosol (see a more detailed evaluation in Hodzic et al., 2020). Rigorously determining the uncertainty associated with this assumption would mean Monte Carlo simulations allowing the kappa to vary using different formulations, which is extremely costly computationally.*

**- BrC optical properties may change upon chemical aging and photolysis, as shown by recent laboratory studies (e.g., review by Laskin et al., Chem. Rev., 2015). I understand that it is very challenging to accurately estimate BrC optical properties and you do acknowledge relatively large uncertainty (L399). It may be worth to also mention dynamic and complex nature of BrC optical properties with appropriate references.**

*Yes, this is a good point. We have added some discussion on the complexity and uncertainty in BrC properties, although we believe these are well-enclosed by our rough factor-of-three estimate. We've also kept it brief given the already long manuscript.*

*Line 707:*
*"In addition to broad-spectrum absorption by rBC and dust, certain organic species absorb light in blue and near-UV wavelengths; these compounds are referred to as brown carbon (BrC). Most of the BrC in the remote atmosphere is believed to originate from biomass burning (e.g., Washenfelder et al., 2015). Absorption due to BrC may change with time from emission due to photo-bleaching of chromophores or to secondary production of absorbing organic species (e.g., Forrister et al., 2015; Liu et al., 2020). Secondary production is believed to take place near combustion sources, while initial bleaching takes over time scales of a day ( Forrister et al., 2015; Wang et al., 2016; Wong et al., 2019; Wu et al., 2021). However, there is evidence that high-molecular-weight chromophores may persist in aged biomass burning plumes (Di Lorenzo and Young, 2016; Wong et al., 2017).*

*Di Lorenzo, R. A., and Young, C. J.: Size separation method for absorption characterization in brown carbon: Application to an aged biomass burning sample. Geophys. Res. Letts., 43, 458–465, https://doi.org/10.1002/2015GL066954, 2016.*

*Wang, X., Heald, C. L., Sedlacek, A. J., de Sá, S. S., Martin, S. T., Alexander, M. L., Watson, T. B., Aiken, A. C., Springston, S. R., and Artaxo, P: Deriving brown carbon from multiwavelength absorption measurements: Method and application to AERONET and aethalometer*

observations, Atmos. Chem. Phys., 16, 12,733– 12,752, https://doi.org/10.5194/acp-16-12733-2016, 2016.

Wong, J. P. S., Nenes, A., and Weber, R. J.: Changes in light absorptivity of molecular weight separated brown carbon due to photolytic aging. Environ. Sci. Technol., 51, 8414– 8421. https://doi.org/10.1021/acs.est.7b01739, https://doi.org/10.1021/acs.est.7b01739, 2017.

Wong, J. P. S., Tsagkaraki, M., Tsiodra, I., Mihalopoulos, N., Violaki, K., Kanakidou, M., Sciare, J., Nenes, A., and Weber, R. J: Atmospheric evolution of molecular-weight-separated brown carbon from biomass burning, Atmos. Chem. Phys., 19, 7319– 7334, https://doi.org/10.5194/acp-19-7319-2019, 2019.

Wu, H., Taylor, J. W., Langridge, J. M., Yu, C., Allan, J. D., Szpek, K., Cotterell, M. I., Williams, P. I., Flynn, M., Barker, P., Fox, C., Allen, G., Lee, J., and Coe, H.: Rapid transformation of ambient absorbing aerosols from West African biomass burning, Atmos. Chem. Phys., 21, 9417–9440, https://doi.org/10.5194/acp-21-9417-2021, 2021.
* * *
In addition to the changes in response to Reviewers #1 and #2, we have made some additional changes to the manuscript, mainly in the discussion of the ATom dataset in comparison to previous airborne measurements. We have added the following new text:

[revised manuscript text omitted]

---

## Author Response (AR2)

The authors have done a great job in addressing the comments, which I raised in the first round review. From my perspective, the manuscript is ready to be accepted for publication except for very few edits, which either are of technical nature or aim at making some key statements made in the rebuttal directly available in the published article. I can only repeat my original statement: it is a great piece of work in terms of data set, scientific quality, presentation quality and transparency of assumptions and data processing chain behind the final data products.

*Thank you again for the positive comments, and for the very thorough and substantial review that greatly improved the manuscript.*

Minor and Technical Comments

Question and rebuttal concerning potential bias in F_{org} resulting from nitrate being undetected by the PALMS: the rebuttal is perfectly satisfactory. I still recommend to add one or two sentences to the main manuscript making the point that this potential bias is expected to by small based on chemical data from other instruments and nearby size ranges.

*We have added the following sentences to clarify this:*
*"Nitrate mass fraction is not quantified by PALMS for the non-refractory particle classes, but this is likely produces only a minor bias in κ because nitrate concentrations were small (Nault et al., 2021). For example, for submicron sizes, the median AMS nitrate mass fraction was 2.4%, with 25th and 75th percentiles of 0.9% and 4.6%, respectively, when total AMS concentrations were positive."*

Question and rebuttal concerning refractive index and assuming spherical core-shell morphology (→ Mie theory) for calculating MAC of BC: The key argument made in the rebuttal that good closure was achieved with this set of assumptions between calculated and measured absorption, at least when concentration levels were above LOD, should be added to the manuscript. It is important that resulting absorption is as accurate as feasible for this peculiar data set, whereas the study is not designed to assess the optimum assumptions regarding refractive and morphology in general.
*We have added the following:*
*" It is important to note that this study is not designed to evaluate the characteristics of BC refractive index and morphology (e.g., core/shell), but that these parameters are assumed. These assumptions are discussed in more detail in Sect. 4.1.3.*

Question and rebuttal concerning Equation 3: there appears to be a misunderstanding. I was not questioning whether or not Eq. 3 is appropriate to calculate kappa for OA as a function of O/C. Instead I questioned whether this equation dealing with a second order effect is the very one deserving emphasis with being placed on a separate line with equation number, while the ZSR mixing rule, which addresses the first order effect, is "hidden" in text form (lines 403-404 for AMS constrained composition) or not marked as being a specific implementation of the ZSR mixing rule (Eq. 4 for PALMS constrained composition). Small edits to the text could guide the reader even better.

*We have not added an equation for the ZSR formalism, as this is widely understood and we feel is simply described by the terms "volume-weighted" (or "mass-weighted" for the PALMS F_Org method). We have added clarifying statements at the indicated lines pointing out that this we are using the ZSR approach.*

Line 228: …to infer particle hygroscopicity…
*Added this phrase.*

Lines 243-245: Partially redundant to previous paragraph. Is it actually required in the paragraph on light absorption? I suggest to blend it into the previous paragraph, e.g. the sentence starting on line 229.
*Done.*